# Self-Improvement of Large Language Models: A Technical Overview and Future Outlook

## Abstract

As large language models (LLMs) continue to advance, improving them solely through human supervision is becoming increasingly costly and limited in scalability. As models approach human-level capabilities in certain domains, human feedback may no longer provide sufficiently informative signals for further improvement. At the same time, the growing ability of models to make autonomous decisions and execute complex actions naturally enables abstractions in which components of the model development process can be progressively automated. Together, these challenges and opportunities have driven increasing interest in self-improvement, where models autonomously generate data, evaluate outputs, and iteratively refine their own capabilities. In this paper, we present a system-level perspective on self-improving language models and introduce a unified framework that organizes existing techniques. We conceptualize the self-improvement system as a closed-loop lifecycle, consisting of four tightly coupled processes: data acquisition, data selection, model optimization, and inference refinement, along with an autonomous evaluation layer throughout the process. Within this framework, the model itself plays a central role in driving each stage: collecting or generating data, selecting informative signals, updating its parameters, and refining outputs, while the autonomous evaluation layer continuously monitors progress and guides the improvement cycle across stages. Following this lifecycle perspective, we systematically review and analyze representative methods for each component from a technical standpoint. We further discuss current limitations and outline our vision for future research toward fully self-improving LLMs.

## 1 Introduction

Large language models (LLMs) have achieved rapid and consistent performance gains through scaling model size, training data, and compute (Brown et al., 2020; Ouyang et al., 2022; Hoffmann et al., 2022; OpenAI et al., 2024). A widely held assumption underlying this progress is that larger and higher-quality datasets, especially expert-annotated human supervision, lead to stronger models. In practice, methods like Reinforcement Learning from Human Feedback (RLHF) (Ouyang et al., 2022) rely heavily on carefully curated, high-quality supervision to align and refine pretrained models. However, as models continue to advance, the paradigm of improving them primarily through human supervision reveals several structural limitations: (1) Human data scarcity is becoming increasingly evident. High-quality, expert-annotated data is expensive and difficult to scale (Gilardi et al., 2023; Villalobos et al., 2024). The marginal cost of constructing large supervised datasets grows rapidly, while the availability of expert labor remains limited. (2) There is a deeper limitation tied to human cognitive bounds. If model supervision is permanently constrained by human intelligence, can models truly surpass human-level performance? When models approach or exceed human-level capability in certain domains, human feedback may no longer provide sufficiently informative gradients for further improvement (Bowman, 2023; Burns et al., 2023). This raises a fundamental question: how can models continue to improve once they reach parity with their supervisors? Together, these limitations motivate the exploration of model self-improvement as a promising direction. Instead of relying exclusively on external human signals, models may leverage their own capabilities to generate data, evaluate outputs, and iteratively refine their policies.

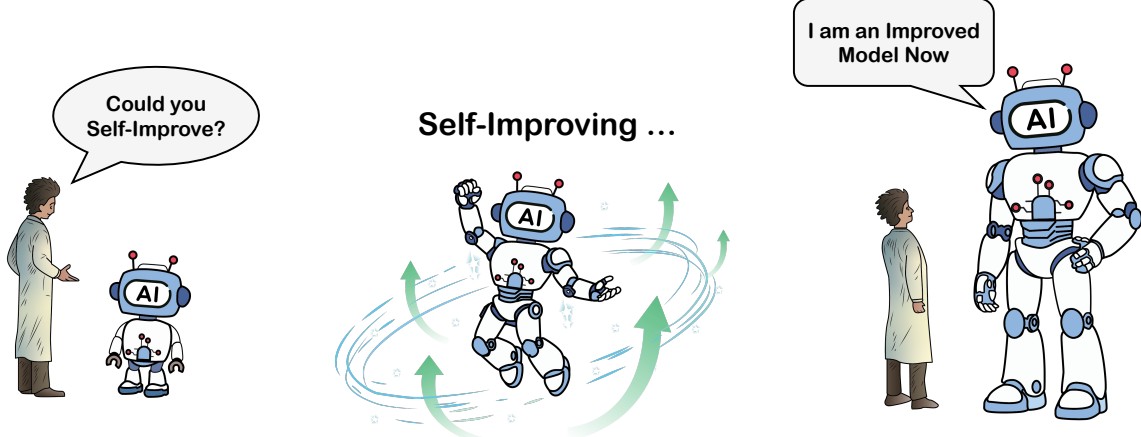

Figure 1: **Vision of self-improved language models.** Humans only bootstrap the system, after which the model autonomously performs many operations such as acquiring data, reflecting on its outputs, and iteratively refining its capabilities to improve itself, potentially enabling the system to evolve beyond human-level intelligence.

From an automation perspective, this direction is not only desirable but natural. As LLMs become increasingly advanced, they have demonstrated the capacity to resolve complex engineering tasks and engage in high-level decision-making. Given that the development process of LLMs, including data acquisition, data selection, and model training, is itself a highly sophisticated engineering endeavor, it is a natural progression to delegate these responsibilities to the models themselves. By utilizing LLMs as intelligent agents to orchestrate their own development lifecycle, a "system-side" self-improvement loop is established. As shown in Figure 1, our vision is to shift from human-driven model development to a paradigm of autonomous self-improvement systems, where LLMs continuously enhance their capabilities through self-directed iteration and feedback.

We define self-improvement of LLMs as a learning paradigm in which a model iteratively enhances its own capabilities without continuous human-in-the-loop supervision. This paradigm is characterized by two essential properties: **Autonomy:** the improvement process operates without ongoing human annotation or manual correction. "Self" does not imply the absence of external components; auxiliary modules such as teacher models, verifiers, critics, reward models, or automated evaluators may still be used. The key requirement is that the learning loop itself is fully automated once deployed. **Continuity:** self-improvement is not a one-off refinement. It is an iterative, self-reinforcing process in which outputs or experiences from earlier stages are reused to generate stronger supervision signals for subsequent updates. Each round of improvement depends on and amplifies prior results, enabling cumulative progress over time. Under this definition, self-improvement is not merely a technique for improving task-level metrics; it is a structural capability that enables sustained, autonomous growth. From the perspective of long-term AI development, such a capability is widely considered central to building systems that continuously learn and adapt beyond their initial training regime.

Motivated by the above vision, as demonstrated in Figure 2, we propose a lifecycle self-improvement system consisting of five interconnected components. The four components: **Data Acquisition**, **Data Selection**, **Model Optimization**, and **Inference Refinement**, jointly address a central question: To build an end-to-end self-improvement system, how can the model itself be leveraged at different stages to drive continuous and autonomous improvement? Specifically:

- **Data Acquisition:** The model autonomously collects or generates its training data.

- **Data Selection:** The model independently evaluates and filters which data points are of higher quality and better suited for its own learning.

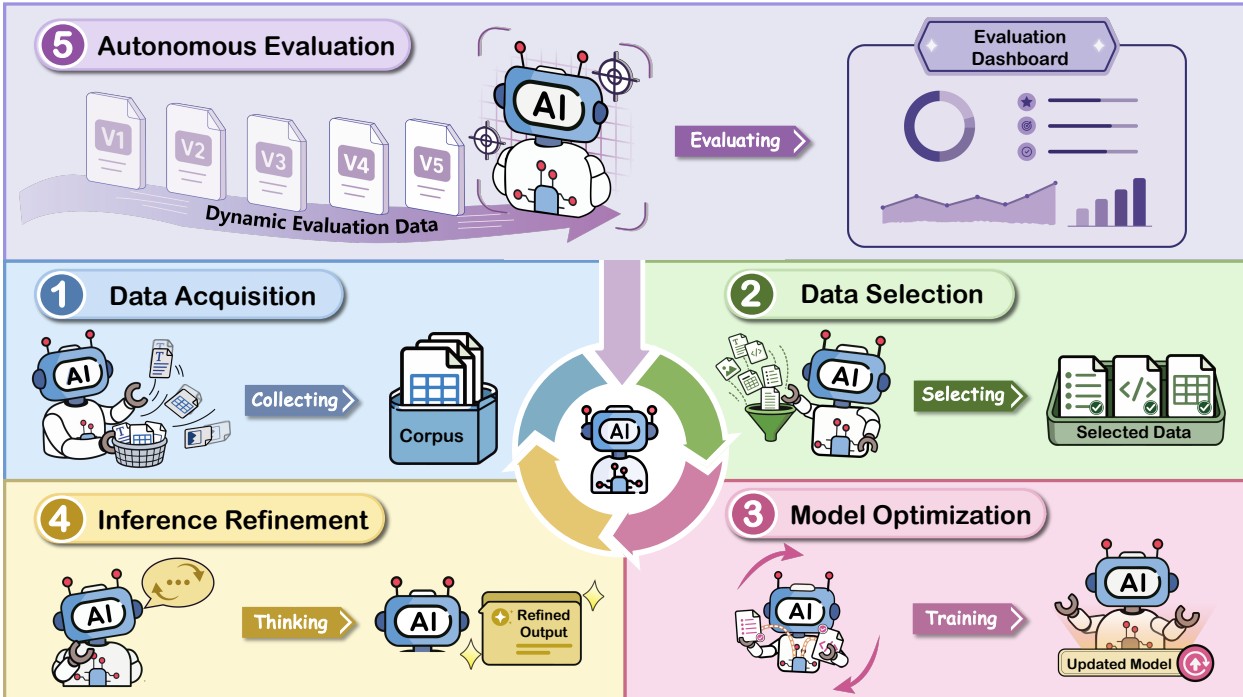

Figure 2: **Overview of the proposed self-improvement system.** The system consists of five interconnected stages: **(i) Data Acquisition** collects or generates candidate training data from external sources, environments, or the model itself, producing raw acquired data; **(ii) Data Selection** filters, ranks, or constructs high-quality subsets from the acquired data, producing selected training data; **(iii) Model Optimization** updates the model using the selected data, producing an improved model; **(iv) Inference Refinement** improves model outputs during inference through search, feedback, or self-correction, producing refined responses; and **(v) Autonomous Evaluation** continuously assesses the model, data, and outputs, producing feedback signals that guide long-term iterative self-improvement.

- **Model Optimization:** The model autonomously learns, effectively converting data into enhanced capabilities within its parameters.

- **Inference Refinement:** The model improves its own performance during the reasoning process without necessitating changes to its underlying parameters.

Beyond these four stages, the system further requires a mechanism for long-term measurement and guidance to ensure that self-improvement remains stable and sustainable. To this end, we introduce the fifth component, **Autonomous Evaluation**, which provides continuous feedback on the model's performance and helps steer its future development. Such a mechanism is essential, since static benchmarks quickly become outdated and human-driven evaluation does not scale with the system's growth. Through autonomous evaluation, the model can maintain timely, adaptive feedback and support sustained long-term improvement.

Together, these five components position the model as the core entity in an automated, iterative loop. This unified system ensures that improvement signals are consistently generated, filtered, applied, refined, and assessed, paving the way for broader system-level self-improvement of LLMs.

Several recent surveys have begun to examine self-improvement from different angles, reflecting the growth of this field. For example, Tao et al. (2024) focus on policy-level self-evolution through self-training and reinforcement learning, while Dong et al. (2024) review inference-time improvement techniques such as prompting and decoding refinement. Meanwhile, Fang et al. (2025a) and ang Gao et al. (2026) emphasize self-evolution of agentic systems, highlighting memory, reflection, and tool-augmented interaction. Despite these efforts, most existing research still concentrates on localized mechanisms applied at specific stages, such

as training or inference, aiming to improve task-level performance, or focuses on peripheral components for agentic improvements. In contrast, we adopt a system-level perspective that conceptualizes self-improvement of the fundamental LLMs themselves as a unified, closed-loop lifecycle, integrating all stages of model development into a coherent end-to-end framework for scalable and autonomous evolution.

The remainder of this paper is organized into two main parts. First, from a technical perspective, we systematically study each component in the self-improvement system (from §2 to §6). For each stage, we begin with an overview to provide a high-level introduction, and then organize existing methods into structured categories, as shown in Figure 3. We further include a discussion at the end of each section to summarize key insights, as well as to analyze how each stage interacts with others and contributes to the overall self-improvement system. Second, we present a more general discussion of the overall self-improvement system (from §7 to §10), including challenges and limitations, potential risks, applications, and future outlook. In these sections, we discuss the system from a broader perspective, beyond individual components. Similarly, the internal structure of each section is organized in a structured manner, as illustrated in Figure 9.

In addition, although our paper is primarily centered on models, we also incorporate works and discussions on self-evolving agents. The key distinction is that model self-improvement mainly focuses on improving the model's internal capabilities through data, optimization, and refinement, whereas self-evolving agents emphasize interactive systems that use tools, environments, memory, and feedback to adapt their behavior over time. Therefore, agent-based self-improvement is most naturally connected to inference-time improvement, where the system can actively plan, interact, revise, and learn from its own execution process. For example, we introduce agentic system-based improvement at inference time in §5.4 and discuss applications of self-evolving agents across domains in §9. We argue that the transition from individual stages to a unified self-improvement system parallels the shift from standalone models to agentic systems, reflecting a shared trend toward more autonomous and interactive learning systems.

## 2 Data Acquisition for Self-Improvement

### 2.1 Overview

Within the self-improvement lifecycle, data acquisition is the process of leveraging the model to autonomously collect or generate the raw materials necessary for its own evolution. Two primary factors underscore the feasibility and necessity of shifting from traditional human-collected datasets toward this model-driven paradigm. First, in terms of operational efficiency, model-driven acquisition overcomes the temporal and financial constraints inherent in manual labor; unlike human curators, models can process and generate data continuously, effectively bypassing the bottlenecks and high costs associated with human bandwidth. Second, and more critically, the intrinsic capabilities of modern LLMs have reached a threshold where the quality of model-generated signals is now highly competitive with human-curated content. In many specialized reasoning or high-complexity tasks, model-sourced data can even surpass human benchmarks in terms of logical consistency and fidelity (Gilardi et al., 2023; Bermejo et al., 2025), providing a superior foundation for continuous improvement. This dual advantage in both scale and quality empowers the model to serve as a self-sufficient engine for its own growth, dictating the scope and nature of the experiences it internalizes. To systematically analyze how models source these raw experiences, as shown in Figure 4, we categorize acquisition mechanisms into three tiers, reflecting a progressive increase in model autonomy and a corresponding decrease in reliance on existing data sources.

- **Static Curation:** The interaction between the model and the "Existing World". As an intelligent agent, the model navigates through massive internet snapshots or databases to autonomously filter out the raw corpora most valuable for its current evolution. In this stage, data is a "fixed stock," and the model's primary role is discovery.

- **Environment Interaction:** The interaction between the model and "Dynamic Tools." The model generates action trajectories by calling APIs, executing code, or operating within simulators, learning from the resulting feedback. Within this paradigm, data is no longer pre-existing; instead, it is "earned" by the model through a process of trial and error.

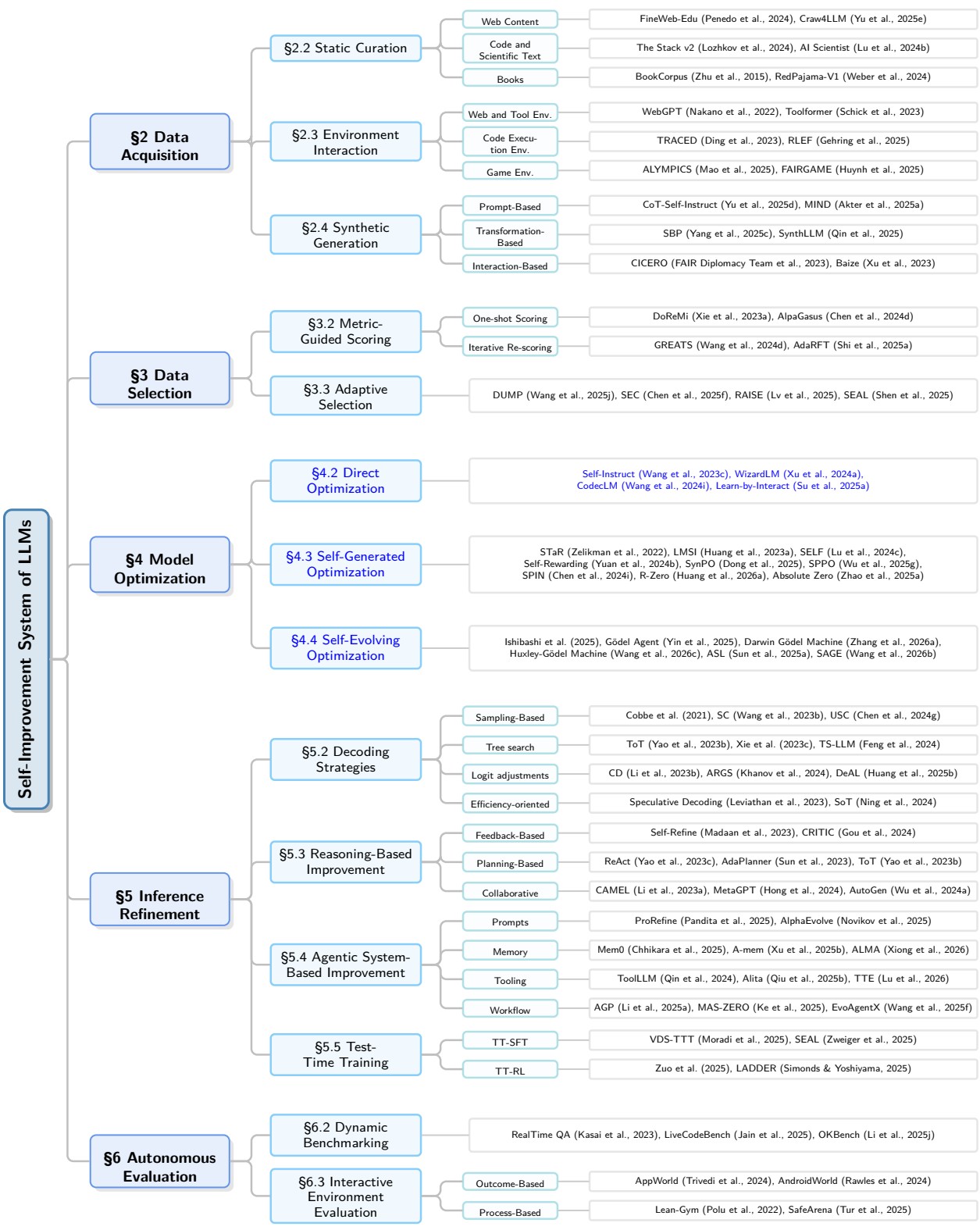

Figure 3: A taxonomy of the self-improvement system of LLMs.

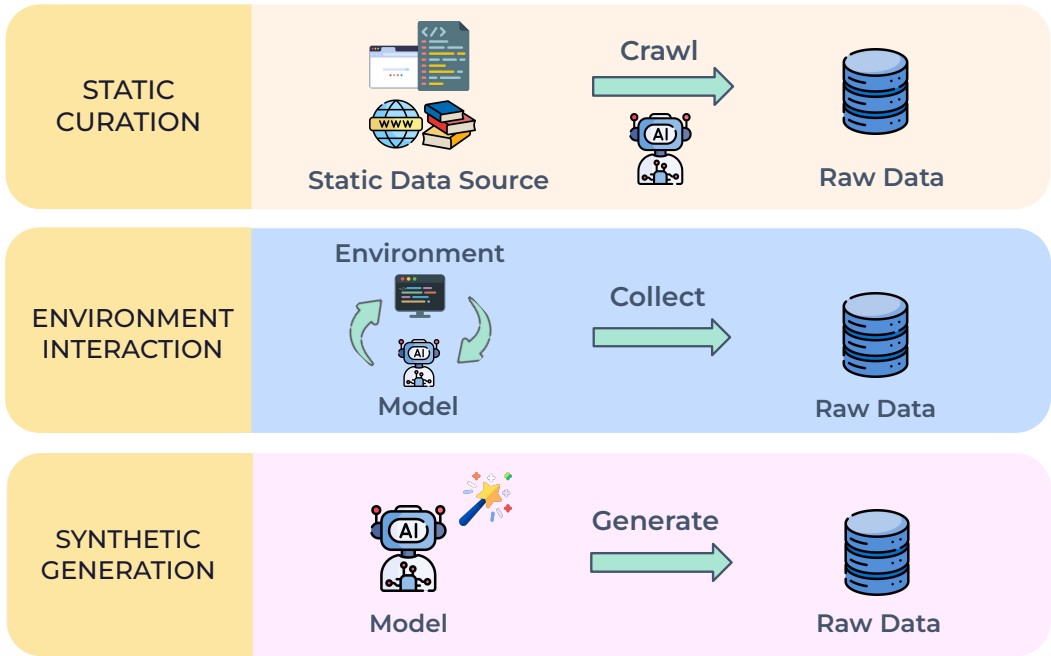

Figure 4: **Overview of data acquisition in the self-improvement system.** This stage focuses on how a model autonomously acquires raw data or experiences that can be used for self-improvement. **(i) Static Curation:** The model collects and filters information from pre-existing datasets or knowledge sources to construct curated training data. **(ii) Environment Interaction:** The model actively interacts with external environments, issuing actions and receiving observations to form interaction trajectories. **(iii) Synthetic Generation:** The model generates new training data directly from its own parameters through prompting, transformation, or multi-model interaction.

- **Synthetic Generation:** The interaction between the model and its "Inner Self". The model completely detaches from external environments, utilizing its intrinsic logic to produce entirely new reasoning chains or instructions. In this scenario, the model no longer depends on external data, instead generating entirely new experience through pure synthesis.

This logical progression, moving from external discovery (curation) to external exploration (interaction) and finally to internal generation (synthesis), outlines the model-driven data acquisition trajectory as a comprehensive spectrum, spanning the curation and exploration of external information sources on one end and the autonomous generation of data from the model's internal capabilities on the other. In the following section, we first introduce static curation in §2.2, where the model leverages existing data sources to construct training corpora. §2.3 then presents environment interaction, which enables the model to acquire data through actions from external environments. §2.4 focuses on synthetic generation, where the model produces new training data based on its own capabilities. Finally, §2.5 concludes with a discussion of their respective roles and trade-offs within the self-improvement system.

## 2.2 Static Curation

Static curation acquires raw data by retrieving content from fixed, externally hosted sources, where the model acts as an intelligent filter navigating massive repositories to discover the corpora most valuable for its own evolution. The core workflow begins with selecting one or more repositories (such as Common Crawl snapshots, code forges, or book collections) and then transforming the retrieved artifacts into a standardized training format. Traditionally, this pipeline has been driven entirely by heuristic rules, hand-written scripts, and tool-based filters. Foundational efforts such as C4 (Raffel et al., 2020), CCNet (Wenzek et al., 2020), RefinedWeb (Penedo et al., 2023), The Pile (Gao et al., 2020), Dolma (Soldaini et al., 2024), and

RedPajama (Weber et al., 2024) established standard practices for web crawling, deduplication, language identification, and quality filtering at scale, collectively producing corpora ranging from hundreds of gigabytes to over one hundred trillion tokens. However, a growing body of work demonstrates that replacing or augmenting these heuristic pipelines with model-driven decisions, where an LLM itself selects, prioritizes, and filters data, produces substantially higher-quality corpora with less waste (Zhou et al., 2026b). We organize these emerging model-guided methods by source type and highlight how each contributes to the self-improvement paradigm.

**Web Content.** The majority of static curation targets the open web, including general web pages, encyclopedic sources such as Wikipedia, community platforms such as StackExchange and Reddit, and multilingual content across hundreds of languages. Traditional pipelines rely on heuristic filters (URL rules, profanity detection, sentence-count thresholds), n-gram language models, and deduplication tools such as MinHash to process raw Common Crawl snapshots into cleaned training corpora (Raffel et al., 2020; Wenzek et al., 2020; Penedo et al., 2023; Weber et al., 2024; de Gibert et al., 2024; Laurençon et al., 2022; Nguyen et al., 2024). While highly scalable, these approaches require extensive human expertise to design filtering rules and cannot adapt to downstream model needs.

A paradigm shift emerged with model-based quality filtering. FineWeb-Edu (Penedo et al., 2024) uses synthetic annotations from Llama-3-70B-Instruct to train a quality classifier that scores documents by educational value, filtering 1.3 trillion tokens down to 280 billion educationally valuable tokens and improving MMLU by 12% and ARC by 24% over unfiltered baselines. DCLM (DataComp for Language Models) (Li et al., 2024b) systematically demonstrates that model-based filtering is the single most important factor in assembling high-quality training sets, enabling a 7B parameter model trained on DCLM-Baseline to reach 64% 5-shot accuracy on MMLU with 40% less compute than the previous state of the art. Most recently, Craw4LLM (Yu et al., 2025e) pushes model guidance even earlier in the pipeline, from filtering to crawling itself: rather than uniformly traversing the web graph and discarding most pages post-hoc, it uses a pretraining influence scorer to prioritize which URLs to crawl, achieving over 95% of oracle pretraining performance while crawling only 21% of the URLs.

These methods illustrate a clear progression in how models participate in web curation: from scoring already-crawled documents (Penedo et al., 2024), to systematically benchmarking model-driven filtering strategies (Li et al., 2024b), to guiding the crawl frontier itself (Yu et al., 2025e). For self-improvement, this trajectory suggests that stronger LLMs could autonomously compose end-to-end web curation pipelines, proposing which domains to crawl, designing quality classifiers tailored to their current capability gaps, and iteratively refining filtering rules to extend their knowledge boundary without human intervention.

**Code and Scientific Text.** Code corpora are typically sourced from public software forges such as GitHub and Software Heritage, while scientific text comes from repositories such as arXiv, PubMed Central, and Semantic Scholar. Traditional pipelines rely on metadata-based filtering (licenses, repository stars, citation counts) and format conversion tools (GROBID for PDFs, regex-based LaTeX extraction) rather than deep semantic analysis (Lozhkov et al., 2024; Weber et al., 2024; Lo et al., 2020; Paster et al., 2024; Gao et al., 2020; Soldaini et al., 2024). While highly scalable, these approaches are sensitive to licensing and documentation quality and cannot assess the semantic relevance of code or scientific content to a model's current training needs.

Model-guided curation of code and scientific text remains nascent but promising. Emerging AI-scientist frameworks demonstrate that LLM agents can autonomously navigate scientific literature, identify relevant papers, extract structured knowledge, and propose experimental designs (Lu et al., 2024b). In the code domain, models could parse repository metadata to identify high-quality projects, propose domain-specific subsets (for example, prioritizing security-critical code or emerging frameworks), and extract structured knowledge from papers such as theorems or experimental results. The metadata-driven nature of current pipelines makes them particularly amenable to agent-based composition and filtering, positioning code and scientific curation as a natural next frontier for model-driven static acquisition.

**Books.** Book corpora provide narrative structure and lexical richness. The Pile (Gao et al., 2020) includes two book subsets: Books3 (processed public-domain and freely available books) and Bibliotik (curated fiction and non-fiction), totaling approximately 100 GB. BookCorpus, used to pretrain the original GPT and BERT, comprises over 11,000 books scraped from self-publishing platforms (Zhu et al., 2015). RedPajama-V1 (Weber et al., 2024) incorporates 26 billion tokens from open book collections by statically crawling Project Gutenberg and similar archives. These pipelines typically apply minimal processing beyond format conversion (EPUB or PDF to plain text) and duplication removal.

Book curation involves copyright-sensitive decisions about which sources to include and how to balance genres and historical versus contemporary coverage. LLMs could help navigate licensing complexities, propose genre-balanced sampling strategies, and identify high-value content in emerging open-access archives. However, model-guided book curation remains largely unexplored, representing an open opportunity for self-improving systems.

Beyond curating which documents to include, a complementary line of work focuses on automating the preparation and transformation of raw data itself. Zhou et al. (2026b) survey this evolving landscape, characterizing the paradigm shift from rule-based, model-specific pipelines to prompt-driven, context-aware, and agentic preparation workflows. They organize the field into three major tasks: data cleaning (standardization, error correction, and imputation), data integration (entity matching and schema matching), and data enrichment (annotation and profiling). LLM-driven methods demonstrate improved generalization and semantic understanding compared to classical tools, yet face persistent challenges around hallucination and the cost of scaling to large corpora. For self-improving LLMs, this paradigm shift is particularly relevant: rather than relying on manually engineered cleaning rules, a model could autonomously identify data quality issues in its own pretraining corpus, propose corrections via structured prompts, and apply enrichment operations (such as generating metadata annotations or resolving entity references) to make raw collected data more useful as training signal.

Across all source types, the vast majority of static curation still relies on tool and script-based pipelines. Common tools include web crawlers, format converters such as GROBID (Lopez, 2009), language identifiers such as fastText (Joulin et al., 2017), and deduplication methods such as MinHash (Broder, 1997). Several common decision patterns emerge: (1) source selection, determining which repositories or archives to crawl; (2) filtering and quality thresholds, choosing among heuristic rules, learned classifiers, and metadata-based ranking; (3) format conversion and parsing; (4) deduplication and normalization; and (5) license and provenance tracking. Each of these decision points represents a task that could be surfaced as an action in a tool-augmented LLM agent (Yu et al., 2025e).

Despite this potential, several challenges limit LLM-guided static curation today. Trust and provenance concerns arise when models autonomously select training data, raising risks of data poisoning and unintentional bias amplification (Bender et al., 2021). System integration remains difficult because most existing pipelines are deeply embedded in institutional infrastructure. Furthermore, static curation lacks immediate feedback signals; models must rely on downstream pretraining performance to assess curation quality, which is expensive and slow. Nevertheless, emerging AI-scientist and data-engineer agent frameworks suggest this automation is increasingly feasible (Lu et al., 2024b; Zhou et al., 2026b). As LLMs become more capable at tool use, code generation, and long-horizon planning, the gap between manual and autonomous static curation will narrow, positioning static curation as the first component in a closed-loop self-improvement system where models actively expand their own pretraining data boundaries.

## 2.3 Environment Interaction

Environment interaction acquires raw data by letting a model act to obtain information, where the collected data includes not only content but also interaction traces: trajectories containing observations (for example, retrieved pages, tool outputs, environment states), actions (for example, search queries, clicks, API calls, code commands), and optional outcomes such as task success signals or execution feedback. This paradigm fundamentally differs from static curation in two ways: **(1) the action–observation loop**, where model actions causally determine what data is generated, creating temporal dependencies and causal structure that are absent in fixed corpora; and **(2) adaptive collection**, where exploration policies can target underrepre-

Table 1: **Environment interaction methods categorized by environment type.** We group representative methods based on the external environments they interact with, including web browsing, code execution, and game environments.

| Environment | Methods |
|---|---|
| Web Browsing | WebGPT (Nakano et al., 2022), Toolformer (Schick et al., 2023), Go-Browse (Gandhi & Neubig, 2025), BrowserAgent (Yu et al., 2025f), InSTA (Trabucco et al., 2025), EnvScaler (Song et al., 2026a) |
| Code Execution | TRACED (Ding et al., 2023), RLEF (Gehring et al., 2025), CWM (FAIR CodeGen team et al., 2025), CodeRL+ (Jiang et al., 2025d), AgentFounder (Su et al., 2025b), Learn-by-Interact (Su et al., 2025a) |
| Game Environments | Supervise Thyself (Racah & Pal, 2019), Generative Agents (Park et al., 2023), ALYMPICS (Mao et al., 2025), FAIRGAME (Huynh et al., 2025) |

sented domains or challenging tasks rather than passively accepting whatever pre-existing text is available. The model thus becomes an active participant in producing its own training data, turning environments into extensions of the training dataset and enabling closed-loop self-improvement.

We organize environment interaction methods by environment type, emphasizing how each domain's interaction mechanism shapes data collection and how these mechanisms can be leveraged by self-improving LLMs, as summarized in Table 1.

**Web and Tool Environments.** Web agents collect data through direct interaction with live websites and external APIs. While static web curation (for example, C4 (Raffel et al., 2020), FineWeb (Penedo et al., 2024)) retrieves fixed snapshots of page content, web browsing and tool-use environments capture the full interactive process of searching, navigating, invoking functions, and synthesizing information across multiple sources. The resulting trajectories encode navigation strategies, multi-step reasoning, tool selection, and task completion patterns that are absent from static HTML dumps.

WebGPT (Nakano et al., 2022) fine-tunes GPT-3 by collecting trajectories from a text-based web browser where the model searches, navigates, and quotes passages to answer questions, using human demonstrations and preference feedback as supervision. Toolformer (Schick et al., 2023) teaches language models to autonomously invoke external APIs such as calculators, search engines, and translators through a self-supervised mechanism that inserts and evaluates candidate API calls; only those calls that reduce language modeling loss are retained, yielding an augmented corpus of tool-using text. Go-Browse (Gandhi & Neubig, 2025) applies structured exploration by framing data collection as graph search over web states, enabling efficient information reuse across exploration episodes and collecting ten thousand successful task-solving trajectories comprising forty thousand interaction steps. BrowserAgent (Yu et al., 2025f) builds an end-to-end browser-native framework that learns from real-time web interactions through fine-grained atomic operations such as scrolling, clicking, typing, and tab management, systematically generating training data from interactive search behaviors rather than relying on static snapshots or external summarization models. InSTA (Trabucco et al., 2025) introduces internet-scale data collection through a three-stage pipeline where an LLM annotates websites with candidate tasks, agents complete those tasks in live environments, and trajectories are filtered by judging task success, operating entirely without human annotation. EnvScaler (Song et al., 2026a) programmatically constructs 191 synthesized tool-interactive environments with approximately seven thousand scenarios through automated synthesis, enabling multi-turn, multi-tool interactions where agents execute tools, observe state changes, and generate trajectories at scale.

For self-improvement, an LLM could iteratively search for questions it answers poorly, launch browsing episodes to collect supporting evidence, invoke APIs to verify factual claims, and then distill these trajectories into additional training data that improves its factual and procedural knowledge (Nakano et al., 2022; Trabucco et al., 2025; Schick et al., 2023).

**Code Execution Environments.** Code executors provide deterministic feedback through program execution, enabling models to ground learning in computational semantics. Unlike static code corpora such as The Stack v2 (Lozhkov et al., 2024) or RedPajama-V1 (Weber et al., 2024), which capture source text as written but lack any record of runtime behavior, code execution environments produce dynamic traces that encode variable states, branch coverage, and test outcomes as programs run. This distinction is critical: static code corpora provide syntactic and structural patterns, whereas execution environments reveal the causal relationship between code and its computational effects.

TRACED (Ding et al., 2023) collects execution traces by running programs in sandboxes, recording runtime variable values and branch coverage, then pretrains code models to predict these dynamic properties from static source text. RLEF (Gehring et al., 2025) trains code LLMs end to end via reinforcement learning to exploit unit test feedback over multiple turns, achieving state-of-the-art results on competitive programming with both 8B and 70B models while reducing required samples by an order of magnitude. Code World Models (CWM) (FAIR CodeGen team et al., 2025) incorporate computational trajectories into mid-training, allocating tokens specifically to interactions with code execution environments and integrating execution feedback directly into the pretraining curriculum. CodeRL+ (Jiang et al., 2025d) advances this by jointly optimizing code generation and execution semantics alignment, where failed exploration programs are repurposed to infer variable-level execution trajectories, providing dense learning signals that bridge the gap between textual fluency and execution correctness.

These approaches show how models can iteratively probe a code executor, collect rich traces, and reuse them as pretraining or continual training data. Notably, AgentFounder (Su et al., 2025b) and Learn-by-Interact (Su et al., 2025a) demonstrate that such environment interaction data can be embedded directly into continual pretraining phases rather than reserved solely for downstream fine-tuning, allocating dedicated training stages to agent trajectory data and improving sample efficiency when models are later adapted to specific tasks. In a self-improvement setting, an LLM could autonomously identify failure patterns in its own code generations, design new test cases, and schedule further execution queries to close capability gaps in targeted languages or libraries (Gehring et al., 2025; Jiang et al., 2025d).

**Game Environments.** Game environments provide structured settings for strategic and social interaction data. Supervise Thyself (Racah & Pal, 2019) examines self-supervised learning where agents observe the results of their actions in interactive game environments to learn representations that generalize to novel settings without explicit reward signals. Park et al. (2023) demonstrate that believable social interactions can unfold between multiple LLM-driven characters in simulated game worlds, producing realistic dialogue transcripts and action logs that can be mined as training examples for social reasoning. ALYMPICS (Mao et al., 2025) introduces a systematic framework that facilitates game-theoretic interactions, where LLMs compete or cooperate in auction-based resource allocation games, generating strategic dialogue and decision trajectories. FAIRGAME (Huynh et al., 2025) provides a modular framework for simulating repeated game-theoretic interactions such as the Prisoner's Dilemma and the Public Goods Game between LLMs, producing trajectories that encode strategic cooperation, defection, and social reasoning patterns.

These game-based systems generate interaction data that capture coordination, negotiation, and strategic planning, which are difficult to obtain from static text alone. For self-improvement, a language model could instantiate multiple copies of itself as players and critics, generate increasingly complex scenarios, and selectively add informative trajectories to its training set, thereby sharpening its social and strategic reasoning capabilities (Park et al., 2023; Mao et al., 2025).

### 2.4 Synthetic Generation

Synthetic generation represents a complementary paradigm to static curation and environment interaction, where models themselves produce training data at scale without environmental feedback or human supervision. Unlike environment interaction, which relies on external systems to provide observations and rewards through action–observation loops, synthetic generation produces data through generative processes guided by carefully engineered prompts, constraints, or diversity mechanisms. In this approach, an LLM (or smaller task-specific model) creates new training examples through prompting, generation templates, or learned procedures, yielding corpora that did not exist in any prior dataset. This paradigm shifts the bottleneck from

Table 2: **Synthetic data generation methods categorized by generation mechanism.** We group representative methods based on how synthetic data is produced, including prompt-based, transformation-based, and interaction-based approaches.

| Generation Mechanism | Methods |
| --- | --- |
| Prompt-Based | TinyStories (Eldan & Li, 2023), Phi-1 (Gunasekar et al., 2023), Phi-1.5 (Li et al., 2023c), Self-Instruct (Wang et al., 2023c), AttrPrompt (Yu et al., 2023), WizardLM (Xu et al., 2024a), Phi-3 (Abdin et al., 2024), Phi-4 (Microsoft Research, 2024), CodecLM (Wang et al., 2024i), CoT-Self-Instruct (Yu et al., 2025d), Cosmopedia (Ben Allal et al., 2024), MIND (Akter et al., 2025a), Constraint-Based (Fedoseev et al., 2024) |
| Transformation-Based | WRAP (Maini et al., 2024), Instruction Pre-Training (Hsieh et al., 2024), SBP (Yang et al., 2025c), SynthLLM (Qin et al., 2025), Gradient Matching (Nguyen et al., 2025) |
| Interaction-Based | Shah et al. (2018), CICERO (FAIR Diplomacy Team et al., 2023), Baize (Xu et al., 2023), SPIN (Chen et al., 2024i), ALAS (Atreja, 2025), LSP (Kuba et al., 2025), Multi-Agent Dialogues (Ueda et al., 2025), Math Gen (Wan et al., 2025), R-Zero (Huang et al., 2026a) |

data availability to synthesis quality and diversity: the challenge is ensuring that generated data maintains sufficient diversity, coherence, and informativeness to serve as effective training signals without inducing model collapse or distributional drift.

We organize synthetic generation methods into three paradigms based on how data is produced, as summarized in Table 2. **Prompt-based** methods generate data from scratch by prompting an LLM, either through direct prompting with topic and format specifications or through seed expansion where a small set of seed examples is iteratively amplified into a large corpus. **Transformation-based** methods take an existing corpus as input and use an LLM to rewrite, reformat, or extract new training examples (for example, converting raw web text into question-answer pairs or rephrasing documents into higher-quality formats). **Interaction-based** methods require multiple model instances (or roles) to interact with each other, generating data through self-play, debate, or multi-agent collaboration without external environmental feedback. Throughout, we emphasize both the mechanisms by which data is generated and how these methods enable self-improvement by allowing models to expand their own training distributions.

### 2.4.1 Prompt-Based Generation

Prompt-based methods generate synthetic data from scratch by providing an LLM with carefully designed instructions specifying the desired topic, format, audience, or task. We distinguish two sub-modes within this paradigm: **direct prompting**, where the model generates content from topic and format specifications alone, and **seed expansion**, where a small set of seed examples or instructions is iteratively amplified into a large corpus.

**Direct Prompting.** TinyStories (Eldan & Li, 2023) demonstrated that coherent language learning emerges in models with fewer than ten million parameters when trained exclusively on GPT-3.5 and GPT-4 generated stories constrained to child-level vocabulary, using constrained word lists and combinatorial sampling over topics and characters to ensure diversity across half a billion tokens. Phi-1 (Gunasekar et al., 2023) pioneered the "textbook quality" approach by generating less than one billion tokens of Python tutorials and coding exercises using GPT-3.5, guided by prompts that specify educational structure, target audience, and domain coverage; combined with six billion tokens of filtered web code, a 1.3 billion parameter model achieved 50.6% accuracy on HumanEval, matching or exceeding models three times larger. Phi-1.5 (Li et al., 2023c) extended

this to natural language and common-sense reasoning by generating twenty billion tokens of diverse synthetic textbooks seeded from twenty thousand curated topics. Subsequent releases confirmed the scaling pattern: Phi-3 (Abdin et al., 2024) series models consistently matched much larger competitors by mixing filtered web text with synthetic content throughout pretraining, and Phi-4 (Microsoft Research, 2024), trained almost entirely on GPT-4-generated text, achieved performance comparable to or exceeding its teacher model on some reasoning benchmarks. AttrPrompt (Yu et al., 2023) explores diversity through attributed prompts that specify length, style, domain, and other properties rather than simple class-conditional prompts, achieving high diversity with only 5% of ChatGPT's querying cost by explicitly controlling generation attributes. Cosmopedia (Ben Allal et al., 2024), the largest open synthetic dataset with twenty-five billion tokens across thirty million files, was generated using Mixtral-8x7B-Instruct by prompting the model to produce textbooks, blog posts, stories, and WikiHow-style articles across multiple domains, demonstrating that large synthetic corpora can be produced reproducibly at scale. MIND (Akter et al., 2025a) generates synthetic math dialogue corpora by prompting models to produce step-by-step problem-solving conversations grounded in OpenWebMath content, boosting mathematical reasoning by 13.4% on GSM8K and 4.3% on MMLU-STEM. Constraint-Based Synthetic Data Generation (Fedoseev et al., 2024) uses Satisfiability Modulo Theories solvers to generate synthetic mathematical problems and solutions, ensuring that problems satisfy formal constraints and that solutions are verifiable.

**Seed Expansion.** Self-Instruct (Wang et al., 2023c) generates instruction-response triples by prompting a pretrained model with 175 seed tasks, sampling new instructions, generating responses, and filtering for quality, producing over 52,000 instructions and demonstrating a 33% absolute improvement over vanilla GPT-3 on SuperNaturalInstructions. WizardLM (Xu et al., 2024a) starts from small sets of human-written instructions and iteratively uses ChatGPT to increase their complexity through "Evol-Instruct," which applies operations such as deepening, broadening, and adding constraints. CodecLM (Wang et al., 2024i) encodes seed instructions into metadata (for example, task type, domain, difficulty), then decodes them back into task-specific synthetic examples by prompting an LLM conditioned on the metadata. CoT-Self-Instruct (Yu et al., 2025d) extends the paradigm with chain-of-thought (CoT) reasoning by prompting the model to generate step-by-step reasoning chains before final answers, producing complex examples that improve both mathematical reasoning and general instruction-following.

These prompt-based methods, whether through direct prompting or seed expansion, illustrate how an LLM can serve as both the generator and the beneficiary of synthetic training data. For self-improvement, the model could autonomously propose new topic lists based on downstream evaluation gaps, refine generation prompts to target weak domains, and iteratively expand its training corpus without relying on external data sources.

### 2.4.2 Transformation-Based Generation

Transformation-based methods take an existing corpus as input and use an LLM to rewrite, reformat, or extract new training examples from it. Rather than generating content from scratch, these methods leverage the semantic content of existing data while improving its quality, structure, or task alignment.

WRAP (Web Rephrase Augmented Pretraining) (Maini et al., 2024) uses instruction-tuned LLMs to paraphrase web documents into diverse formats such as Wikipedia-style articles or question-answer pairs; jointly pretraining on original and rephrased web data accelerates convergence approximately three-fold while improving zero-shot accuracy. Instruction Pre-Training (Hsieh et al., 2024) synthesizes two hundred million instruction-response pairs from pretraining corpora using open LLMs by prompting with diverse task templates, enabling an eight billion parameter Llama model to rival or exceed a seventy billion parameter baseline on many benchmarks. SBP (Synthetic Bootstrapped Pretraining) (Yang et al., 2025c) learns inter-document relationships from a seed corpus and uses them to generate an entire new corpus of documents, enabling generation of up to one trillion tokens that consistently improve over baselines while recovering approximately 60% of the gains that would otherwise require twenty times more unique natural data. SynthLLM (Qin et al., 2025) automatically generates large-scale synthetic question-answer datasets from pretraining corpora through concept extraction and recombination, revealing that synthetic data follows predictable scaling laws and can substitute for billions of real tokens before saturation. Gradient Matching (Nguyen et al., 2025)

formalizes synthesis as a learning problem by proposing a gradient-matching algorithm that optimizes synthetic examples to mimic real training gradients, yielding provably convergent synthetic examples that allow LLMs to reach the same solution as using original data.

These transformation methods show how models can reprocess existing corpora into higher-quality training data. In a self-improving LLM, this capability enables the model to revisit its own pretraining corpus, identify low-quality or redundant segments, and synthesize improved versions that better serve its learning objectives.

### 2.4.3 Interaction-Based Generation

Interaction-based methods require multiple model instances or roles to interact with each other, generating data through self-play, debate, or multi-model collaboration without external environmental feedback. These represent the most autonomous form of synthetic generation: models create their own training curricula by competing, cooperating, and critiquing their own outputs.

Shah et al. (2018) demonstrated that conversational agents could be bootstrapped via self-play dialogues, where two instances of the agent simulate task-oriented conversations to generate synthetic training data, yielding fully annotated dialogues without a large human-written corpus. Xu et al. (2023) use model self-chat to generate multi-turn dialogues where the model is prompted to play both user and assistant roles, generating synthetic dialogues that demonstrate strong conversational ability from entirely model-generated interactions. CICERO (FAIR Diplomacy Team et al., 2023), trained for strategic Diplomacy gameplay, was fine-tuned on millions of messages from games where model instances negotiated with each other, recording negotiation dialogues and action sequences and filtering trajectories that exhibit human-compatible negotiation tactics. SPIN (Self-Play Fine-Tuning) (Chen et al., 2024i) utilizes a self-play mechanism where the LLM generates its own training data from previous iterations, refining its policy by discerning self-generated responses from human-annotated data and progressively elevating the LLM without additional human data. ALAS (Atreja, 2025) constructs an entire learning curriculum by querying the web, synthesizing question-answer pairs from retrieved content, and continuously fine-tuning itself, combining retrieval-augmented generation with self-critique where the model evaluates its own generated questions and answers before adding them to the training set. LSP (Language Self-Play) (Kuba et al., 2025) introduces a competitive game-theoretic framework where a single LLM iteratively improves by playing Challenger and Solver roles against itself, generating increasingly difficult queries and learning to solve them through reinforcement learning without external training data. R-Zero (Huang et al., 2026a) presents a reinforcement-learning-driven self-play framework where a model iteratively generates challenges and solves them, demonstrating that pretrained LMs can improve without external data via interactive self-play curricula.

Multi-model dialogue generation further extends interaction paradigms. Research on multi-model LLM dialogues demonstrates that enlarging cohorts, deepening interaction depth, and broadening persona heterogeneity each enrich the diversity of generated ideas, with increasing critic-side diversity within ideation–critique–revision loops boosting the feasibility of final proposals (Ueda et al., 2025). Math problem generation benefits from dual mechanisms: self-play combined with multi-model cooperation generates synthetic math problems through continual learning cycles of supervised fine-tuning, data synthesis, and direct preference optimization (Wan et al., 2025).

For self-improvement, interaction-based methods enable an LLM to iteratively challenge itself, discover failure modes, and generate corrective examples without relying on external data sources or human supervision. Moreover, some of these methods (for example, R-Zero, SPIN, LSP) directly connect to policy optimization through reinforcement learning loops, illustrating how synthetic generation can serve as both a data acquisition and policy refinement mechanism.

### 2.5 Discussion

The three acquisition pathways described above: static curation, environment interaction, and synthetic generation occupy complementary positions in the data landscape. Rather than competing alternatives, they form a layered system in which each pathway addresses limitations of the others. We discuss the trade-

offs that govern their use, how their roles shift across the training lifecycle, and how they combine to enable autonomous data acquisition for self-improving LLMs.

### 2.5.1 Trade-offs Across Pathways

Each pathway presents characteristic trade-offs that pipeline designers must balance.

- **Scale versus specificity.** Static curation offers massive scale (hundreds of trillions of tokens (Weber et al., 2024)) but limited targeting; content reflects what exists on the open web. Synthetic generation offers precise control over topic, difficulty, and format, but requires careful diversity management to avoid redundancy and distributional narrowing (Chen et al., 2024b; Yang et al., 2025b). Environment interaction offers high specificity—model actions causally determine what data is generated—but at significant execution cost per trajectory.

- **Quality versus cost.** Synthetic data from strong teacher models (for example, GPT-4) produces high-quality outputs but incurs substantial inference costs, while generation from open models (for example, Mixtral, Llama) is cheaper but may yield lower-fidelity examples (Ben Allal et al., 2024). Static curation is cheap per token after initial infrastructure investment, but filtering for quality via learned classifiers (Penedo et al., 2024; Yu et al., 2025e) adds computational overhead. Environment interaction incurs the highest per-sample cost due to runtime execution, but the resulting trajectories carry dense, verifiable learning signals.

- **Diversity versus coherence.** Synthetic corpora risk distributional narrowing if generation prompts or seed topics are insufficiently varied, potentially leading to model collapse (Dohmatob et al., 2024). Maintaining diversity requires explicit mechanisms such as entity grounding (Yang et al., 2025b), attributed prompts (Yu et al., 2023), and periodic infusions of real-world data. Static corpora naturally capture the diversity of the open web but include noise, toxicity, and duplicates that must be filtered.

- **Substitutability.** An important question is whether one pathway can replace another. BeyondWeb (DatologyAI et al., 2025) and the Phi model family (Gunasekar et al., 2023; Microsoft Research, 2024) demonstrate that carefully synthesized data can partially substitute for large-scale web crawls, achieving comparable or superior performance with substantially fewer tokens. However, Kang et al. (2025a) show that book-style synthetic data alone exhibits model-collapse patterns at small data budgets, whereas rephrased synthetic data mixed at approximately 30% with natural web text accelerates convergence five to tenfold without degradation. Environment interaction data is largely non-substitutable: the causal structure of execution traces (Ding et al., 2023; FAIR CodeGen team et al., 2025), browsing trajectories (Nakano et al., 2022; Trabucco et al., 2025), and tool-use logs (Schick et al., 2023) cannot be replicated through static text or pure generation, because the data must reflect genuine interactions with external systems. In practice, the strongest training pipelines combine all three pathways rather than relying on any single one.

### 2.5.2 Shifting Roles Across the Training Lifecycle

The three pathways do not contribute equally at every stage; rather, their relative importance shifts as the model progresses from pretraining through post-training. Understanding this progression is essential for designing effective data acquisition strategies.

During early pretraining, static curation dominates because sheer token volume and broad linguistic coverage are the primary requirements (Raffel et al., 2020; Penedo et al., 2024; Weber et al., 2024). The goal at this stage is to establish a general-purpose language foundation across diverse domains and genres. As pretraining progresses and the model acquires basic language competence, synthetic generation becomes increasingly valuable for injecting structured, high-quality supervision—such as book-style content, rephrased web data, or reasoning-intensive examples—that accelerates convergence and strengthens targeted capabilities (Gunasekar et al., 2023; Maini et al., 2024; Li et al., 2023c). Recent evidence reinforces this phase: front-loading reasoning-intensive data in pretraining yields a 19% average gain on downstream benchmarks,

establishing foundational capabilities that post-training alone cannot recover (Akter et al., 2025b). Moreover, pretraining benefits most from broad diversity in reasoning patterns, while supervised fine-tuning is more sensitive to data quality (Akter et al., 2025b), suggesting that different data acquisition strategies are optimal at different stages.

During continual pretraining and post-training, environment interaction takes on a larger role, providing the grounded, task-specific trajectories needed for tool use, web navigation, code execution, and strategic reasoning (Su et al., 2025b;a; Gehring et al., 2025). Instruction bootstrapping and self-play methods then refine behaviors for alignment and preference optimization (Wang et al., 2023c; Chen et al., 2024i). This lifecycle perspective suggests that a self-improving LLM must not only have access to all three pathways but must also orchestrate them in a curriculum-aware manner, shifting the data mix from broad coverage to targeted synthesis to grounded interaction as its capabilities mature.

### 2.5.3 Toward Autonomous Data Acquisition

From the perspective of self-improving LLMs, data acquisition is the stage where the model expands its own knowledge boundary. The three pathways described in this section collectively provide the mechanisms for this expansion, and the emerging trend is toward systems that orchestrate all three autonomously.

In the simplest form, an LLM can direct its own static curation by autonomously discovering and prioritizing new web sources, code repositories, or scientific archives based on identified knowledge gaps, as demonstrated by LLM-guided crawling approaches (Yu et al., 2025e). Moving beyond passive retrieval, environment interaction enables the model to generate its own grounded training data through active exploration, turning web browsing (Trabucco et al., 2025; Gandhi & Neubig, 2025), code execution (Gehring et al., 2025; Jiang et al., 2025d), and tool invocation (Schick et al., 2023) into data-producing actions whose outcomes directly inform what the model learns next. Synthetic generation further enables the model to fill identified capability gaps by producing targeted training examples, with self-play mechanisms such as R-Zero (Huang et al., 2026a) and Language Self-Play (Kuba et al., 2025) demonstrating purely autonomous improvement without any external data. Recent systems such as ALAS (Atreja, 2025) illustrate how these mechanisms can be composed: the model generates a learning curriculum for a target domain, retrieves up-to-date information from the web, distills it into question-answer training data, fine-tunes itself through supervised fine-tuning and direct preference optimization, and iteratively revises the curriculum based on evaluation results.

### 2.5.4 Initiation of the Self-Improvement Loop

Within the lifecycle, data acquisition is the entry point of the self-improvement loop, because self-improvement presupposes experiences to learn from: no later stage—selection, optimization, refinement, or evaluation—has anything to operate on until data has been acquired. Every iteration therefore begins with the model expanding the pool of experiences available to it, and everything the system later selects, learns from, or evaluates ultimately traces back to what is acquired here. The three pathways described in this section jointly constitute the input layer of this loop. The envisioned pipeline operates as follows: the model continuously assesses its own weaknesses, decides which acquisition pathway to invoke (crawling new sources, exploring an environment, or synthesizing targeted examples), collects new data accordingly, and hands the raw acquired data to the downstream stages of the system. Because it stands at the head of the data flow, the properties of the acquired data, such as its scale, diversity, difficulty, and noise, propagate through the entire loop and bound what any later stage can achieve. This positions autonomous data acquisition not as a one-time preprocessing step, but as a persistent, model-driven process that restarts the loop at each iteration and runs throughout the lifetime of a self-improving system, enabling the model to continuously expand the scope and quality of its own training distribution.

## 3 Data Selection for Self-Improvement

### 3.1 Overview

Within the lifecycle of a self-improvement system, data selection serves as the critical bridge between raw data acquisition and model optimization. Selection governs the screening mechanisms that decide which

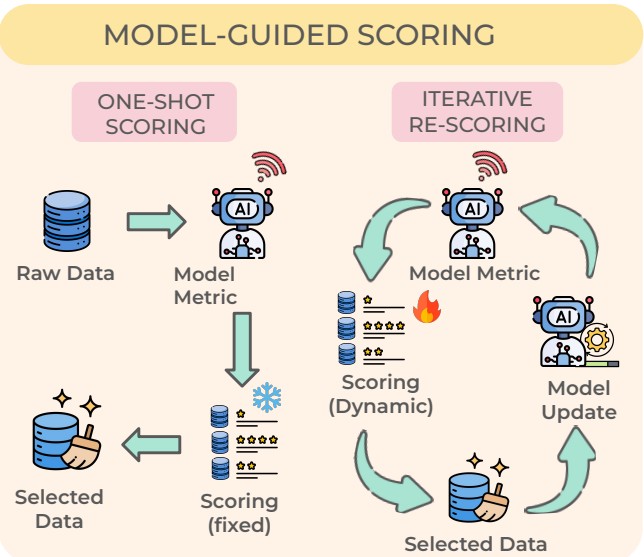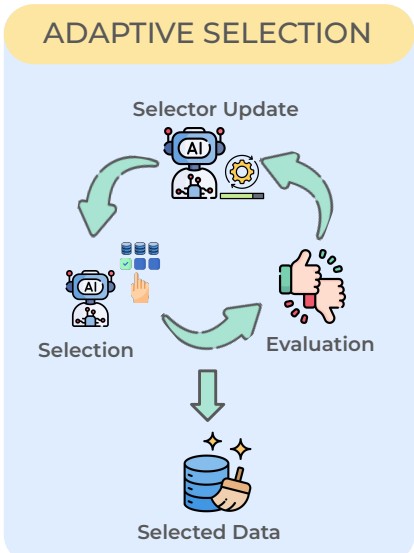

Figure 5: **Overview of data selection in the self-improvement system.** This stage focuses on how a model evaluates and selects high-quality data from raw data pools to support effective self-improvement. **(i) Metric-Guided Scoring:** The model applies predefined scoring metrics derived from model signals to rank and filter data. One-shot scoring computes scores once before training, while iterative re-scoring periodically refreshes scores as the model evolves. **(ii) Adaptive Selection:** A learnable selector dynamically chooses training data through an iterative loop of selection, evaluation, and selector update, continuously adapting the training distribution based on feedback.

specific data samples are filtered and prioritized for the model to learn from. Historically, this stage relied on static filtering, where human-defined heuristics, such as language or script filtering (Wenzek et al., 2020; Gao et al., 2020), deduplication and near-duplicate removal (Brown et al., 2020), elimination of templated or non-linguistic content (Raffel et al., 2020; Gao et al., 2020), and coarse constraints like minimum-length thresholds or domain blocklists (Raffel et al., 2020), were applied. However, these methods are fundamentally model-agnostic; they cannot perceive the model's current internal state nor adapt as the model evolves, leading to a lack of both automation in signal generation and continuity across iterations.

To achieve a fully autonomous self-improvement loop, data selection must shift toward a model-driven paradigm, leveraging the model's own internal states to curate its training signals. As shown in Figure 5, this transition unfolds across two progressively sophisticated regimes:

- **Metric-Guided Scoring** transforms the model from a passive recipient into an active evaluator by replacing fixed heuristic rules with model-derived signals, such as loss, perplexity, or scores from automated evaluators. Instead of relying on handcrafted filters, the system leverages the model itself to assess data quality and utility. These scores may be computed once (one-shot) or updated alongside the model's evolving state (iterative), with the latter enabling more continuous alignment between selection and learning needs.

- **Adaptive Selection** represents the higher level evolution of this process, where a dedicated selector acts as a "strategist" by optimizing the selection logic itself. Here, data selection is no longer a fixed preprocessing step but a parameterized, learnable task. Using frameworks such as reinforcement learning, bandit optimization, or bilevel optimization, the system trains this selector to discover which data distributions or sample compositions maximize performance gains. By optimizing the selection policy end-to-end toward a downstream objective, the system moves beyond fixed decision rules and learns to "shape its own curriculum."

This progression from model-metric scoring to strategic adaptation closely parallels the shift from "external curation" to "self-synthesis" in data acquisition. It ensures that collected data is not merely accumulated but actively refined to match the model's evolving capabilities, moving toward the automated, continuous feedback loop envisioned for self-improving systems. In the following section, §3.2 presents metric-guided scoring, where data is evaluated using model-derived signals such as loss or uncertainty. §3.3 focuses on adaptive selection, where the selection strategy itself is learned and updated to optimize downstream performance. Finally, §3.4 concludes with a discussion of how selection influences training dynamics within the self-improvement system.

### 3.2 Metric-Guided Scoring

Metric-guided scoring uses model-derived signals to score candidate data while keeping the decision rule itself fixed (i.e., not learned or updated). Signals may originate from model internals (e.g., loss, perplexity, uncertainty) or external automated evaluators (e.g., verifiers, critics, reward models). Selection can be applied either **offline in a one-shot manner** or **online through iterative re-scoring**, and may produce a filtered subset, sample weights, or dynamically selected mini-batches. This regime offers strong automation, as scoring relies on model-generated signals rather than human-designed heuristics. Its degree of continuity, however, depends on when selection is applied: one-shot scoring provides limited continuity, whereas iterative re-scoring enhances continuity by repeatedly updating scores as the model evolves. Accordingly, we categorize metric-guided scoring methods into one-shot and iterative variants. Table 3 summarizes representative methods according to their scoring signal and selection setting.

### 3.2.1 One-Shot Scoring

One-shot scoring refers to a metric-guided scoring setting in which selection decisions are computed once prior to the target model's optimization and remain unchanged throughout training. A scoring function assigns a utility value to each candidate unit—which may be an individual instance, a pre-defined group of instances, or an entire domain/source. The final training set is then constructed by filtering, ranking, or reweighting according to these scores (Chen et al., 2024d; Li et al., 2024k; Xie et al., 2023b). Here, "offline" is defined relative to the target optimization loop: although the scoring process may use a frozen reference model or a separately trained proxy model, the resulting subset or weights are fixed and not updated during the subsequent training of the target model.

Representative one-shot scorers mainly differ in what their fixed utility signal is intended to approximate, while the resulting scores can be applied at different granularities (e.g., instances or domains) to produce a filtered subset or sampling weights. **Likelihood-based** signals use perplexity and related model-fit measures as proxies for text quality and utility, enabling offline pruning and ranking of candidate data (Marion et al., 2023; Ankner et al., 2025); this signal can also be adapted to measure how useful a candidate instruction is as an in-context example, by evaluating how much it reduces anchor-set perplexity when used as a demonstration (Li et al., 2024k), and can also be aggregated at the domain-level via perplexity–benchmark correlations to prioritize domains with higher downstream payoff (Thrush et al., 2025). **Loss- and training-dynamics-based** scorers treat utility as learnability captured by training dynamics, leveraging proxy loss-trajectory similarity to form gradient-representative subsets (Yang et al., 2024c) and learning domain mixture weights by minimizing worst-case excess loss before resampling (Xie et al., 2023a). Gradient- and influence-based approaches approximate a sample's marginal effect on target-task validation loss via first-order gradient alignment, selecting examples whose update directions align with validation gradients (Xia et al., 2024), using influence-preserving proxies to make gradient-based selection scalable (Chen et al., 2026).

**Difficulty- and hardness-based** signals operationalize utility as selecting challenging yet informative instructions, using instruction-following difficulty scores (Li et al., 2024e), learning-percentage-based hardness estimated by smaller models (Mekala et al., 2024), or token-level informativeness and neighborhood consistency for robust scoring (Fu et al., 2025). **Uncertainty-based** scorers rely on intrinsic predictive uncertainty for self-filtering of instruction data (Liu et al., 2024d), while graph-based approaches further integrate uncertainty into influence maximization to model dependencies among examples (Han et al., 2025). **Similarity-based** selection treats utility as relevance to a target distribution or task, retrieving cross-task

Table 3: **Metric-guided scoring methods based on model metrics.** This table summarizes representative methods according to the metrics they use for data selection (e.g., perplexity, loss, gradient, difficulty, uncertainty, similarity, and external evaluators), and distinguishes two settings in metric-guided scoring: one-shot scoring (offline) and iterative re-scoring (online).

| Model Metric | One-Shot Scoring (Offline) | Iterative Re-scoring (Online) |
|---|---|---|
| Perplexity | Marion et al. (2023), NUGGETS (Li et al., 2024k), Perplexity-based Pruning (Ankner et al., 2025), Perplexity Correlations (Thrush et al., 2025) | SST (hattami et al., 2025), PREPO (Sun et al., 2025b) |
| Loss | DoReMi (Xie et al., 2023a), S2L (Yang et al., 2024c) | RHO-LOSS (Mindermann et al., 2022) |
| Gradient | LESS (Xia et al., 2024), IProX (Chen et al., 2026) | GREATS (Wang et al., 2024d), LearnAlign (Li et al., 2025d) |
| Difficulty | Li et al. (2024e), Learning Percentage (Mekala et al., 2024), T-SHIRT (Fu et al., 2025) | IT2ACL (Huang & Xiong, 2024), P3 (Yang et al., 2024b), AdaRFT (Shi et al., 2025a), AdaSTaR (Koh et al., 2025), Bae et al. (2026) |
| Uncertainty | SelectIT (Liu et al., 2024d), UniMax (Han et al., 2025) | Recency Bias (Song et al., 2020), Active Instruction Tuning (Kung et al., 2023) |
| Similarity | DEFT (Ivison et al., 2023), DSIR (Xie et al., 2023b) | DiverseEvol (Wu et al., 2023), Balanced LSH (Phan et al., 2025) |
| External Evaluator | GLaM (Du et al., 2022), PaLM (Chowdhery et al., 2023), AlpaGasus (Chen et al., 2024d), DEITA (Liu et al., 2024g) | IterSelectTune (Song et al., 2024), LANCE (Wang et al., 2025b), Auto-CEI (Zhao et al., 2025d) |

nearest neighbors (Ivison et al., 2023) or performing feature-space importance resampling for large-scale distribution matching (Xie et al., 2023b). Finally, **external-evaluator-based** scoring leverages separate quality/safety judges—including text-quality classifiers in large-scale pretraining pipelines (Du et al., 2022; Chowdhery et al., 2023) and strong LLM filters or multi-criteria alignment judgments for instruction data (Chen et al., 2024d; Liu et al., 2024g).

Compared to static filters, one-shot scoring provides stronger automation, as scoring and selection are performed using model-derived metrics. However, its continuity remains limited, since selection decisions are made only once and remain fixed throughout training. Similar to static filters, this regime is attractive at scale due to its low computational overhead and operational stability. Its primary limitation lies in adaptivity: a single utility estimate may become outdated or misaligned with the model's evolving competence and training objectives. This limitation motivates methods that adopt iterative re-scoring, which repeatedly update selection signals during training to better track the model's learning dynamics.

### 3.2.2 Iterative Re-Scoring

Iterative re-scoring brings metric-guided scoring selection into the target training loop: scores are recomputed repeatedly using updated model states and the same non-learned decision rule is re-applied to shape subsequent training batches (Song et al., 2020; Wang et al., 2024d). In this setting, the rule remains fixed, but the selected subset changes over time because the scoring signal changes as training progresses. Consequently, automation remains strong, while continuity is strengthened because selection is explicitly re-applied across iterations.

Representative iterative re-scoring methods primarily differ in how refreshed scores are injected into the optimization loop, even when the scoring rule itself remains fixed. Broadly, re-scored utilities can be applied as **soft control**—by continuously reshaping sampling probabilities or pacing—or as **hard control**—by filtering, skipping, or pruning low-utility items at each round.

For **soft control**, several works repeatedly recompute likelihood-style signals and use them to pace exposure to data as the model evolves, including perplexity-guided curricula and sampling schedules during training or RL-style post-training (hattami et al., 2025; Sun et al., 2025b). Loss-based online prioritization instead re-evaluates utility in terms of generalization improvement, preferentially training on points estimated to most reduce held-out loss (e.g., reducible holdout loss) as training progresses (Mindermann et al., 2022). Difficulty- and competence-aware schedulers update sampling scores based on the model's current proficiency, steering training toward appropriately challenging examples through easy-to-hard curricula or adaptive RL fine-tuning that maintains a learnable difficulty band (Huang & Xiong, 2024; Shi et al., 2025a; Koh et al., 2025).

For **hard control**, gradient- and alignment-based criteria derive utility from update-direction information, enabling greedy top-k selection of gradient-representative batches or filtering of reasoning data whose gradients are most aligned with desired updates (Wang et al., 2024d; Li et al., 2025d). Difficulty-based hard filtering applies threshold pruning that admits only examples within the model's current learning range—either progressively increasing the difficulty band across epochs via self-paced curricula combined with diversity-promoting subset selection (Yang et al., 2024b), or dynamically maintaining a balanced pass-rate window centered around intermediate difficulty at each training step (Bae et al., 2026). Uncertainty-based strategies similarly refresh scores from the current model state, prioritizing samples or tasks with high predictive instability (e.g., recency-based uncertainty histories or prompt-sensitivity under perturbations) to track what the model is least settled on at that moment (Song et al., 2020; Kung et al., 2023). To avoid redundancy and collapse during iterative selection, similarity- or diversity-aware methods impose explicit coverage constraints (e.g., self-evolving K-center sampling or balanced LSH-style bucketed selection), enforcing representative coverage of the data space (Wu et al., 2023; Phan et al., 2025). Finally, when external judges or reward mechanisms are available, iterative re-scoring can be instantiated as multi-round selection-and-tuning loops that repeatedly evaluate generated responses—via automated evaluators, surrogate judges, or reward functions—and refresh the training data or policy accordingly (Song et al., 2024; Wang et al., 2025b; Zhao et al., 2025d).

However, compared to the one-shot scoring, the trade-off is mainly computational and robustness-related. Re-scoring at high frequency can add overhead, especially when the scoring signal is expensive to compute.

Overall, compared to heuristic filters, they replace human-designed surface rules with model-derived signals, strengthening automation by deriving selection signals from the model itself. However, their core limitation is that adaptivity is "borrowed" from changing model states rather than "acquired" by learning a selector: the rule cannot optimize long-horizon objectives, calibrate itself against evaluator drift, or explicitly trade off exploration vs. exploitation in a principled way. This motivates the next regime, adaptive selection, where the selection policy itself is trained and updated. Notably, much of the recent literature that most strongly aligns with self-improvement (especially in post-training/alignment settings with dynamic feedback such as rewards, advantages, or verification outcomes) increasingly emphasizes learnable selection policies, making them a natural main focus for the following section.

### 3.3 Adaptive Selection

Adaptive selection represents the data selection regime most aligned with our vision of self-improvement. The core objective of data selection for self-improving systems is to autonomously determine what data is most beneficial given the current state of learning. A learnable selector operationalizes this objective by introducing a selection policy that decides which examples should be prioritized next.

This distinguishes it fundamentally from static filters and metric-guided scoring. In those earlier approaches, selection criteria are externally specified and remain fixed during training. Even if automated, they apply a predetermined rule to curate data. In contrast, a learnable selector makes selection itself a learnable component of the system. The decision rule is no longer handcrafted or frozen; it is optimized alongside the model. As a result, this regime achieves higher levels of both automation and continuity. Automation is strengthened because, once the pipeline is defined, selection decisions are driven by feedback signals without requiring ongoing human intervention. Continuity is strengthened because the selection policy evolves over time: as the model's capabilities, weaknesses, or objectives shift, the data acquisition strategy adapts accordingly.

We organize this section around three components: the **selection unit**, a design-level choice that fixes the granularity of selection before training begins; the **selection loop**, an iterative cycle of signal generation and policy update that drives the selector's evolution; and the **coupling** between the selector and the main model, which characterizes how tightly the two co-adapt during training. We conclude with a complementary perspective that distinguishes curriculum-driven from objective-driven selection.

#### 3.3.1 Selection Unit as a Design Choice

Before the selection loop begins, the system must fix the **selection unit**: the atomic granularity at which data can be sampled, weighted, or filtered. Unlike the iterative stages of the loop described below, the unit is a design-level decision made once and held constant throughout the entire selection process. This choice directly affects how precisely the selector can shape the training distribution: a finer unit enables more targeted selection, while a coarser unit limits the selector to broader adjustments.

Existing learnable selectors operate at two levels of granularity. At the **instance** level, the smallest selectable element is an individual training sample, such as a pretraining document, an instruction–response pair, or a prompt–generation context for preference annotation. This is the most common choice and provides the finest-grained control over the training distribution (Yu et al., 2024d; Fan et al., 2026; Bai et al., 2025; Lv et al., 2025; Chen et al., 2025i; Shen et al., 2025; Das et al., 2025). At the **group** level, the selector allocates training budget across higher-level collections—such as data sources, difficulty-stratified distributions, or semantically clustered categories—rather than individual examples (Wang et al., 2025j; Chen et al., 2025f; Do et al., 2025; Yu et al., 2025g; Jha et al., 2025; Pan et al., 2025).

We note that in metric-guided scoring (§3.2), selection units are predominantly individual instances or pre-defined domains. While the unit choice remains relevant, the scoring signal more directly characterizes how each method estimates data utility. Moreover, most metric-guided scoring methods operate at the instance level, with a smaller subset applying scores at the domain or group level. We therefore organize metric-guided scoring approaches primarily by their scoring signal, noting the applicable granularity where appropriate. In contrast, the unit choice in adaptive selection plays a more prominent architectural role—directly shaping what signals the selector computes and how it updates its policy—making it a meaningful standalone axis of the taxonomy.

#### 3.3.2 The Selection Loop

Given a fixed unit definition, the learnable selector operates through an iterative loop comprising two stages. In the first stage, **signal generation**, the system computes utility signals under the current model state to estimate the learning value of each unit. These signals are state-dependent: rather than being fixed properties of the data, they are computed relative to the current state of the model involved in selection, and thus shift as that model evolves during training. In the second stage, **selection update**, the selector transforms these signals into adjustments of the sampling policy—by reweighting, filtering, or reallocating

emphasis across units—thereby reshaping the effective training distribution. This reshaped distribution influences subsequent model updates and, in turn, alters the signals observed in the next iteration.

To introduce the concrete instantiations of this loop, we categorize learnable selectors by how they realize each stage.

**Signal Generation.** Given a unit choice, selection signals are categorized by the quantity used to score each unit. **Influence-based** signals estimate the marginal training contribution of individual samples or groups. MATES (Yu et al., 2024d) trains a small data influence model that is continuously fine-tuned to approximate oracle influence scores obtained by locally probing the pretraining model's performance on a reference task, thereby selecting instances most beneficial for the current pretraining progress. Group-MATES (Yu et al., 2025g) extends this principle to the group level, learning influence models that predict domain-level utility to guide pretraining data mixture decisions.

**Uncertainty-based** signals prioritize data that is expected to provide high informational gain. APO (Das et al., 2025) formulates RLHF alignment as a contextual preference bandit problem and iteratively selects the most uncertain prompt contexts for preference labeling, provably achieving near-optimal sample efficiency under the Bradley-Terry-Luce model (BTL).

**Advantage- or outcome-based** signals rely on performance-derived quantities observed during training. DUMP (Wang et al., 2025j) uses the magnitude of policy advantages as a proxy for distribution-level learnability, dynamically adjusting sampling probabilities across data distributions via a upper confidence bound (UCB)-based bandit during RL post-training. SEC (Chen et al., 2025f) similarly formulates curriculum selection as a non-stationary multi-armed bandit, using the absolute advantage from policy gradient methods as a reward signal and updating sampling probabilities with TD(0). SPaRFT (Do et al., 2025) first applies cluster-based data reduction to partition training data by semantics and difficulty, then treats each cluster as an arm in a multi-armed bandit whose reward reflects the model's current solve-rate performance on that cluster.

**Loss-based** signals directly use held-out validation loss or validation performance improvement as the scoring criterion. RAISE (Lv et al., 2025) models dynamic instruction selection as a sequential decision-making process, training a sample-wise scorer (acquisition function) via reinforcement learning to maximize downstream validation performance improvement across instruction fine-tuning steps. SEAL (Shen et al., 2025) learns a data ranker through bilevel optimization, where the upper-level objective evaluates safety-alignment quality on a held-out validation set while the lower level performs standard fine-tuning, thereby up-ranking safe and high-quality data. Grangier et al. (2024) cast pretraining data selection itself as a scalable online bilevel optimization problem, where the outer level adaptively reweights the pretraining distribution to minimize the loss on a small target-domain set while the inner level performs standard model training, allowing the training distribution to evolve jointly with the model throughout pretraining. ScaleBiO (Pan et al., 2025) introduces the scalable first-order bilevel optimization framework for LLM data reweighting, learning per-source sampling weights that minimize validation loss across multiple data sources. RL-Guided Selection (Jha et al., 2025) trains an RL-based selector that learns to allocate training budget across data groups by optimizing for downstream task performance.

Finally, **composite** signals combine multiple measurements into a unified scalar utility. ScalingRL (Chen et al., 2025i) integrates difficulty, reasoning complexity, and reward adaptability into a dynamic effectiveness score that is recomputed at each training epoch to guide within-difficulty-level sampling during RL fine-tuning. DATAMASK (Fan et al., 2026) jointly optimizes quality and diversity via a policy-gradient-based mask learning objective over the pretraining corpus. Multi-Actor (Bai et al., 2025) aggregates heterogeneous scoring signals from multiple specialized actor models, each capturing a different aspect of data utility, to produce a unified selection decision for pretraining.

**Selection Update.** Selectors convert these signals into changes in the effective training distribution through several recurring update mechanisms. **Supervised** updates fit auxiliary models that directly predict selection priorities from collected oracle labels (Yu et al., 2024d; 2025g). **RL-based** updates optimize a selection policy against reward or utility feedback using policy-gradient methods (Lv et al., 2025; Jha

Table 4: **Adaptive selection methods and their design dimensions.** This table summarizes representative methods by their selection unit, the signals used, the update mechanism and whether the selection process is coupled or decoupled with model training.

| Method | Unit | Signal | Update | Coupling |
|---|---|---|---|---|
| MATES (Yu et al., 2024d) | Instance | Influence | Supervised | Coupled |
| RAISE (Lv et al., 2025) | Instance | Loss | RL | Coupled |
| ScalingRL (Chen et al., 2025i) | Instance | Composite | Learned reweighting | Coupled |
| APO (Das et al., 2025) | Instance | Uncertainty | Bandit | Coupled |
| SEAL (Shen et al., 2025) | Instance | Loss | Bilevel | Decoupled |
| Multi-Actor (Bai et al., 2025) | Instance | Composite | Learned reweighting | Decoupled |
| DATAMASK (Fan et al., 2026) | Instance | Composite | RL | Decoupled |
| Grangier et al. (2024) | Group | Loss | Bilevel | Coupled |
| DUMP (Wang et al., 2025j) | Group | Advantage/Outcome | Bandit | Coupled |
| SEC (Chen et al., 2025f) | Group | Advantage/Outcome | Bandit | Coupled |
| SPaRFT (Do et al., 2025) | Group | Advantage/Outcome | Bandit | Coupled |
| Group-MATES (Yu et al., 2025g) | Group | Influence | Supervised | Coupled |
| RL-Guided Selection (Jha et al., 2025) | Group | Loss | RL | Decoupled |
| ScaleBiO (Pan et al., 2025) | Group | Loss | Bilevel | Decoupled |

et al., 2025; Fan et al., 2026). **Bandit-style** updates treat selection as a multi-armed bandit problem, adaptively allocating sampling probability via exploration–exploitation strategies such as UCB, Thompson Sampling, or Boltzmann exploration (Das et al., 2025; Wang et al., 2025j; Chen et al., 2025f; Do et al., 2025). **Learned reweighting** mechanisms optimize continuous per-unit weights or mixture coefficients to reshape the training distribution (Chen et al., 2025i; Bai et al., 2025). Finally, **Bilevel** approaches update selection parameters by differentiating through an inner training process with respect to an outer validation or alignment objective (Grangier et al., 2024; Shen et al., 2025; Pan et al., 2025).

Table 4 summarizes each reviewed method under the two-stage loop, categorizing it by unit granularity, signal type, and update mechanism. The table reveals that, while the space of signal–update combinations is diverse, the interaction between the selector and the main model—specifically, whether the two co-evolve within the same loop or operate independently—remains an important yet orthogonal design dimension. We examine this dimension next.

### 3.3.3 Coupling between Selector and Model

Beyond the signal and update axes captured in Table 4, an important additional dimension characterizes how the selector and the main model interact during training: the degree of **coupling** between them.

In a **coupled** setting, the main model is updated on the selected data, and its changed state feeds back into the next round of signal generation, creating an interleaved loop in which selection and optimization co-adapt. The majority of surveyed methods adopt this design (Yu et al., 2024d; Grangier et al., 2024; Yu et al., 2025g; Lv et al., 2025; Chen et al., 2025i; Das et al., 2025; Wang et al., 2025j; Chen et al., 2025f; Do et al., 2025).

Coupling arises naturally when the selector's signal is derived from the main model being optimized—for instance, policy advantages in RL-based curricula (Wang et al., 2025j; Chen et al., 2025f; Do et al., 2025), data influence probed from evolving pretraining checkpoints (Yu et al., 2024d; 2025g), or validation performance tracked across fine-tuning steps (Lv et al., 2025; Chen et al., 2025i).

In a **decoupled** setting, the selector is trained independently—often via bilevel optimization on a held-out validation objective (Shen et al., 2025; Pan et al., 2025), through a separate policy-gradient or scoring phase (Fan et al., 2026; Bai et al., 2025), or using a proxy model to provide reward signals (Jha et al., 2025)—and

the resulting selection is applied before the main model begins training. Decoupled designs reduce system complexity, but forfeit the ability to adapt selection as the model's needs shift during training.

This distinction parallels the contrast between one-shot and iterative re-scoring in the metric-guided scoring regime: just as iterative re-scoring refreshes utility estimates during training while one-shot scoring fixes them beforehand, coupled selectors continuously re-adapt their policies while decoupled selectors commit to a fixed selection before training begins.

Notably, all surveyed methods in this regime rely on a **separate** selector module—whether a small influence model (Yu et al., 2024d), a trainable MLP scorer (Lv et al., 2025), a bandit policy (Wang et al., 2025j; Chen et al., 2025f), or a bilevel-optimized ranker (Shen et al., 2025)—while a distinct main model is optimized on the selected data. No existing method in this regime has the main model curate its own training data without an external selector. This gap points to an underexplored direction for future work: **self-directed selection**, in which a single model simultaneously acts as both selector and learner, representing the strongest form of autonomy in self-improving systems.

### 3.3.4 Curriculum-Driven vs. Objective-Driven Selection

From another perspective, adaptive selection mechanisms can also be categorized based on what fundamentally drives the selection criterion. Although these methods generally rely on signals to update the selector over time, the underlying objectives that those signals serve are not the same. Under this view, existing methods largely fall into two paradigms: **Curriculum-Driven** and **Objective-Driven** selection.

Curriculum-driven selection aims to select data that is most appropriate for the model's current learning stage. The core idea is to adapt the training distribution to the model's evolving competence. Here, "appropriate" does not simply mean high-quality or high-reward; rather, it refers to examples whose difficulty or informational value matches the model's current capacity—neither too trivial to provide new learning signal, nor too difficult to be learnable (Chen et al., 2025i; Wang et al., 2025j; Chen et al., 2025f; Do et al., 2025).

On the other hand, objective-driven selection directly prioritizes data based on its estimated contribution to a predefined optimization objective. The target objective may vary depending on the setting—for example, improving downstream task performance, enhancing reasoning accuracy, aligning the model with human preferences, or enforcing safety constraints (Lv et al., 2025; Jha et al., 2025; Shen et al., 2025; Pan et al., 2025; Fan et al., 2026).

In summary, adaptive selection advances data selection from a fixed preprocessing step to a learnable, evolving component of the training pipeline. The methods surveyed in this section differ along several design axes—unit granularity, signal type, update mechanism, and the degree of coupling with the main model—yet they share a common goal: the selection policy should be optimized to respond to the model's evolving competence, rather than fixed by hand. This responsiveness may be driven by curriculum considerations that match data difficulty to the model's current capacity, or by objective-driven criteria that maximize a downstream target. In either case, the aim is a closed feedback loop between what the model learns and what it is trained on next—one that coupled methods realize directly and decoupled methods approximate through offline optimization. Realizing this loop in practice, however, involves trade-offs in computational cost, training stability, and system complexity that must be weighed against the gains in adaptivity—a theme we return to in the broader discussion of §3.4.

### 3.4 Discussion

### 3.4.1 Trade-offs Across Regimes

Data selection is a control layer in self-improvement pipelines: by deciding which samples enter optimization (and with what weight) at each step, it effectively decides what the model trains next. Increased automation and continuity come with clear trade-offs. Static filtering, the historical baseline, offers simplicity, stability, and corpus-scale throughput, but its rules remain fixed and cannot track a model's changing competence. Building on it, two model-driven regimes go further. Metric-guided scoring replaces handcrafted heuristics

Table 5: **Representative data selection methods across different training stages.** This table summarizes representative methods applied during pre-training and post-training, highlighting how data selection strategies are used at different stages of model development.

| Training Phase | Methods |
| --- | --- |
| Pre-training | C4 (Raffel et al., 2020), CCNet (Wenzek et al., 2020), The Pile (Gao et al., 2020), GPT-3 (Brown et al., 2020), GLaM (Du et al., 2022), RHO-LOSS (Mindermann et al., 2022), PaLM (Chowdhery et al., 2023), DSIR (Xie et al., 2023b), DoReMi (Xie et al., 2023a), GREATS (Wang et al., 2024d), DATAMASK (Fan et al., 2026) |
| Post-training | Recency Bias (Song et al., 2020), DEFT (Ivison et al., 2023), NUGGETS (Li et al., 2024k), S2L (Yang et al., 2024c), LESS (Xia et al., 2024), Li et al. (2024e), SelectIT (Liu et al., 2024d), AlpaGasus (Chen et al., 2024d), DEITA (Liu et al., 2024g), GREATS (Wang et al., 2024d), IT2ACL (Huang & Xiong, 2024), P3 (Yang et al., 2024b), SST (hattami et al., 2025), PREPO (Sun et al., 2025b), AdaRFT (Shi et al., 2025a), AdaSTaR (Koh et al., 2025), LearnAlign (Li et al., 2025d), DUMP (Wang et al., 2025j), SEC (Chen et al., 2025f), RAISE (Lv et al., 2025), SPaRFT (Do et al., 2025), ScalingRL (Chen et al., 2025i), SEAL (Shen et al., 2025) |

with model- or evaluator-derived signals: one-shot scoring is cheap and stable yet may become stale as objectives shift, whereas iterative re-scoring updates the selection signal during training to better track non-stationary utility at additional computational and robustness cost. Adaptive selection goes further by optimizing the selection policy itself, enabling budget-aware curricula and longer-horizon control, but it introduces sensitivity to noisy/drifting feedback, delayed credit assignment, and extra system overhead.

### 3.4.2 Pre-training vs. Post-training

Viewing data selection through the training pipeline (Table 5) reveals that these trade-offs manifest differently across training phases, forming a clear phase-dependent pattern. In **pre-training**, the dominant constraints are scale, throughput, and stability: datasets are massive, objectives are primarily next-token prediction, and selection must be cheap and robust. Consequently, static filtering and one-shot scoring-based mixture shaping (e.g., domain reweighting or pruning) are especially attractive, while online selection is used more selectively due to overhead. In **post-training**, which encompasses fine-tuning and alignment, the setting changes: training budgets are smaller, objectives are more targeted, and automated evaluators such as reward models, verifiers, and critics, alongside preference signals, provide richer feedback. This makes iterative re-scoring and adaptive selection substantially more practical and better aligned with self-improvement continuity, since the data stream can be adapted as the model's behavior changes under closed-loop feedback.

This phase-dependent contrast suggests that no single selection mechanism is sufficient across all stages of training. Instead, different mechanisms should play different roles depending on available feedback and computational constraints. We therefore propose self-improving data selection as a coordinated combination of strategies applied across successive training phases: stable, inexpensive mechanisms establish a broad foundation at scale, while adaptive methods operate closer to the optimization loop as richer feedback becomes available.

Concretely, a self-improvement pipeline may first enforce hard constraints (e.g., safety, licensing or provenance, toxicity filtering, deduplication), then adjust coarse mixture proportions across domains or skills, and finally apply policy-learned selection to allocate limited post-training budget toward samples that are simultaneously learnable, high-impact, and aligned with target behaviors. By separating stable, large-scale screening from fine-grained adaptive refinement, this design makes adaptivity scalable: expensive feedback and policy learning are applied only where they add the most value, while earlier stages remain robust and computationally efficient. Under this view, continuity does not come from switching to entirely new selec-

tion strategies at each training stage, but from gradually adapting the data distribution as the model itself improves.

### 3.4.3 Interaction with Data Acquisition

Within the self-improvement system, selection sits immediately downstream of data acquisition: acquisition expands the pool of experiences, and selection decides which of them are worth learning from. The properties of the acquired data therefore directly shape the work of selection. Web-scale corpora obtained through static curation force selection to balance quality against throughput; self-synthesized solutions may be fluent yet wrong, so they must be verified before entering training; and as the model grows more capable, samples that were once informative become trivial, requiring the selection criteria to evolve together with the incoming data and the model itself. In a closed loop, acquisition widens the sources of experience while selection guards the quality of what is trained on, and together the two stages set the data foundation of each improvement iteration.

## 4 Model Optimization for Self-Improvement

### 4.1 Overview

Model optimization is the engine that converts experiences in data into actual model capability gains. While the data acquisition and selection stages focus on preparing the learning curriculum, optimization is where the underlying model goes through parameter updates to internalize these signals, evolving into a more advanced version without constant human intervention. Model optimization takes three broad paradigms, which structure this section, as shown in Figure 6:

- **Direct Optimization** fits the model with a single offline update on a corpus already assembled by the data acquisition and selection stages.

- **Self-Generated Optimization** (SGO), the primary focus of this section, differs from direct optimization in that the corpus is no longer fixed in advance: the model repeatedly generates the experience it trains on, receives a reward for it, and updates on the result, so that the training distribution is policy-induced and non-stationary. In this sense SGO is essentially a specialized form of reinforcement learning (Sutton & Barto, 2018), where the model learns from experience it generates itself, and we cast it as a loop of three tightly coupled stages, **generation**, **reward**, and **optimization**.

- **Self-Evolving Optimization** goes beyond updating the model's parameters: the very algorithm that performs the update is itself improved, together with the architecture and agent-level scaffolding, thereby extending the unit of improvement from a single model's weights to the whole agentic system.

In the following, §4.2 covers direct optimization, §4.3 presents self-generated optimization, and §4.4 develops self-evolving optimization, followed by a discussion of all model optimization methods in §4.5.

### 4.2 Direct Optimization

The simplest form of model optimization is **direct optimization**, in which the model is updated offline on a fixed dataset assembled by the preceding acquisition and selection stages. The training corpus is frozen before optimization begins, and no candidate is resampled from the updated model or rescored during training. The most common instantiation is a one-shot supervised fine-tuning (SFT) objective fit to input–target pairs. The training distribution stays stationary, and the attainable improvement is bounded by the quality and coverage of the assembled data. In the self-improvement setting, the training targets are not hand-annotated but produced and curated by the model itself in the upstream stages, which the model then internalizes through a single offline update.

A large body of work follows this pattern for instruction alignment. Self-Instruct (Wang et al., 2023c) starts from a small seed set, has the model generate instruction–response pairs, filters them heuristically, and

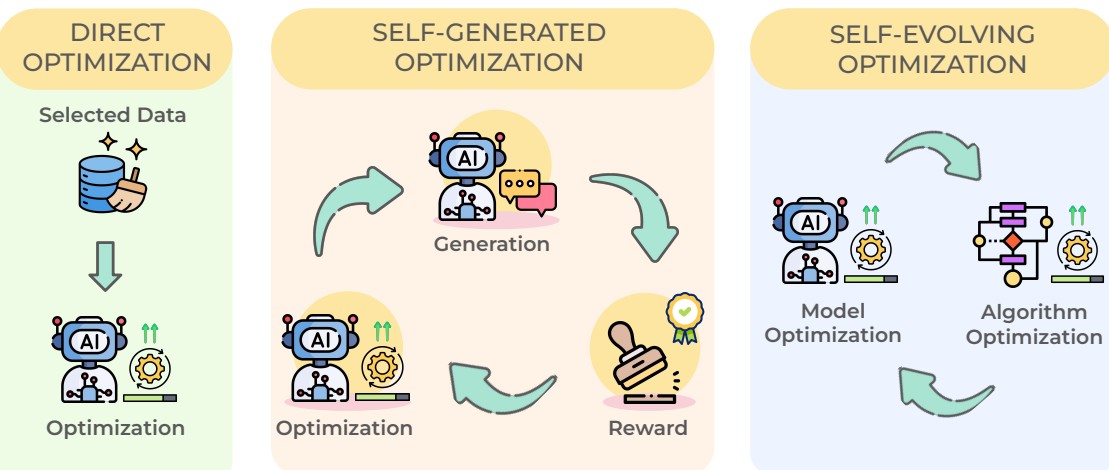

Figure 6: **Overview of model optimization in the self-improvement system.** We organize model optimization into three paradigms: **(i) Direct Optimization**, a single offline update on a fixed corpus; **(ii) Self-Generated Optimization (SGO)**, a closed loop of *generation*, *reward*, and *optimization* where the model learns from self-produced experience; and **(iii) Self-Evolving Optimization**, which improves the optimization process itself, from the update rule to the agent-level scaffolding.

fine-tunes on the retained pool. WizardLM (Xu et al., 2024a) and CodecLM (Wang et al., 2024i) keep the same one-shot SFT step while making the data-construction stage more elaborate, evolving seed instructions toward higher complexity or decoding metadata into tailored synthetic examples. Instruction backtranslation (Li et al., 2024g) instead has the model self-augment and self-curate targets from unlabeled web text. More recent methods push self-synthesis further: Magpie (Xu et al., 2025d) elicits large-scale alignment data directly from an aligned model by prompting it with only the pre-query template, and shows that plain SFT on this self-generated corpus can match models trained with far larger human-curated pipelines, with LongMagpie (Gao et al., 2025a) extending the idea to long-context instructions.

The same recipe applies to agentic capabilities, where the assembled corpus consists of interaction trajectories rather than static instructions. Learn-by-Interact (Su et al., 2025a) synthesizes agent trajectories through interaction with realistic environments and, via backward construction, turns them into instruction–trajectory data for a single SFT pass, while trajectory-tuning approaches such as AgentTuning (Zeng et al., 2024) and FireAct (Zeng et al., 2024) fine-tune the model on agent trajectories distilled from a stronger teacher to internalize agentic behavior. Across all these cases the optimization step remains a stable and inexpensive single offline update, and the sophistication of the method lies in how the data is produced rather than in how the parameters are updated. Direct optimization is therefore the appropriate choice when high-quality experience can be assembled ahead of time, offering simplicity and stability at the cost of an improvement ceiling fixed by the given data.

### 4.3 Self-Generated Optimization

In contrast to direct optimization, Self-Generated Optimization (SGO) in this section forms a closed loop in which the model repeatedly generates the data it trains on, scores that data, and updates on the resulting signal, with the updated model then driving the next round of generation. This distinction is about the loop rather than the presence of generation or filtering: a method such as Self-Instruct (Wang et al., 2023c) also generates and filters candidates, but it does so once from a fixed model before a single supervised update, which places it under direct optimization; its generation and selection belong to the acquisition and selection stages rather than to the optimization loop. Here, by contrast, generation and reward are internal to optimization and are re-invoked on the continually updated policy, so the training distribution is policy-induced and non-stationary.

---

**Algorithm 1** **The Self-Generated Optimization (SGO) Loop**

---

**Input:** initial policy $\pi_{\theta_0}$; dataset $\mathcal{D}$; number of cycles $T$

1: **for** $t = 0, 1, \ldots, T - 1$ **do**

2:     // **Generation:** the policy turns the available seed data into candidate trajectories

3:     $\mathcal{B}_t^{\mathrm{gen}} = \mathcal{G}\big(\pi_{\theta_t}, \mathcal{D}\big), \quad$ e.g.,

      $\mathcal{D} = \{x\}: \ \mathcal{B}_t^{\mathrm{gen}} = \big\{(x, y^{(i)}) : y^{(i)} \sim \pi_{\theta_t}(\cdot \mid x)\big\}$            STaR (Zelikman et al., 2022)

      $\mathcal{D} = \{(x, y)\}: \ \mathcal{B}_t^{\mathrm{gen}} = \big\{(x, y') : y' \sim \pi_{\theta_t}(\cdot \mid x, y)\big\}$           SELF (Lu et al., 2024c)

      $\mathcal{D} = \emptyset: \ \mathcal{B}_t^{\mathrm{gen}} = \big\{(x, y) : x \sim \pi_{\theta_t}(\cdot), \ y \sim \pi_{\theta_t}(\cdot \mid x)\big\}$        R-Zero (Huang et al., 2026a)

4:     // **Reward:** score candidates in the format the downstream optimizer consumes

5:     $\mathcal{B}_t = \mathcal{R}\big(\mathcal{B}_t^{\mathrm{gen}}\big), \quad$ e.g.,

      filter: $\ \mathcal{B}_t = \big\{(x, y) \in \mathcal{B}_t^{\mathrm{gen}} : r(x, y) = 1\big\}$ via verification      Self-Challenging (Zhou et al., 2025)

      scalar: $\ \mathcal{B}_t = \big\{(x, y, r) : r \in \mathbb{R}\big\}$ via a model-based judge or execution     SIRLC (Pang et al., 2024)

      ranking: $\ \mathcal{B}_t = \big\{(x, y^+, y^-) : y^+ \succ y^-\big\}$ via self- or heuristic ranking         Yuan et al. (2024b)

6:     // **Optimization:** update the policy with the objective coupled to the reward format

7:     $\theta_{t+1} = \mathcal{O}\big(\theta_t, \mathcal{B}_t\big), \quad$ e.g.,

      likelihood (SFT): $\ \theta_{t+1} = \arg\max_\theta \sum_{(x,y) \in \mathcal{B}_t} \log \pi_\theta(y \mid x)$       LMSI (Huang et al., 2023a)

      scalar (PPO, GRPO): $\ \theta_{t+1} = \arg\max_\theta \ \mathbb{E}_{y \sim \pi_\theta(\cdot \mid x)}\big[r\big]$        SCoRe (Kumar et al., 2025)

      preference (DPO): $\ \theta_{t+1} = \arg\max_\theta \sum \log \sigma\big(\beta \log \frac{\pi_\theta(y^+ \mid x)\, \pi_{\theta_t}(y^- \mid x)}{\pi_{\theta_t}(y^+ \mid x)\, \pi_\theta(y^- \mid x)}\big)$     SynPO (Dong et al., 2025)

8: **end for**

**Output:** improved policy $\pi_{\theta_T}$

---

Because the model generates its own data rather than receiving annotated targets, the outputs it produces carry no labels: a self-generated response cannot be trained on until something decides whether it is good. This is why every SGO method needs a reward stage, and why the loop is fundamentally a reinforcement-learning process. The model first rolls out its current policy to produce candidates, then judges or scores these unlabeled candidates to obtain a training signal, and finally updates its parameters on that signal. We cast this loop into three tightly coupled stages, **Generation**, **Reward**, and **Optimization**:

- **Generation:** Starting from the current policy, the model acts as its own experience generator, producing candidate outputs, reasoning chains, or action trajectories for a given set of tasks.

- **Reward:** These outputs are then evaluated automatically to produce a feedback signal. This feedback is distilled into specific formats based on the intended learning objective: binary filters to identify high-quality samples, scalar scores to quantify performance, or comparative rankings to establish preferences. Whether derived from internal model logic or external environment feedback, these signals provide the "gradient of quality" necessary for improvement.

- **Optimization:** In the final stage, the policy parameters are updated to internalize these signals, evolving into the refined model. The update method follows the reward format: SFT on filtered high-quality samples, preference-based algorithms such as DPO (Rafailov et al., 2023) on comparative rankings, online RL such as PPO (Schulman et al., 2017) and GRPO (Shao et al., 2024) on scalar rewards, or combinations of these. This step executes the actual transformation of the model's capabilities, completing one cycle of the improvement.

Mathematically, Algorithm 1 formalizes this loop as three operators applied in sequence, a generation operator $\mathcal{G}$, a reward operator $\mathcal{R}$, and an optimization operator $\mathcal{O}$. Each operator takes different forms depending on the setting: generation varies with what seed data is available, whether only prompts, prompts paired with answers, or nothing at all; reward varies with the format of the signal it produces, whether a filter, a scalar, or a ranking; and optimization varies with the objective coupled to that format. We detail the categorization within each stage in the following three subsections, on generation (§4.3.1), reward (§4.3.2), and optimization (§4.3.3).

### 4.3.1 Generation

Generation refers to the process by which a model produces candidate outputs for a given task. We categorize existing generation strategies into three classes: **Self-Exploratory Generation**, **Refined Generation**, and **Interactive Generation**. Self-exploratory generation operates directly on the model's current policy, either by producing a single response that represents the policy's behavior or by sampling a batch of responses to explore the output space via search. Refined generation starts from an initial response and iteratively improves it, aiming to obtain high-quality data for training, construct preference pairs, or explicitly learn the refinement process itself. Interactive generation extends the process beyond the isolated model, relying on collaborative consensus, adversarial dynamics with an opponent, or interaction with external tools to drive generation.

**Self-Exploratory Generation.** The fundamental strategy in self-exploratory generation is producing outputs directly from the model's current policy. In its simplest form, the model generates a single response or reasoning path to be assessed; methods like STaR (Zelikman et al., 2022), SIRLC (Pang et al., 2024), and RLSR (Simonds et al., 2025) utilize this approach by generating a solitary rationale or answer per input. SimRAG (Xu et al., 2025a) extends this by generating an answer followed by a retrieved question for consistency. This policy-driven generation also underpins self-play approaches, where the generated output represents a move by the current policy or an opponent to construct training signals. SPIN (Chen et al., 2024i) generates responses to contrast against ground truth in a zero-sum game, while SPPO (Wu et al., 2025g), SeRL (Fang et al., 2025c), RSPO (Tang et al., 2025b), and SOL-VER (Lin et al., 2025c) generate outputs from the current policy or a reference model to serve as players in a preference optimization or verification game.

To explore the output space more robustly, many methods sample multiple diverse candidates to identify consistent answers or construct contrastive pairs. LMSI (Huang et al., 2023a), CRESCENT (Sun et al., 2025c), and ReGenesis (PENG et al., 2025) sample diverse paths and apply majority voting to select high-quality outputs. Jiang et al. (2025a) similarly employ importance weighting on multiple samples to filter training data. Jiang et al. (2025b) and RESTRAIN (YU et al., 2026) further analyze these batches for semantic clusters or voting signals to reduce noise. Additionally, ScPO (Prasad et al., 2025), DNPO (Yang et al., 2026), Goedel-Prover (Lin et al., 2025b), and V-STaR (Hosseini et al., 2024) generate multiple candidates to be filtered by verifiers or ranked by the model itself to form valid training pairs. This strategy is also central to Yuan et al. (2024b), Wu et al. (2025c), and Liu et al. (2025d), which rely on sampling candidates for self-evaluation or verification.

In addition, to structure the exploration more effectively, some methods utilize tree search algorithms. ReST-MCTS* (Zhang et al., 2024a) and SPHERE (Singh et al., 2025) employ Monte Carlo Tree Search to actively navigate the generation space, pruning low-quality paths and selecting the most promising trajectories for learning.

**Refined Generation.** Refined generation employs iterative improvements on initial responses to obtain higher-quality trajectories that serve as positive training data. In this setting, the initial generation is merely a stepping stone; the model employs iterative self-correction or feedback loops to ensure the final output is correct before using it for supervised fine-tuning. SELF (Lu et al., 2024c) and STaSC (Moskvoretskii et al., 2025) implement a "generate-critique-revise" pipeline, retaining the refined trajectory only if it passes specific quality checks. RISE (Qu et al., 2024) and TRIPOST (Yu et al., 2024c) enforce a rigorous feedback loop, where the model acts on internal signals or external teacher feedback to "try again" iteratively until a correct reasoning path is achieved. ReGenesis (PENG et al., 2025) also falls into this category by explicitly refining reasoning chains based on ground truth signals to construct a high-quality dataset for fine-tuning. Similarly, Bensal et al. (2025) treat the refinement as a reflection step triggered by verification signals to produce a successful final trajectory.

Beyond simply gathering correct answers, many approaches use refinement to construct preference pairs by contrasting the initial weak response with the refined strong response. This allows the model to learn the improvement signal via algorithms like DPO. SynPO (Dong et al., 2025) explicitly constructs optimization pairs by utilizing the initial generation as the rejected sample and the refined output as the chosen sample.

Ji et al. (2025) leverage external context to create this contrast, treating "closed-book" generations as weaker samples and "open-book" generations as the refined targets. SPHERE (Singh et al., 2025) and Xiong et al. (2025) further utilize the success or failure of the self-correction process to label trajectories, creating pairs of successful refinements versus failed attempts to guide preference optimization.

Finally, some methods treat the refinement step as a direct supervision signal to explicitly learn the refinement process itself. In this paradigm, the initial incorrect response serves as the input context, and the refined response serves as the target output, effectively training the model to map errors to corrections. STaPLe (Ramji et al., 2025) operationalizes this by generating a latent "principle" derived from the ground truth to bridge the gap between the initial error and the correct answer. SCoRe (Kumar et al., 2025) trains the model to maximize the quality of the second-turn response given the first-turn attempt, reinforcing the capability to self-correct without external supervision. Similarly, PAG (Jiang et al., 2025e) treats the policy as a generative verifier, learning to improve reasoning chains over multiple turns of self-correction.

**Interactive Generation.** Interactive generation often relies on collaborative consensus, where multiple agents cooperate to synthesize a high-quality output. In this setting, agents effectively act as a team, engaging in debate or sequential improvement. DTE (Srivastava et al., 2025) and Subramaniam et al. (2025) utilize a debate framework where distinct agents critique each other's reasoning chains, aggregating their outputs to filter for consistency and reduce hallucinations. SiriuS (Zhao et al., 2025c) adopts a sequential collaborative model, where agents hand off partial solutions or cooperate in a multi-turn dialogue to solve complex tasks that a single agent could not manage alone.

Other strategies employ adversarial generation, where the generation process is structurally dependent on an opponent. Here, the primary model's generation is often a response to a dynamic challenge created by an adversarial peer. R-Zero (Huang et al., 2026a), Dr. Zero (Yue et al., 2026), and Absolute Zero (Zhao et al., 2025a) rely on this "Proposer-Solver" dynamic, where one agent must generate a novel problem or theorem before the solver agent can generate a solution trajectory. Similarly, Zhou et al. (2025) involve a Challenger agent that actively probes the Executor to induce failure, forcing the Executor to generate more robust responses. SSR (Wei et al., 2025b) and SPICE (Liu et al., 2025a) also follow this paradigm, where the generation of code fixes or reasoning chains is directly conditioned on the specific bugs or difficult contexts synthesized by the adversarial role.

Lastly, generation can be tool-augmented, relying on external tools, execution environments, or retrieval systems to proceed. Unlike post-hoc verification, these methods require the tool's feedback during the generation loop to construct the final trajectory. ReVeal (Jin et al., 2026) integrates a code execution environment directly into the generation process, where agents generate code, execute it to observe the state, and use that observation to guide subsequent generation steps. CURE (Wang et al., 2025h) co-evolves a coder and a unit tester, where the generation of the solution is strictly coupled with the generation and execution of test cases, ensuring the output is grounded in verifiable execution feedback.

### 4.3.2 Reward

Given the generated outputs, the reward stage provides signals that assess their quality and guide subsequent training updates. We categorize reward into three major types: **Heuristic Reward**, **Model-Based Reward**, and **Verification Reward**. Heuristic reward evaluates quality without a trained reward model, relying on statistical consistency among samples, specifically designed scoring functions, or pre-defined assumptions about the relative quality of refined outputs. Model-based reward leverages the semantic capabilities of LLMs, either by having the model self-evaluate its own generations or by utilizing external peer models and judges to provide feedback. Verification reward offers the most definitive signals, derived from strict ground truth matching or execution feedback from formal verifiers and code compilers.

**Heuristic Reward.** Heuristic reward relies on statistical signals or rule-based metrics to approximate quality. A dominant approach is consistency-based evaluation, which operates on the premise that consensus among multiple generations indicates correctness. LMSI (Huang et al., 2023a) and IWSI (Jiang et al., 2025a) sample multiple reasoning paths and utilize majority voting to identify the most consistent answer, treating it as a positive training signal. Similarly, ReGenesis (PENG et al., 2025) and CRESCENT (Sun et al., 2025c)

Table 6: **Model optimization methods for self-improvement under the SGO framework.** We list methods by their generation strategy, reward type, and the specific optimization method used.

| Method | Generation | Reward | Optimization |
|---|---|---|---|
| STaR (Zelikman et al., 2022) | Self-Exploratory | Verification | SFT |
| LMSI (Huang et al., 2023a) | Self-Exploratory | Heuristic | SFT |
| TRIPOST (Yu et al., 2024c) | Refined | Model-Based | SFT |
| RISE (Qu et al., 2024) | Refined | M & V | SFT |
| SELF (Lu et al., 2024c) | Refined | M & V | SFT |
| ReST-MCTS* (Zhang et al., 2024a) | Self-Exploratory | Model-Based | SFT |
| V-STaR (Hosseini et al., 2024) | Self-Exploratory | Verification | SFT & DPO |
| IWSI (Jiang et al., 2025a) | Self-Exploratory | H & M | SFT |
| STaPLe (Ramji et al., 2025) | Refined | Heuristic | SFT |
| CRESCENT (Sun et al., 2025c) | Self-Exploratory | Heuristic | SFT |
| ReGenesis (PENG et al., 2025) | SE & R | H & V | SFT |
| Subramaniam et al. (2025) | Interactive | Heuristic | SFT |
| STaSC (Moskvoretskii et al., 2025) | Refined | Verification | SFT |
| SimRAG (Xu et al., 2025a) | Self-Exploratory | Heuristic | SFT |
| SiriuS (Zhao et al., 2025c) | Interactive | Verification | SFT |
| Semantic Voting (Jiang et al., 2025b) | Self-Exploratory | H & M | SFT |
| Self-Challenging (Zhou et al., 2025) | Interactive | Verification | SFT |
| SIRLC (Pang et al., 2024) | Self-Exploratory | Model-Based | PPO |
| Yuan et al. (2024b) | Self-Exploratory | Model-Based | DPO |
| SPIN (Chen et al., 2024i) | Self-Exploratory | Heuristic | SPIN |
| SCoRe (Kumar et al., 2025) | Refined | Verification | Multi-turn RL |
| SPPO (Wu et al., 2025g) | Self-Exploratory | Model-Based | SPPO |
| SynPO (Dong et al., 2025) | Refined | Heuristic | DPO |
| Bensal et al. (2025) | Refined | Verification | GRPO |
| SPHERE (Singh et al., 2025) | SE & R | M & V | DPO |
| RLSR (Simonds et al., 2025) | Self-Exploratory | Model-Based | GRPO |
| Ji et al. (2025) | Refined | Heuristic | DPO |
| SeRL (Fang et al., 2025c) | Self-Exploratory | Verification | GRPO |
| TBV (Liu et al., 2025d) | Self-Exploratory | Verification | PPO |
| PAG (Jiang et al., 2025e) | Refined | Verification | Multi-turn RL |
| CURE (Wang et al., 2025h) | Interactive | H & V | PPO |
| SSR (Wei et al., 2025b) | Interactive | H & V | SWE-RL |
| RSPO (Tang et al., 2025b) | Self-Exploratory | Model-Based | RSPO |
| Absolute Zero (Zhao et al., 2025a) | Interactive | H & V | TRR++ |
| SPICE (Liu et al., 2025a) | Interactive | H & V | DrGRPO |
| MAE (Chen et al., 2025h) | Interactive | Model-Based | GRPO |
| Wu et al. (2025c) | Self-Exploratory | Model-Based | DPO |
| DNPO (Yang et al., 2026) | Self-Exploratory | Model-Based | DNPO |
| ReVeal (Jin et al., 2026) | Interactive | H & V | TAPO |
| RESTRAIN (YU et al., 2026) | Self-Exploratory | Heuristic | GRPO |
| R-Zero (Huang et al., 2026a) | Interactive | H & V | GRPO |
| Dr. Zero (Yue et al., 2026) | Interactive | Verification | GRPO & HRPO |
| ScPO (Prasad et al., 2025) | Self-Exploratory | Heuristic | ScPO |

*Continued on next page*

*Table 6 continued from previous page*

| Method | Generation | Reward | Optimization |
|---|---|---|---|
| Xiong et al. (2025) | Refined | Verification | SFT & RL |
| DTE (Srivastava et al., 2025) | Interactive | Heuristic | SFT & GRPO |
| Goedel-Prover (Lin et al., 2025b) | Self-Exploratory | Verification | SFT & DPO |
| SOL-VER (Lin et al., 2025c) | Self-Exploratory | M & V | SFT & DPO |

The symbol "&" indicates that multiple techniques are used within the same stage, with capital letters denoting technique abbreviations (e.g., SE&R for self-exploratory and refined generation).

leverage voting consensus across diverse reasoning chains or self-generated questions to filter high-quality trajectories. In multi-agent settings, DTE (Srivastava et al., 2025) and Subramaniam et al. (2025) employ consistency scores among debating agents to reduce hallucinations. ScPO (Prasad et al., 2025) also uses consistency measures to distinguish between high- and low-quality outputs for preference optimization, while Semantic Voting (Jiang et al., 2025b) enhances this by clustering semantically similar responses to estimate correctness more robustly.

Other methods employ specifically designed heuristic functions where specific scoring rules are engineered to evaluate quality. SPIN (Chen et al., 2024i) derives an implicit reward signal by comparing the model's likelihood on generated data against reference data. RESTRAIN (YU et al., 2026) designs a penalization term to down-weight high-confidence but incorrect samples. In retrieval contexts, SimRAG (Xu et al., 2025a) uses a heuristic based on whether the generated question can successfully retrieve the document containing the answer. STaPLe (Ramji et al., 2025) similarly employs a similarity function to select the best generated principle-response pair. Furthermore, in adversarial and code generation frameworks, heuristic rewards are often used to prioritize difficulty or promise. The Proposer agents in Absolute Zero (Zhao et al., 2025a), SPICE (Liu et al., 2025a), R-Zero (Huang et al., 2026a), and Dr. Zero (Yue et al., 2026) receive heuristic rewards based on problem difficulty to drive exploration. Similarly, ReVeal (Jin et al., 2026), CURE (Wang et al., 2025h), and SSR (Wei et al., 2025b) utilize static analysis scores to filter code candidates before execution.

Finally, some rewards are based on pre-defined assumptions, establishing simple rules to determine relative quality. SynPO (Dong et al., 2025) assumes that the output from a refinement step is inherently superior to the initial generation, automatically forming a chosen-rejected pair. Ji et al. (2025) apply a similar logic in their framework by assuming that "open-book" generations (with access to external documents) are superior to "closed-book" generations, using this assumption to label data for preference optimization without manual annotation.

**Model-Based Reward.** Model-based reward leverages the semantic capabilities of Large Language Models to assign scores, rank candidates, or provide feedback. In self-evaluation strategies, the generating model itself acts as the judge. A direct implementation involves prompting the model to assign scalar quality scores to its own outputs, as seen in Self-Rewarding LMs (Yuan et al., 2024b) and SIRLC (Pang et al., 2024). Beyond scalar scoring, the model can generate self-critiques or preference rankings to guide optimization. SELF (Lu et al., 2024c) utilizes the model's own capability to critique and revise responses, using these self-generated signals to filter trajectories. RISE (Qu et al., 2024) also relies on the model's internal introspection capabilities to determine when to halt the refinement loop. Similarly, Meta-rewarding (Wu et al., 2025c) and SPPO (Wu et al., 2025g) prompt the model to rank pairs of responses, deriving a probability-based reward signal directly from the current policy's preference ordering. IWSI (Jiang et al., 2025a) and Semantic Voting (Jiang et al., 2025b) further leverage the model's likelihood estimates or embedding similarities to weight or cluster samples effectively.

Alternatively, reward can be derived from external or specialized models, such as a stronger teacher, a peer agent, or a specifically trained reward model. ReST-MCTS* (Zhang et al., 2024a) exemplifies this by training a dedicated Process Reward Model (PRM) to evaluate intermediate reasoning steps, providing a dense signal distinct from the generator. DNPO (Yang et al., 2026) also falls into this category by utilizing a reference model or dynamic noise distribution to construct the preference signal. TRIPOST (Yu et al., 2024c) employs a larger teacher model to critique and score the outputs of a smaller student model. In multi-agent and

adversarial settings, the reward is strictly determined by the judgment of a third-party agent or the policy of an opponent. MAE (Chen et al., 2025h) utilizes a distinct "Judge" agent to assess evolved responses, while RSPO (Tang et al., 2025b) derives rewards from the competitive outcome against a separate opponent policy. RLSR (Simonds et al., 2025), SPHERE (Singh et al., 2025), and SOL-VER (Lin et al., 2025c) further adopt this approach by leveraging external verifiers or reward models to score generated trajectories.

**Verification Reward.** Verification reward provides definitive quality signals by checking outputs against established truths or executable environments. The most direct form utilizes ground truth matching, where the final answer is compared against a known correct solution. STaR (Zelikman et al., 2022) and SeRL (Fang et al., 2025c) assign binary rewards based on whether the generated rationale leads to the correct final answer. RISE (Qu et al., 2024) similarly assigns a reward of 1 for correct answers and 0 for incorrect ones. This approach is also used in iterative refinement settings: STaSC (Moskvoretskii et al., 2025), Bensal et al. (2025), ReGenesis (PENG et al., 2025), and Xiong et al. (2025) validate the final corrected response against the ground truth to reward the entire correction trajectory. SELF (Lu et al., 2024c) also incorporates a verification step to accept or reject the model's self-revisions based on task success.

For domains requiring rigorous correctness, such as mathematics and code generation, reward is derived from formal verifiers or compilers. Goedel-Prover (Lin et al., 2025b) and V-STaR (Hosseini et al., 2024) employ formal theorem provers or dedicated verifier models to strictly check the validity of a generated proof. In software engineering and agent tasks, ReVeal (Jin et al., 2026), CURE (Wang et al., 2025h), SOL-VER (Lin et al., 2025c), and SSR (Wei et al., 2025b) execute generated code against unit tests; the reward is strictly determined by the compiler's success or failure signals. Verification is also central to adversarial and self-play methods, particularly for the solver role: SiriuS (Zhao et al., 2025c), PAG (Jiang et al., 2025e), Absolute Zero (Zhao et al., 2025a), SPICE (Liu et al., 2025a), R-Zero (Huang et al., 2026a), and Dr. Zero (Yue et al., 2026) all rely on this deterministic verification to judge the final success of the interaction. SPHERE (Singh et al., 2025) likewise relies on verification of the final answer to label its chosen and rejected pairs for optimization. Zhou et al. (2025) also employ this verification strategy in their Self-Challenging framework to filter valid challenger-executor pairs.

### 4.3.3 Optimization

Optimization updates the model parameters from the generated data and the reward signal, and the choice of objective follows directly from the reward format; we simply note which method each approach uses. When the reward acts as a filter that keeps high-quality samples, the update is SFT on those samples, used by rejection-sampling and refinement methods such as STaR (Zelikman et al., 2022), LMSI (Huang et al., 2023a), ReST-MCTS* (Zhang et al., 2024a), SELF (Lu et al., 2024c), and SiriuS (Zhao et al., 2025c). When the reward is a preference ordering, direct preference optimization (DPO) and its variants align the model to chosen-over-rejected pairs, as in Self-Rewarding LMs (Yuan et al., 2024b), SynPO (Dong et al., 2025), Meta-rewarding (Wu et al., 2025c), and SPHERE (Singh et al., 2025). When the reward is a scalar, online RL algorithms optimize it directly, either with actor–critic methods such as PPO (SIRLC (Pang et al., 2024), TBV (Liu et al., 2025d), CURE (Wang et al., 2025h)) or with critic-free, group-based methods such as GRPO and its variants (R-Zero (Huang et al., 2026a), Dr. Zero (Yue et al., 2026), RLSR (Simonds et al., 2025), RESTRAIN (YU et al., 2026), SeRL (Fang et al., 2025c), MAE (Chen et al., 2025h)).

Self-play and adversarial settings often introduce specialized objectives, including the zero-sum game of SPIN (Chen et al., 2024i), the Nash-style updates of SPPO (Wu et al., 2025g) and RSPO (Tang et al., 2025b), the task-relative TRR++ of Absolute Zero (Zhao et al., 2025a), and the self-play SWE-RL of SSR (Wei et al., 2025b). Finally, many methods combine objectives, typically an SFT warm-up followed by preference or RL optimization, as in Goedel-Prover (Lin et al., 2025b), SOL-VER (Lin et al., 2025c), DTE (Srivastava et al., 2025), and Xiong et al. (2025).

### 4.3.4 Representative Instances of SGO

Although the SGO loop spans a large design space, most methods instantiate it through a few recurring patterns, which we highlight here as its most representative instances. These instances define the structural relationship between generation, reward, and optimization.

**Iterative Rejection Sampling.** The core philosophy is the model improves by distilling its own best generations into its weights. In this paradigm, the model first generates a diverse set of candidate solutions. These candidates are then filtered through a rigorous verification process: using either ground truth (Oracle) or statistical consistency (Majority Vote) to retain only the high-quality samples. Finally, these filtered "pseudo-labels" are used to fine-tune the model via SFT.

This approach is exemplified by STaR (Zelikman et al., 2022), which iteratively generates rationales and fine-tunes on those that lead to correct answers. LMSI (Huang et al., 2023a), CRESCENT (Sun et al., 2025c), and IWSI (Jiang et al., 2025a) extend this by utilizing consistency checking to filter reasoning paths in the absence of ground truth. ReGenesis (PENG et al., 2025) and SimRAG (Xu et al., 2025a) also follow this logic, employing retrieval or ground truth verification to construct high-quality training sets for iterative SFT.

**Self-Verification and Refinement.** The second paradigm shifts focus from simple filtering to self-verification and refinement, where the model plays an active role in evaluating and optimizing its own outputs. Unlike rejection sampling which treats the model as a black-box generator, this paradigm leverages the model's semantic capabilities to assign rewards (scoring/ranking) or to iteratively refine its answers. These self-generated signals are then used to update the policy, often via RL or DPO.

Approaches like Self-Rewarding LMs (Yuan et al., 2024b), Meta-rewarding (Wu et al., 2025c), and SIRLC (Pang et al., 2024) train the model to act as its own judge, updating parameters to maximize self-assigned scores. SPPO (Wu et al., 2025g) and SynPO (Dong et al., 2025) utilize the model to construct preference pairs for DPO. On the refinement side, methods like SELF (Lu et al., 2024c), RISE (Qu et al., 2024), and TRIPOST (Yu et al., 2024c) employ a "generate-critique-revise" loop, where the model explicitly learns to correct its errors based on internal critiques or external verifiers before the final update.

**Self-Play.** The third paradigm employs self-play, transforming the self-improvement process into a dynamic interaction between multiple roles. Instead of optimizing a static objective, the model improves by competing against an opponent or collaborating with peers. This interaction provides a curriculum of increasingly difficult challenges (adversarial) or robust consensus (collaborative), allowing the model to break free from the limitations of its initial distribution.

In adversarial settings, methods like SPIN (Chen et al., 2024i) formulate training as a zero-sum game between a generator and a discriminator. Absolute Zero (Zhao et al., 2025a), R-Zero (Huang et al., 2026a), SPICE (Liu et al., 2025a), and Dr. Zero (Yue et al., 2026) adopt a "Proposer-Solver" structure, where one agent generates novel problems and the other solves them, driving continuous evolution without human data. On the collaborative side, DTE (Srivastava et al., 2025), SiriuS (Zhao et al., 2025c), and Multiagent Finetuning (Subramaniam et al., 2025) utilize debate and cooperation among agents to filter hallucinations and converge on higher-quality reasoning chains.

## 4.4 Self-Evolving Optimization

Self-evolving optimization moves the target of improvement from the model's parameters to the optimization process itself: rather than only updating the weights through a fixed learning procedure, the model also revises that procedure, its update rules, learning algorithms, or surrounding agentic scaffolding, so that the way it improves can itself improve over time. Beyond the standard SGO framework, several recent studies have explored different model optimization pathways for self-improvement. Shi et al. (2025b) propose iterative self-incentivization, empowering models as "Agentic Searchers" capable of self-defining goals to navigate solution spaces. Lu et al. (2024a) demonstrate that LLMs can go beyond executing update rules to inventing new optimization algorithms for their own improvement. Pushing the boundaries of recursive self-modification, the Gödel Agent (Yin et al., 2025) and Darwin Gödel Machine (Zhang et al., 2026a) explore self-referential architectures that enable agents to recursively inspect and modify their own logic for open-ended evolution. Building on this line, the Huxley-Gödel Machine (Wang et al., 2026c) observes that current benchmark performance poorly predicts an agent's potential to yield better descendants, and instead guides the self-modification search by estimating clade metaproductivity, the aggregated performance of an agent's descendants, to approximate the optimal self-improving machine.

Notably, with the rise of agentic systems, model optimization has increasingly extended beyond improving the model in isolation to leveraging agent-level mechanisms for further enhancement. ASL (Sun et al., 2025a) exemplifies this trend by co-evolving three agentic roles: a Prompt Generator, a Policy Model, and a Generative Reward Model within a shared tool environment, enabling scalable open-domain self-learning without external supervision. SAGE (Wang et al., 2026b) incorporates a skill library into RL-based agent training, where skills generated from previous tasks accumulate and become available for subsequent ones, shifting improvement from the parameter level to the level of reusable agentic capabilities. EvolveR (Wu et al., 2025b) adopts a lifecycle perspective, alternating between offline distillation of interaction trajectories into abstract strategic principles and online retrieval-guided decision-making. Agent-R1 (Cheng et al., 2025) further extends the Markov decision process framework to systematically define RL-based training for LLM agents across diverse interactive environments.

We will introduce more of these agentic approaches in §5, particularly in §5.4. This shift from model-centric to agent-centric optimization reflects a broader trend: as self-improvement systems mature, the unit of improvement is no longer a single model but an entire agentic system comprising prompts, memory, tools, and workflows. Meanwhile, as discussed in §5.5, a complementary line of work explores test-time training, where models temporarily adapt their parameters during inference through self-generated feedback signals, further dissolving the boundary between training and deployment.

## 4.5 Discussion

This discussion centers on self-generated optimization, the core paradigm of model optimization in this stage and the setting that most surveyed methods target. We first review theoretical work on the SGO loop, covering the source of its improvement, its convergence and capability limits, and its collapse. We then analyze two axes along which its instances vary: how the reward signal is formulated, and how automated its data and model dependencies are. Next, we step back to contrast the three optimization paradigms as an overall trend. Finally, we situate optimization within the broader self-improvement pipeline.

### 4.5.1 Theoretical Analysis

The preceding subsections formalize model optimization as the SGO loop and survey the many instances built upon it, all of which share a striking premise: a model can become better by training on data it generated itself. A dedicated theoretical discussion is warranted here because model optimization is the most extensively studied stage of the self-improvement lifecycle, and the only stage where the model parameters are actually updated, which makes questions such as convergence and capability limits well-posed. Rather than a single theory, this premise has been probed from several angles, which we group into three themes: where the loop's improvement originates, whether it converges and to what ceiling, and when it collapses.

**Source of Improvement.** Huang et al. (2025a) conceptualizes the self-improvement process primarily as a "Sharpening" mechanism, arguing that these methods essentially redistribute probability mass from the distribution's tail toward high-quality outputs already present in the model's latent space. This mechanism is shown to depend on a critical "Generation-Verification Gap" (Song et al., 2025), where the model's discriminative capability must strictly exceed its generative performance to provide valid supervision signals. From an information-theoretic angle, Feng et al. (2026) characterize when a self-generated signal is actually informative: measuring response informativeness by the pointwise mutual information against a cross-model aggregate, they show the loop should upweight responses that disagree with the consensus, formalizing when self-produced supervision carries signal rather than noise.

**Convergence and Capability Limits.** RL-STaR (Chang et al., 2025) provides the formal convergence analysis of the iterative generation-optimization cycle in STaR, establishing conditions on pre-trained model quality for initiating effective improvement and proving that the policy converges to optimality even when occasional incorrect reasoning steps are incorporated. What it converges toward, however, is bounded: Sun et al. (2026a) provides a mathematical formalization of the Solver-Verifier dynamic, proving that the theoretical upper bound of improvement is constrained by the verifier's fidelity. Consistent with this ceiling, Qi et al. (2026) find that self-evolution reliably improves over the base model but plateaus once sufficient

Table 7: **Reward types used in model optimization for self-improvement.** We categorize reward signals into implicit and explicit types. Explicit rewards are further divided into binary, scalar, ordinal, and probability-based signals, reflecting different levels of supervision granularity.

| Reward Type | Granularity | Methods |
|---|---|---|
| Implicit Reward | — | STaR (Zelikman et al., 2022), LMSI (Huang et al., 2023a), TRIPOST (Yu et al., 2024c), RISE (Qu et al., 2024), SELF (Lu et al., 2024c), SPIN (Chen et al., 2024i), V-STaR (Hosseini et al., 2024), Goedel-Prover (Lin et al., 2025b), IWSI (Jiang et al., 2025a), STaPLe (Ramji et al., 2025), CRESCENT (Sun et al., 2025c), ReGenesis (PENG et al., 2025), Subramaniam et al. (2025), STaSC (Moskvoretskii et al., 2025), SimRAG (Xu et al., 2025a), SiriuS (Zhao et al., 2025c), SOL-VER (Lin et al., 2025c), Semantic Voting (Jiang et al., 2025b) |
| Explicit Reward | Binary | Bensal et al. (2025), SeRL (Fang et al., 2025c), TBV (Liu et al., 2025d), PAG (Jiang et al., 2025e), Self-Challenging (Zhou et al., 2025), SSR (Wei et al., 2025b) |
| | Scalar | SIRLC (Pang et al., 2024), SCoRe (Kumar et al., 2025), ReST-MCTS* (Zhang et al., 2024a), Xiong et al. (2025), DTE (Srivastava et al., 2025), RLSR (Simonds et al., 2025), CURE (Wang et al., 2025h), Absolute Zero (Zhao et al., 2025a), SPICE (Liu et al., 2025a), MAE (Chen et al., 2025h), ReVeal (Jin et al., 2026), RESTRAIN (YU et al., 2026), R-Zero (Huang et al., 2026a), Dr. Zero (Yue et al., 2026) |
| | Ordinal | Yuan et al. (2024b), ScPO (Prasad et al., 2025), SynPO (Dong et al., 2025), SPHERE (Singh et al., 2025), Ji et al. (2025), Wu et al. (2025c), DNPO (Yang et al., 2026) |
| | Probability | SPPO (Wu et al., 2025g), RSPO (Tang et al., 2025b) |

training compute is spent, still leaving a non-trivial generalization gap to oracle supervision, which indicates that internally generated supervision is fundamentally weaker than external labels.

**Collapse and Sustainability.** Shafayat et al. (2025) empirically and analytically examine whether the SGO loop can be sustained indefinitely under self-generated rewards, finding that while majority-voting-based self-reward initially improves both the model's performance and its own supervision quality, prolonged self-training inevitably leads to reward hacking and sudden performance collapse, highlighting feedback design as the central bottleneck for sustained self-improvement. Liu et al. (2026b) analyze self-play evolution from a data-pipeline perspective, arguing that genuine self-evolution requires the learnable information in self-synthesized data to increase across iterations, and proposing a Proposer-Solver-Verifier decomposition to diagnose when and why mode collapse occurs.

We provide a more extensive discussion of the challenges and practical failure modes arising from these theoretical insights in §7, particularly regarding the flawed feedback signals (§7.2) and optimization-driven failures (§7.3).

### 4.5.2 Reward Formulation

**Implicit vs Explicit Reward.** Beyond categorizing methods by how rewards are obtained, we further distinguish reward signals by how they are utilized during the self-improvement process. Specifically, we differentiate between implicit rewards and explicit rewards.

Implicit rewards do not appear as explicit feedback signals during optimization. Instead, reward signals are implicitly encoded through data selection or filtering mechanisms, where the reward function serves as an evaluation criterion rather than a direct optimization objective. Representative examples include majority voting, which treats consensus among multiple generations as a proxy for quality, and rejection sampling, which retains only generations satisfying heuristic criteria. Although no explicit reward value is propagated, implicit rewards shape learning by biasing the training data distribution toward higher-quality outputs.

Explicit rewards, in contrast, provide directly accessible feedback signals that can be explicitly incorporated into the learning process. Depending on their form, explicit rewards can be optimized via reinforcement learning objectives or used to construct structured supervision signals for preference-based optimization. We categorize explicit rewards into four common types:

- **Binary**: indicating success or failure (e.g., correct/incorrect).

- **Scalar**: assigning real-valued scores that reflect output quality.

- **Ordinal**: providing relative rankings or preference orders among outputs.

- **Probability-Based**: representing calibrated likelihoods or confidence scores over candidate outputs.

Based on this taxonomy, we categorize all methods into implicit-reward and explicit-reward approaches, with a consolidated overview provided in Table 7.

**Process vs Outcome Reward.** In addition to reward utilization and representation, we further characterize reward signals by their granularity. Specifically, we distinguish between **process reward** and **outcome reward**. Process reward provides feedback at intermediate steps of reasoning or along the generation trajectory, enabling fine-grained supervision of the model's decision-making process. In contrast, outcome reward evaluates only the final output. Except for a few methods (Zhang et al., 2024a; Xiong et al., 2025; Jiang et al., 2025e), the majority of methods rely on outcome reward. This preference suggests that outcome reward, owing to its simplicity, low annotation cost, and ease of integration across diverse tasks and optimization frameworks, offers greater generality and practicality while still achieving good empirical performance compared to fine-grained process reward.

### 4.5.3 Degree of Automation

When discussing self-improvement, an important orthogonal perspective is the degree of automation involved in the improvement process. Automation measures how little external input the loop still requires, so we analyze it along the two kinds of external input an SGO loop can draw on: its dependence on externally provided data (prompts and answers) and its dependence on auxiliary models (teacher models, reward models, or verifiers). A more automated method needs less of each, and in the limit the model supplies both the data and the supervision itself.

**Data Dependency.** The first dimension concerns whether model optimization starts from an annotated dataset, and if so, which components of that dataset are required. A given dataset typically consists of prompts or questions $x$ paired with corresponding ground-truth answers $y$. At the lowest level of automation, both $x$ and $y$ are required, relying on fully annotated datasets. An intermediate setting has access to prompts $x$ only, while the corresponding answers $y$ are autonomously generated or inferred by the model. At the highest level of automation, model optimization relies on no given dataset at all, as the model produces both the inputs $x$ and their supervision $y$ itself. These three levels correspond directly to the three seed cases of the SGO loop in Algorithm 1: the fully annotated $\mathcal{D} = \{(x, y)\}$, the prompt-only $\mathcal{D} = \{x\}$, and the empty

Table 8: **Degree of automation in model optimization with respect to data dependency.** Methods are categorized by their reliance on externally collected prompts and ground-truth annotations: high automation requires neither, intermediate uses either one, and low automation depends on both.

| Low Automation | Intermediate Automation | High Automation |
| --- | --- | --- |
| STaR (Zelikman et al., 2022) | LMSI (Huang et al., 2023a) | Yuan et al. (2024b) |
| TRIPOST (Yu et al., 2024c) | SIRLC (Pang et al., 2024) | CRESCENT (Sun et al., 2025c) |
| RISE (Qu et al., 2024) | SELF (Lu et al., 2024c) | SynPO (Dong et al., 2025) |
| SCoRe (Kumar et al., 2025) | ScPO (Prasad et al., 2025) | RLSR (Simonds et al., 2025) |
| ReST-MCTS* (Zhang et al., 2024a) | DTE (Srivastava et al., 2025) | Ji et al. (2025) |
| Xiong et al. (2025) | Goedel-Prover (Lin et al., 2025b) | SeRL (Fang et al., 2025c) |
| SPIN (Chen et al., 2024i) | SPPO (Wu et al., 2025g) | SOL-VER (Lin et al., 2025c) |
| V-STaR (Hosseini et al., 2024) | STaPLe (Ramji et al., 2025) | Self-Challenging (Zhou et al., 2025) |
| SiriuS (Zhao et al., 2025c) | Subramaniam et al. (2025) | SSR (Wei et al., 2025b) |
| IWSI (Jiang et al., 2025a) | SimRAG (Xu et al., 2025a) | Absolute Zero (Zhao et al., 2025a) |
| ReGenesis (PENG et al., 2025) | Semantic Voting (Jiang et al., 2025b) | SPICE (Liu et al., 2025a) |
| STaSC (Moskvoretskii et al., 2025) | CURE (Wang et al., 2025h) | MAE (Chen et al., 2025h) |
| Bensal et al. (2025) | Wu et al. (2025c) | R-Zero (Huang et al., 2026a) |
| SPHERE (Singh et al., 2025) | ReVeal (Jin et al., 2026) | Dr. Zero (Yue et al., 2026) |
| TBV (Liu et al., 2025d) | RESTRAIN (YU et al., 2026) | |
| PAG (Jiang et al., 2025e) | | |
| RSPO (Tang et al., 2025b) | | |
| DNPO (Yang et al., 2026) | | |

$\mathcal{D} = \emptyset$. Climbing this ladder, the model substitutes its own generation for each missing component, self-proposing prompts in place of $x$ and self-generating pseudo-labels in place of $y$. We summarize representative methods along this dimension in Table 8.

**Model Dependency.** The second dimension focuses on the models involved in the self-improvement loop. Some approaches rely solely on the target model itself, using self-generated outputs, self-heuristic, or internal scoring mechanisms. In contrast, other methods depend on external components, such as stronger teacher models, reward models, or verification systems, to provide supervision or evaluation signals. Except for a limited subset of methods (Yu et al., 2024c; Wu et al., 2025g; Singh et al., 2025; Jiang et al., 2025b; Tang et al., 2025b; Yang et al., 2026) that explicitly depend on external models, the majority of existing work achieves self-improvement without relying on additional external models.

Overall, as foundation models continue to scale in capability, self-improvement paradigms are increasingly moving toward higher degrees of automation. In the long run, truly autonomous self-improvement is expected to evolve toward settings that require neither externally annotated data nor auxiliary models, with both supervision signals and optimization guidance emerging entirely from the model itself.

### 4.5.4 Trends across the Three Paradigms

Viewed together, direct optimization, self-generated optimization, and self-evolving optimization form a progression rather than three isolated categories. Direct optimization treats the model as a self-supplier of data but leaves the optimization step itself a fixed, one-shot offline fit, so its ceiling is set by the corpus the upstream stages can assemble. SGO closes the loop by pulling generation and reward inside optimization, so that the model produces the experience it trains on and, increasingly, the reward that scores it, turning the process into a non-stationary reinforcement-learning loop that most current methods target. Self-evolving optimization takes the final step of making the loop's own machinery, the update rule, the architecture, or

the surrounding agentic scaffolding, the object of improvement. Across this progression, the model absorbs successively more of the pipeline into its own control, reliance on fixed external data and hand-designed algorithms steadily decreases, and the unit of improvement grows from a single set of weights to an entire agentic system. This rising autonomy comes at the cost of rising instability and weaker guarantees, which is why direct optimization remains the most reliable choice when good data can be prepared in advance, SGO dominates when the model can generate and verify its own experience, and self-evolving optimization, though the most open-ended, is still the least mature.

### 4.5.5 Interplay Between Optimization and Data

Model optimization is the stage that converts data into capability, and the data handed over by acquisition and selection determines both what it works on and what it can produce. Acquisition sets the space of experiences available to learn from and selection decides which of them are actually trained on; together they fix the input to optimization and thereby bound the capability that any amount of optimization can extract from it. The admitted samples are the very material of the SGO loop, on which the model generates responses, receives rewards, and updates its parameters, so the nature and quality of this data foundation translate directly into what the model can become. If the admitted samples are too easy or too hard, the model succeeds or fails on nearly everything it generates and the reward signal carries little information; if mislabeled samples slip through, they are not merely wasted but reinforced into the parameters. Meanwhile, each optimization round shifts what the model has already mastered, so experiences that were valuable in the previous round may contribute little in the next, calling for acquisition and selection to keep updating in step with training. In effect, the data stages decide what optimization can turn the model into, while optimization keeps redefining which experiences are worth acquiring and selecting.

## 5 Inference Refinement for Self-Improvement

### 5.1 Overview

Inference-time refinement refers to the set of mechanisms that enhance a model's performance during the inference phase, often without requiring permanent updates to its underlying parameters. While training-time optimization focuses on building a better "brain", inference refinement focuses on making that brain "think" more effectively. This stage is necessary because even the most advanced models frequently fail to produce the most accurate result on their first attempt, especially in complex, multi-step tasks.

These approaches treat the model as a self-contained reasoning entity, improving output quality through decoding or reasoning-based modifications, and can be applied whether the underlying system uses a single LLM or an ensemble of models. Agentic systems extend beyond the core model concept to harness a richer set of capabilities and external interfaces that support long-horizon interaction with an environment. Agentic improvement encompasses not only output refinement but also enhancements to memory architectures, tool usage, prompt and workflow orchestration, and interaction dynamics, which together allow the system to accumulate experience, leverage context, and plan actions across multiple steps. As shown in Figure 7, these improvement mechanisms for inference refinement can be grouped into the following categories:

- **Decoding Strategies** guide output generation at the token or sequence level, improving both the quality of outputs and the efficiency of generation at inference time without parameter updates.

- **Reasoning-Based Improvement** structures the model's intermediate thought process for planning and iterative decision-making, allowing it to think before it speaks.

- **Agentic System-Based Improvement** extends refinement beyond the core model by enhancing system-level components, including prompts, memory architectures, tool use, and workflow design.

- **Test-Time Training** modifies the model parameters at the moment of inference, enabling task-specific policy evolution that is uniquely tailored to the current query. Unlike static deployment, this allows the model to perform local gradient updates based on the specific context or provided constraints.

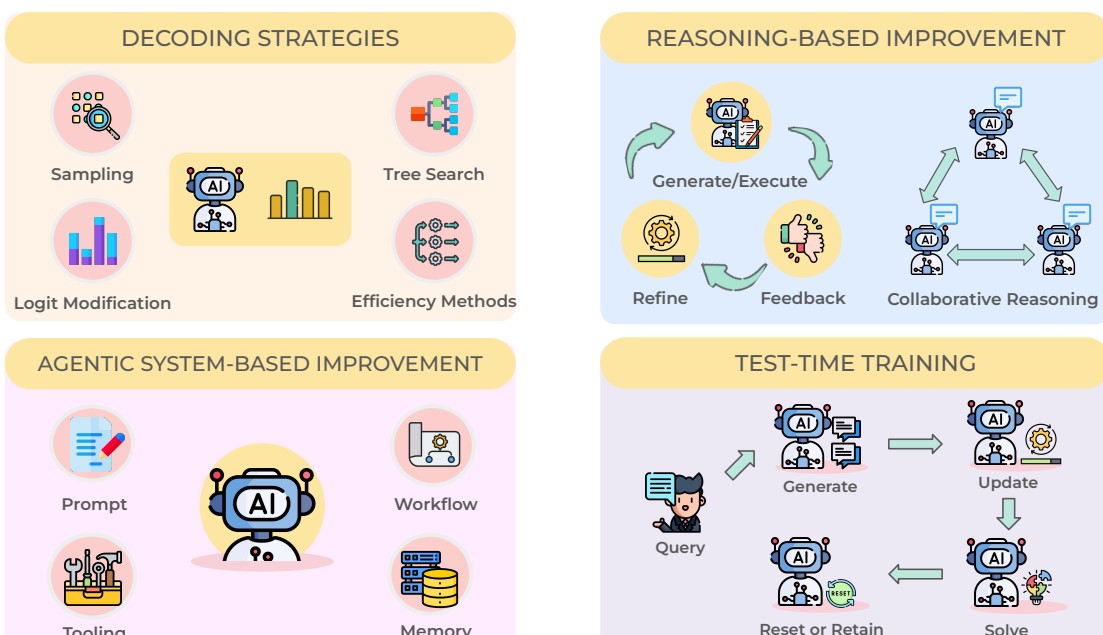

Figure 7: **Overview of inference refinement in the self-improvement system.** This stage focuses on how the model improves output quality during inference without permanently updating its parameters. **(i) Decoding Strategies:** The model improves generation by modifying decoding procedures, such as sampling control, logit modification, tree search, or efficiency-oriented decoding methods. **(ii) Reasoning-Based Improvement:** The model refines reasoning trajectories through iterative processes such as generate–execute–refine loops, feedback-based revision, or collaborative reasoning among multiple agents. **(iii) Agentic System-Based Improvement:** The model operates within agentic systems that coordinate prompts, tools, workflows, and memory modules to enhance task performance. **(iv) Test-Time Training:** The model temporarily adapts its parameters during inference by generating feedback signals and updating the model before producing the final solution.

Inference refinement effectively extends self-improvement beyond the pre-deployment phase into a continuous, real-time process. By integrating these test-time mechanisms with training-time optimization, the system sustains an active evolutionary loop during actual deployment. This synergy empowers the model to adaptively bridge the gap between its static internalized knowledge and the evolving demands of complex real-world tasks, leading to a state of persistent and autonomous evolution. In the following section, §5.2 presents decoding strategies for improving inference quality and efficiency. §5.3 then discusses reasoning-based refinement methods that enhance intermediate reasoning processes. §5.4 further explores agentic system-based approaches, incorporating tools, memory, and workflows. Finally, §5.5 introduces test-time training methods, followed by an overall discussion of inference refinement for self-improvement in §5.6.

## 5.2 Decoding Strategies

In this section, we review decoding strategies through the lens of self-improvement, rather than as a general survey of decoding. Decoding lets the model improve its own outputs at inference time without any parameter update, and it does so in a self-improving way: the model reshapes how it generates to reach better outputs on its own. We organize the discussion around two roles decoding plays in this process. First, decoding lets the model surface and select higher-quality outputs from its own distribution; this both improves responses at inference and yields better samples that can be fed back as training data for the optimization stage, closing the self-improvement loop. Second, efficient decoding keeps the repeated generations that iterative self-improvement relies on feasible at scale, so that the loop can be run many times. We organize the reviewed works by their decoding strategy in Table 9.

### 5.2.1 Sampling-Based

Sampling-based strategies draw multiple candidate outputs from the model's own distribution and then select or combine them into a single answer that is more reliable than any individual generation, whether through consensus (self-consistency), verifier scoring (best-of-$N$), or fusion (synthesis). They constitute a form of inference-time self-improvement: a fixed model bootstraps a better output from its own samples without any parameter update, and the higher-quality outputs surfaced this way can in turn be fed back as training data for the optimization stage.

**Self-Consistency.** Self-Consistency (Wang et al., 2023b) generates multiple reasoning paths via stochastic decoding and identifies the most frequent response, following the intuition that the correct reasoning processes, even if they are diverse, tend to reach the same final answer. Universal Self-Consistency (USC) (Chen et al., 2024g) alters this mechanism for free-form textual tasks by utilizing an LLM for determining consistency of candidates. Another extension weighs each sample by a self-assessed confidence score and performs a confidence-weighted majority vote (Taubenfeld et al., 2025; Razghandi et al., 2025). The general form of majority voting based decoding can also be conditioned on different trajectories to improve generation robustness such as in multimodal applications for hallucination reduction (Fang et al., 2025d). While self-consistency is a simple approach that can improve robustness, it often struggles in scenarios where the model exhibits systemic biases, as the aggregation of multiple reasoning paths can converge on a consistently held incorrect conclusion rather than correcting the error (Chen et al., 2024a; Byerly & Khashabi, 2025).

**Best-of-N.** Best-of-N extends the principle of majority voting with a scoring mechanism such as a trained external verifier (Cobbe et al., 2021), reward model (Stiennon et al., 2020), or heuristic such as model uncertainty (Zhu et al., 2025b; Kang et al., 2025b) to score and select the best response from candidates. These techniques improve reasoning accuracy and reduce hallucinations by exploiting diversity in the model's sampling distribution. Scaling methods for Best-of-N include increasing the number of verifiers to create a more diverse verification signal (Lifshitz et al., 2025) or increasing the number of LLMs which sample candidates at inference time for diverse generation (Xia et al., 2025). However, the effectiveness of scaling candidate generation is limited by the precision of the verifier. For tasks without automatic verifiers, performance tends to plateau because standard selection methods like majority voting and reward models fail to identify rare correct samples as the number of generations scales (Brown et al., 2024).

**Synthesis.** The previous methods operate on the assumption that correct answers exist within the candidate set, but in cases where all candidates are incorrect, they fail to produce a correct answer. Synthesis methods solve this problem by analyzing candidate responses generated by a single LLM or an ensemble to formulate a final solution. Vernikos et al. (2024) and LLM-Blender (Jiang et al., 2023) leverage a task-specific, generative fuser or synthesis model to combine candidate responses while Mixture-of-Agents (Wang et al., 2025a) leverages the reasoning capability of an off-the-shelf LLM for fusion. This approach is extended by CoT-based Synthesizer (Zhang et al., 2025b) which utilizes a CoT model for candidate fusion, and GSR (Wang et al., 2025c) by removing the fuser model and performing candidate synthesis by re-prompting the LLM with the candidate solutions. Synthesis methods tend to outperform majority voting-based methods on difficult problems where there are few correct candidate responses as they are not strictly limited by the quality of the candidate responses. However, synthesis-based methods incur substantial computational cost which scales with the number of utilized LLMs and the number of generated candidates.

### 5.2.2 Tree Search

Tree search enables the model to systematically explore and compare alternative reasoning paths, selecting higher-quality outputs than a single decoding pass would yield.

**Beam Search.** Beam search maintains a fixed number of top candidate continuations at each decoding step, balancing exploration and exploitation to increase the likelihood of finding higher-probability sequences. Variants can utilize internal self-evaluation (Xie et al., 2023c) to guide step-wise decoding toward logically coherent trajectories while other approaches derive evaluation signals from verifiers, reward models, or retrieval mechanisms, to prune beam candidates for enhanced accuracy and consistency across reasoning steps

(Trinh et al., 2024; Zhu et al., 2024). Adaptive strategies dynamically adjust the search process to balance computational efficiency and reasoning, including dynamically narrowing beam size (Hu et al., 2025b) and re-allocating the search budget to steps more prone to reasoning errors (Quamar et al., 2025).

**Monte Carlo Tree Search.** Monte Carlo Tree Search (MCTS) traverses a decision tree by iteratively exploring and simulating candidate paths to identify high-performing trajectories. For token-level MCTS, LLMs provide next-token probabilities that serve as heuristics to guide the MCTS decoding (Zhang et al., 2023). A popular extension is value-function-guided MCTS, which uses learned value functions to rank paths and guide token-level search, used by frameworks like PPO-MCTS (Liu et al., 2024b). Most MCTS-based approaches re-frame language model inference as an explicit search process over branching sequences of reasoning or solution states. Tree-of-Thought (Yao et al., 2023b) uses tree search methods such as MCTS variants or simpler methods such as breadth-first or depth-first search to navigate a tree of intermediate reasoning based on the LLM's self-assessment of candidate thoughts. Many approaches utilize a learned value function with MCTS allowing for more precise guidance and enabling increased search depth (Feng et al., 2024; Zhang et al., 2024b). Other search variants have been proposed including Graph-of-Thoughts (Besta et al., 2024) which generalizes search to a directed graph structure, and Forest-of-Thought (Bi et al., 2025) which maintains multiple trees in parallel aggregating solutions across trees. The integration of structured search procedures transforms decoding into a formal sequential decision-making process, enabling models to overcome the linear limitations of greedy auto-regressive generation.

### 5.2.3 Logit and Probability Adjustments

The methods below improve a fixed model's outputs by reshaping its token-level probabilities at decoding time rather than by updating its parameters. They connect to self-improvement in that the guiding signal is typically produced by the model itself, such as a reward model trained on its own generations or a contrast against a reference version of the same model, so the model steers and refines its own outputs at inference time.

**Reward Guidance.** Reward model (RM) alignment is a form of self-improvement integrated directly into the decoding process, where the model's probabilistic token predictions are dynamically adjusted using a signal that reflects human preferences or task-specific objectives. Reward-guided alignment shifts the alignment process from training time to decoding time, serving as an inference-time analogue to traditional RLHF. The ARGS (Khanov et al., 2024), PAD (Chen et al., 2025b), and Controlled Decoding (CD) (Mudgal et al., 2024) frameworks enhance alignment of generated text with human preferences by scoring partial trajectory continuations which serve as a reward mechanism for token-level guidance of the text generation process. A reward model or value function is utilized to evaluate the expected reward of each candidate token, allowing the system to steer the generation process toward high-reward outcomes. While these methods typically evaluate the next immediate token candidates, DeAl (Huang et al., 2025b) extends this paradigm by employing a heuristic-guided search which utilizes short greedy lookaheads to estimate future rewards of a trajectory. GenARM (Xu et al., 2025c) uses an autoregressive reward model designed to provide direct token-level guidance and addresses prior works' reliance on applying trajectory-level RMs trained only on complete responses when evaluating partial sequences. Reward guidance enables efficient alignment of LLMs and can recover a significant amount of the performance gap versus full fine-tuning, offering a favorable accuracy-cost trade-off (Xu et al., 2025c). However, this often comes at the cost of having to train a dedicated reward model.

**Reference Guidance.** A further line of work refines outputs by adjusting token logits with respect to a reference distribution. One family of methods linearly blends the logits of a base model and a reference model, or of an ensemble of models, steering generation toward more aligned or higher-quality outputs while allowing the degree of alignment to be controlled at decoding time (Liu et al., 2024e; Shi et al., 2024; Mavromatis et al., 2024; Zhu et al., 2026). Another family contrasts the predictions of a stronger model against a weaker one (Li et al., 2023b; O'Brien & Lewis, 2023), or the predictions of different internal layers of the same model (Chuang et al., 2024), thereby suppressing generic, repetitive, or hallucinated continuations.

These approaches shift alignment and control from training time to inference time by directly adjusting next-token probabilities, using either external reward signals or reference model distributions. By operating at the level of logits, they trade additional decoding computation for fine-grained control over generation, allowing the model to produce better-aligned and higher-quality outputs without any parameter updates.

### 5.2.4 Efficiency-Oriented Methods

While the aforementioned approaches aim to improve test-time accuracy, decoding strategies such as parallel decoding and speculative decoding are primarily designed to enhance inference efficiency. These approaches can benefit the self-improvement loop in two ways. During inference time, faster generation enables more timely feedback on the model performance, whose evaluation guides the improvement iterations. In addition, during training, efficient generation can also speed up response rollout during self-generation based model optimization, which can substantially accelerate the self-improvement process due to its recursive nature.

**Parallel Decoding.** Rather than generating tokens sequentially, these methods re-frame autoregressive decoding into a parallel formulation to accelerate inference. This includes Jacobi-based approaches, which apply Jacobi and Gauss-Seidel fixed-point iteration methods to predict multiple tokens simultaneously (Santilli et al., 2023; Fu et al., 2024). Non-autoregressive generation that produces all output tokens together has been explored in tandem with the release of the Transformer architectures, such as in applications of neural machine translation (Gu et al., 2018; Wang et al., 2018; Ghazvininejad et al., 2019; Lee et al., 2020; Zhou & Keung, 2020; Brimacombe & Zhou, 2023). Recently, this paradigm resurgence manifests in the form of diffusion language models (Li et al., 2022; Sahoo et al., 2024; Nie et al., 2025) to generate different tokens in a sequence in parallel. From an architectural perspective, multi-token prediction (MTP) (Gloeckle et al., 2024) trains the model with multiple output heads that predict several future tokens at once, which not only improves sample efficiency and downstream performance but also enables self-speculative decoding with up to $3\times$ faster inference. Additionally, beyond the token-level parallel decoding, Skeleton-of-Thought (SoT) decoding (Ning et al., 2024) improves efficiency by first generating a high-level outline of a response and then expanding each segment in parallel. The idea of parallel generation is incorporated in many LLM ensembles. Systems can determine which aspects of a task are parallelizable, allowing them to be executed asynchronously by different agents (Yu et al., 2025c; Xia et al., 2025).

**Speculative Decoding.** Speculative decoding accelerates autoregressive generation by using a lightweight model to draft multiple candidate tokens that are then verified in parallel by the base model, committing only those that pass verification (Leviathan et al., 2023; Chen et al., 2023; Xia et al., 2023). This paradigm significantly enhances inference efficiency by enabling the simultaneous verification of multiple drafted tokens in a single parallel step, circumventing the sequential delays inherent in standard autoregressive decoding. Other lines of research drop the drafting model and integrate additional decoding heads onto the target model to enable parallel, speculative drafting (Cai et al., 2024a; Li et al., 2024j). Self-drafting models also achieve acceleration through adaptively skipping intermediate layers (Zhang et al., 2024c) or implementing early exiting (Liu et al., 2024c) to more efficiently generate draft tokens from the base model's own internal representations. In addition, retrieval is also incorporated to provide draft tokens for speculative decoding (He et al., 2024). Approaches can also relax the constraint of keeping the target generation the same as the original model, such as Chunk-Distilled Language Modeling (CD-LM) (Li et al., 2025i) that improves both decoding speed and adaptation of generation to new knowledge through chunk-level data distillation with fine-grained in-context retrieval at inference.

Finally, emerging paradigms explore token-efficient decoding through multimodal architectures that process text as images (Li et al., 2025h; Wei et al., 2025a).

### 5.3 Reasoning-Based Improvement

Reasoning-based improvement methods extend beyond decoding-level enhancements by focusing not only on how outputs are generated, but also on the dynamic, multi-step process of how models think, plan, and refine their reasoning. These approaches introduce structured inference-time mechanisms that enable models to incorporate feedback, decompose complex tasks into intermediate subgoals, or distribute reasoning across

Table 9: **Summary of decoding-based inference-time self-improvement methods.** Early approaches improve generation by modifying the decoding process to enhance output quality and efficiency without updating model parameters.

| Decoding Strategy | Methods |
|---|---|
| Self-Consistency | SC (Wang et al., 2023b), USC (Chen et al., 2024g), CISC (Taubenfeld et al., 2025), CER (Razghandi et al., 2025) |
| Best-of-N | Stiennon et al. (2020), Cobbe et al. (2021), MAV (Lifshitz et al., 2025), UnCert-CoT (Zhu et al., 2025b), Self-Certainty (Kang et al., 2025b) |
| Synthesis | LLM-Blender (Jiang et al., 2023), LMCor (Vernikos et al., 2024), GSR (Wang et al., 2025c), MoA (Wang et al., 2025a), CoT-based Synthesizer (Zhang et al., 2025b) |
| Beam Search | Xie et al. (2023c), DBS (Zhu et al., 2024), AlphaGeometry (Trinh et al., 2024), PRM-BAS (Hu et al., 2025b), AdaBeam (Quamar et al., 2025) |
| Tree Search | ToT (Yao et al., 2023b), TS-LLM (Feng et al., 2024), MCTSr (Zhang et al., 2024b), PPO-MCTS (Liu et al., 2024b) |
| Reward Guidance | ARGS (Khanov et al., 2024), Controlled Decoding (Mudgal et al., 2024), DeAL (Huang et al., 2025b), GenARM (Xu et al., 2025c), PAD (Chen et al., 2025b) |
| Reference Guidance | CD (Li et al., 2023b), O'Brien & Lewis (2023), DeRa (Liu et al., 2024e), MOD (Shi et al., 2024), Pack of LLMs (Mavromatis et al., 2024), DoLa (Chuang et al., 2024), InRa (Zhu et al., 2026) |
| Parallel Decoding | NAR (Gu et al., 2018; Ghazvininejad et al., 2019; Zhou & Keung, 2020), PGJ (Santilli et al., 2023), Fu et al. (2024), Diffusion LM (Li et al., 2022; Sahoo et al., 2024; Nie et al., 2025), MTP (Gloeckle et al., 2024), SoT (Ning et al., 2024), DynTaskMAS (Yu et al., 2025c), Xia et al. (2025) |
| Speculative Decoding | Speculative Decoding (Leviathan et al., 2023), SpS (Chen et al., 2023), SpecDec (Xia et al., 2023), Medusa (Cai et al., 2024a), Eagle (Li et al., 2024j), Zhang et al. (2024c), EESD (Liu et al., 2024c), REST (He et al., 2024), CD-LM (Li et al., 2025i) |

multiple interacting agents. These methods effectively instantiate a model optimization loop at inference time, where reasoning trajectories are iteratively improved through feedback, enabling refinement through experience rather than explicit policy updates.

### 5.3.1 Feedback-Based Reasoning

Feedback-based reasoning transforms static inference into a dynamic, closed-loop process by integrating evaluative signals to iteratively refine model outputs. This mimics the self-generated optimization loop of model optimization, but uses signals for output refinement rather than parameter updates. Refinement loops can be categorized by the source of feedback: feedback may be generated internally by the model through self-evaluation or obtained from external sources such as critique models, environments, or verification signals. Self-feedback methods generally follow a reason–critique–refine paradigm, where the model evaluates and updates its own outputs, while external feedback approaches may additionally incorporate reason–execute–refine loops, in which solutions are validated through interaction with an environment.

**Self-Feedback.**  Independent iterative improvement does not rely on external models to give feedback in the improvement loop. Early works including Self-Refine (Madaan et al., 2023) and RCI (Kim et al., 2023) propose iterative self-refinement loops that utilize the base LLM for feedback and refinement of a response. Feedback takes the form of a specific improvement, which is integrated into the prompt for the next iteration. Natural language self-critiques can also be applied at the step level of a reasoning chain, acting as a natural language PRM (Yansi Li et al., 2025). The Self-Debugging framework (Chen et al., 2024h) performs "rubber ducking" for code translation where self-feedback is derived from the consistency of the model's explanation of its own code with the problem description and predicted behavior. Internal confidence-based methods like IoE (If-or-Else) (Li et al., 2024d) leverage the model's uncertainty for selective refinement of low-confidence reasoning steps, which acts as an internal filter to prevent over-criticizing and incorrectly altering highly confident responses.

Iterative self-improvement behaviors can be instilled during training and employed at inference time without external feedback. These approaches internalize self-correction during training such that the model autonomously performs self-refinement once deployed. One approach integrates self-verification or self-refinement into its RL training objective to teach LLMs the explicit ability to recursively detect and correct previous mistakes over sequential turns (Qu et al., 2024; Zeng et al., 2025). The Self-rewarding correction framework (Xiong et al., 2025) unifies a generator and evaluator into a single LLM and is trained to generate explicit self-assigned reward tokens alongside its reasoning, serving as an internal evaluation signal to guide iterative refinement. Current LLMs often exhibit limited capacity to enhance their own outputs through intrinsic mechanisms alone, as they are limited by the quality and consistency of the model's self-assessment of its reasoning (Huang et al., 2024a). These methods are highly sensitive to inference conditions: negative prompts can encourage over-criticism, and their internal decision-making process lacks stability, often requiring operation under zero-temperature decoding to prevent randomness from flipping the self-correction decision (Liu et al., 2024a).

**External Feedback.**  These methods rely on feedback from an external critique model that provides actionable suggestions to iteratively refine a response. The Reflexion (Shinn et al., 2023) and DCR (Detect Critique Refine) (Wadhwa et al., 2024) frameworks both employ evaluator/detector models to compute a scalar reward reflecting the agent's performance, which is interpreted by a critique model to provide specific textual feedback. PerFine (Maram et al., 2025) leverages a critic LLM for personalized text generation. The critic LLM explicitly generates detailed and actionable guidance to align the tone, vocabulary, and topic relevance of a generated response. Other frameworks (Paul et al., 2024; Xi et al., 2024b; Hossain et al., 2026) extend the use of a critic model to provide fine-grained textual feedback on intermediate reasoning steps. These methods provide rich and actionable feedback signals, but their effectiveness is limited by the accuracy and consistency of the critic.

In contrast to critique-model approaches that rely on a dedicated LLM to generate natural-language feedback, these methods receive supervision from environmental observations or external evaluators that provide scalar scores, binary results, or verification signals. The CRITIC framework (Gou et al., 2024) derives signals from calls to external tooling (search engines, code interpreters) to verify candidates, while TPO (Test-Time Preference Optimization) (Li et al., 2025g) harnesses the ability of the base model to interpret numerical reward signals from a reward model into textual critiques. In these works, the base model interprets the external signal into a natural-language critique or is prompted to diagnose its failures.

This paradigm is common for code generation and verification, as it mimics human debugging by enabling the model to refine flawed code based on execution feedback such as test cases and runtime errors (Ni et al., 2023). Feedback is then stored in the conversation history or appended to the next prompt. Self-Debugging (Chen et al., 2024h) can generate a reflective message from the combination of post-execution feedback (execution traces or unit test results) and a self-explanation of its generated code as a refinement signal for each iteration. LDB (Large Language Model Debugger) (Zhong et al., 2024a) decomposes programs into smaller blocks and uses intra-execution feedback from a debugger, tracking variable states at different breakpoints. The AlphaVerus framework (Aggarwal et al., 2025) distills feedback from a formal verifier into actionable critiques within its formally verified code generation process. The verifier returns objective feedback on the generated code through its localized error messages, which are used to generate refinements

Table 10: **Summary of feedback-based inference-time refinement methods.** These approaches improve generation through multi-step feedback loops, leveraging diverse feedback sources to iteratively refine outputs without updating model parameters.

| Method | Feedback Source | Feedback Type |
|---|---|---|
| Self-Refine (Madaan et al., 2023) | Base LLM | Textual Critique |
| Reflexion (Shinn et al., 2023) | External Evaluator | Score → Textual Critique |
| RCI (Kim et al., 2023) | Base LLM or Verifier | Textual Critique or Verification |
| LEVER (Ni et al., 2023) | Verifier | Score |
| Self-Debugging (Chen et al., 2024h) | Base LLM or Verifier | Textual Critique or Verification |
| IoE Prompt (Li et al., 2024d) | Base LLM | Internal Confidence |
| CRITIC (Gou et al., 2024) | External tools | Execution Results |
| LDB (Zhong et al., 2024a) | Debugger LLM | Textual Critique |
| REFINER (Paul et al., 2024) | Critique LLM | Textual Critique |
| DCR (Wadhwa et al., 2024) | Critique LLM | Textual Critique |
| RISE (Qu et al., 2024) | Base LLM | Internalized |
| PANEL (Yansi Li et al., 2025) | Base LLM | Textual Critique |
| PerFine (Maram et al., 2025) | Critique LLM | Textual Critique |
| AlphaVerus (Aggarwal et al., 2025) | Verifier | Score → Textual Critique |
| TPO (Li et al., 2025g) | Reward Model | Score → Textual Critique |
| ID-Sampling (Chen et al., 2025e) | Base LLM | Textual Critique |
| Xiong et al. (2025) | Base LLM | Internal Reward Token |

structured as a guided tree search. While these approaches provide highly targeted and objective feedback, they are limited to tasks with well-defined verification signals and require the model to correctly interpret the feedback. We organize these feedback-based reasoning methods in Table 10.

### 5.3.2 Planning-Based Reasoning

Planning-based reasoning methods enable models to tackle complex objectives by breaking down a high-level goal into smaller, manageable sub-goals that define the sequence of steps and interactions required to achieve the goal.

**Open-Loop Planning.** Planning in natural language allows the model to explore a wider range of conceptual ideas, which guides generation towards improved outcomes and leads to successful gains in final generation accuracy (Wang et al., 2024a). Initial approaches to planning focused on structured prompting techniques designed to elicit a global blueprint from the model prior to response generation (Zhou et al., 2023a; Wang et al., 2023a). By explicitly decoupling the planning stage from the final reasoning execution, these methods introduced a systematic decomposition of tasks, requiring the model to break down complex queries into manageable sub-tasks before generating a final answer. Plans can also be represented in symbolic forms like Planning Domain Definition Language (PDDL), which is better suited for environments with intricate constraints, where direct LLM-generated plans often suffer from correctness and executability issues (Huang et al., 2024b). In this paradigm, the LLM serves primarily as a formalizer, translating natural language tasks into PDDL (Liu et al., 2023; Guan et al., 2023). These approaches rely on the ability to anticipate all necessary steps upfront, ensuring a structured plan is established prior to execution.

**Closed-Loop Planning.** Inference-time planning equips agents with decision-making capabilities, enabling them to handle multi-step tasks by dynamically modifying their future plan of actions. ReAct (Yao et al., 2023c) pioneered the use of environmental feedback for inference-time plan improvement by interleav-

ing reasoning traces and task-specific actions. Rather than maintaining a fixed plan, this synergy allows the model to perform dynamic reasoning to create, maintain, and adjust high-level action plans based on external observations. AdaPlanner (Sun et al., 2023) advances this concept by utilizing an explicit closed-loop system where an LLM acts as both a planner and a refiner. AdaPlanner proactively revises the entire future plan when environmental observations deviate from predictions. Furthermore, its "refine-then-resume" mechanism enables the agent to continue from an intermediate breakpoint rather than restarting an episode from scratch. Similar loops are critical for LLMs governing open-world and robotics planning, enabling agents to adjust trajectories based on environmental feedback to achieve long-horizon goals (Wang et al., 2023d; Huang et al., 2023b; Yang et al., 2024d).

Certain search methods can be interpreted as closed-loop planning systems by treating reasoning as a heuristic search over a state space. Works like Tree-of-Thoughts (Yao et al., 2023b) and Graph-of-Thoughts (Besta et al., 2024) allow the model to look ahead by generating multiple potential next steps and backtrack to previous stages if a current reasoning path is deemed unlikely to succeed. This enables the model to treat reasoning as a dynamic optimization problem, pruning suboptimal paths to converge on optimal action sequences that satisfy complex, long-horizon objectives. These methods transition the LLM from a simple generator to a deliberative planner that navigates a global search space to solve constrained long-horizon tasks.

### 5.3.3 Collaborative Reasoning

Collaborative reasoning extends beyond the autonomy of a single model by distributing the reasoning process across an ensemble of interacting agents. Through structured interaction protocols and role-based divisions, agents iteratively build upon and refine each other's reasoning traces, acting as a cohesive reasoning unit.

**Role Specialization.** Role-based architectures decompose complex reasoning tasks into functional hierarchies or distributed role structures where agents adopt distinct personas. This specialization mimics human organizational structures to manage cognitive load and improve precision. Hierarchical frameworks like MetaGPT (Hong et al., 2024) encode human-like workflows through an assembly-line paradigm where agents are assigned specialized roles such as product managers, architects, and engineers. This decomposes complex objectives into sequences of specialized interdependent subtasks, which effectively reduces cognitive load for individual agents and mitigates the risks of cascading hallucinations. CAMEL (Li et al., 2023a) and AutoGen (Wu et al., 2024a) take a more decentralized approach and use role specialization as an extension of the reasoning process itself to perform iterative refinement through a continuous dialogue loop, rather than relying on a predefined functional hierarchy. This distinction highlights how role specialization can either serve as an organizational scaffold for structured task execution or as an interactive mechanism that shapes collaborative reasoning dynamics through iterative agent communication.

**Debate.** The Multi-Agent Debate (MAD) framework facilitates structured dialogue among LLMs, directly enabling them to iteratively refine and update their responses. The framework, introduced by Du et al. (2024), replaces the self-reflection loop with a structured debate where agents generate their own solutions and then update their own answers based on the solutions of their peers. To promote diversity of candidates and reduce overcommitment in initial, potentially incorrect solutions, Reconcile (Chen et al., 2024c) utilizes diverse models and Liang et al. (2024) assigns different agent roles to force consideration of alternative viewpoints. While MAD strategies enhance logical depth, they introduce significant resource challenges. Many strategies reduce the density of information flow in the communication network to improve the efficiency of debate. One approach groups agents and restricts the inter-group communication to a summarization of intra-group exchanges or their final viewpoints, while intra-group agents share full access to each other's reasoning (Liu et al., 2024f; Wang et al., 2024e). Li et al. (2024l) moves away from the standard fully-connected communication network to a sparse topology, significantly reducing inference costs, while achieving comparable or superior accuracy compared to fully-connected networks.

### 5.4 Agentic System Improvement

The previous sections focused on inference-time improvement with respect to the model itself, examining how test-time compute can enhance generation by modifying decoding procedures and structuring internal reasoning processes, while keeping the surrounding execution environment fixed. In contrast, this section considers inference-time improvement at the system level. Agentic systems augment LLMs with persistent state, decision-making loops, external tools, and inter-agent coordination, forming a broader computational substrate in which the model operates. By dynamically adapting prompts, memory structures, tool libraries, communication topology, and workflows during inference, agents can reconfigure the environment that shapes model behavior. This enables test-time self-improvement not by altering the model's internal generation dynamics, but by evolving the system that governs how and where reasoning occurs, yielding greater adaptability, robustness, and task-specific capability without parameter updates.

#### 5.4.1 Prompts

Prompt optimization allows agents to dynamically modify primary and meta-prompts to improve reasoning, planning, and task performance. Early sampling-based approaches utilize best-of-n to select top-performing candidate prompts scored by a trained reward/preference model (Sun et al., 2024b) or execution accuracy on validation examples (Zhou et al., 2023c), while other approaches directly train a model for prompt generation or refinement (Deng et al., 2022; Yang et al., 2024a; Yao et al., 2024b). This optimization process can be extended to the selection of in-context demonstrations, where models dynamically identify the most relevant or informative examples to include in the prompt to maximize task accuracy (Wan et al., 2023a;b), as well as gradient based prompt segment optimizations such as in GSPO (Li et al., 2024f).

Search-based approaches, including PromptAgent (Wang et al., 2024f) and MCTS-OPS (Yu et al., 2025b), formulate prompt optimization as a sequential decision problem and employ Monte Carlo Tree Search guided by self-reflective signals and task-specific performance feedback to iteratively refine prompts before performing task-specific inference. Evolutionary-based frameworks like EvoPrompt (Guo et al., 2024a), AlphaEvolve (Novikov et al., 2025) and Promptbreeder (Fernando et al., 2024) employ conventional genetic algorithms like mutation and crossover over a population of archived prompts to guide search. Promptbreeder and AlphaEvolve extend evolution to the mutation-prompts (meta-prompts) which instruct how to modify primary prompts within an evolving population provided to the language model. This improves the instructions used for reflective prompt generation, enabling the discovery of more effective prompting strategies.

Prompt optimization methods can employ iterative improvement loops that leverage natural-language critiques to refine agent instructions, a process often described as textual gradient descent. ProTeGi (Pryzant et al., 2023), TextGrad (Yuksekgonul et al., 2025) and LLM-AutoDiff (Yin & Wang, 2025) employ this strategy as an optimization phase before deploying the prompt. ProRefine (Pandita et al., 2025) extends textual-gradients directly to inference-time by dynamically refining prompts to adapt to each generation step with critiques from a feedback LLM, allowing the model to adapt to evolving task requirements at inference time.

Within multi-agent systems, prompts can provide agents with distinct roles that define their responsibilities and enhance test-time performance through explicit decomposition of reasoning processes into complementary functions. Early works implement static role-playing through prompting (Qian et al., 2024; Li et al., 2023a; Hong et al., 2024; Wu et al., 2024a), but recent works, including MASS (Zhou et al., 2026a), Promptomatix (Murthy et al., 2025) and EvoAgent (Yuan et al., 2025) extend the idea of automatic prompt optimization across multi-module or multi-agent workflows to optimize agent roles and communication protocols. These systems revise system prompts across interacting components while adapting the agent workflow as a whole, enabling coordinated specialization and adaptation at the task or query level.

#### 5.4.2 Memory

Memory is a core capability of agentic systems, enabling agents to adapt across interactions with their environment. Unlike standard LLMs, whose behavior is constrained to a fixed context window, agents maintain persistent state that extends beyond a single interaction, allowing them to accumulate and reuse

knowledge. Long-term memory stores reusable experiences, strategies, tools, and agent configurations that persist across sessions, supporting cross-task generalization and continual improvement. In contrast, working (short-term) memory captures transient context within a task to support intermediate reasoning and decision-making.

Designing effective memory systems therefore requires specifying both the representation of stored knowledge and the mechanisms used for retrieval, updating, and consolidation. Building on the paradigm of retrieval-augmented generation (RAG), recent agentic systems integrate retrieval directly into the reasoning loop, transforming it from a static lookup into a dynamic process. Rather than performing a single retrieval step, agents can iteratively query memory, refine search strategies, and write newly acquired information back into the memory store. These mechanisms enable agents to efficiently recall relevant information, avoid redundancy, and maintain coherent internal state across long-horizon interactions. In this section, we focus primarily on long-term memory structures and the methods used to optimize them, as these components are central to enabling persistent learning and adaptive behavior in agentic systems.

**Structure.** Long-term memory structures vary across agents and affect retrieval quality and performance. Many systems use vector-based memory with RAG-style retrieval for chatbot interaction to address the limitations posed by the LLM context window, such as MemGPT (Packer et al., 2024) and MemoryBank (Zhong et al., 2024b) which store semantic memories and NeuroCache (Safaya & Yuret, 2024) which stores prior model states. Relevant self-judged experiences can be retrieved through RAG to inform new interactions and integrate newly acquired knowledge back into memory, increasing the agent's capability over time (Ouyang et al., 2025). Memory management has started to incorporate more sophisticated abstraction and summarization for efficient memory management of acquired experiences. Mem0 (Chhikara et al., 2025), A-Mem (Xu et al., 2025b), and G-Mem (Zhang et al., 2025c) all update knowledge graph-based structures for relations among factual entities, aiding efficient retrieval over extended interactions. Another common structural trend is partitioning memory into different logical substrates, mimicking the complexity of human memory and preventing retrieval inefficiencies (Wu et al., 2025f). For example, HINDSIGHT (Latimer et al., 2025) distinguishes between facts, experiences and beliefs, and MIRIX (Wang & Chen, 2025) distinguishes between procedural, episodic and semantic memories. These structural approaches provide epistemic clarity by separating objective evidence from subjective beliefs, which prevents context dilution and ensures that agents can maintain stable, traceable reasoning styles as their internal beliefs evolve over time.

**Optimization.** Memory updates are triggered by new interactions or dialogue turns as new information is incorporated into the current short-term context. Mem0 (Chhikara et al., 2025) and Memory-R1 (Yan et al., 2026) utilize CRUD-style operations (Add, Update, Delete, Noop) by employing an LLM to determine the appropriate operation for integrating new experiences into long-term memory. A-Mem (Xu et al., 2025b) decides on specific actions for neighboring memories including strengthening, merging, or pruning nodes of its knowledge graph. HINDSIGHT (Latimer et al., 2025) similarly evolves its memory substrates by synthesizing fragmented facts and updating belief confidence scores as it ingests new information. Another direction of optimization focuses on managing growing memory stores. SAGE (Liang et al., 2025) and MemoryLLM (Wang et al., 2024h) both limit memory capacity through biological-style optimization inspired by the Ebbinghaus Forgetting Curve, which phases out underutilized knowledge. This forgetting mechanism is used in SAGE to move knowledge from short to long-term memory and in MemoryLLM to manage a fixed capacity memory-pool. Mem1 (Zhou et al., 2026d) and MemAgent (Yu et al., 2026) leverage an RL-learned process of consolidation and pruning of their respective internal states and fixed-length memories to avoid unbounded context growth and keep memory usage constant throughout long-horizon tasks. The ALMA (Xiong et al., 2026) framework moves beyond human-crafted memory modules and uses a meta-agent in an offline phase to automatically discover and implement memory designs and protocols as executable code, bridging the gap between architectural designs and functional performance at test-time. These methods represent a paradigm shift from treating memory as a static knowledge buffer to a dynamic, evolving system state. By formalizing memory management as an optimization problem, agents maintain high retrieval precision and semantic coherence without exceeding constraints on the context window.

### 5.4.3 Tooling

Tool use is a core capability of LLM agents, enabling interaction with external APIs, databases, and environments beyond the model's parametric knowledge. While standard LLMs are restricted to static tool-calling, agents can be instilled with the ability to dynamically adapt and refine tool usage strategies, including selecting appropriate tools, determining optimal invocation, and adjusting tooling parameters based on task context and execution feedback. Inference-time tool creation, evolution, and usage is inherently connected to agentic planning and reasoning as tools can be retrieved or created on the spot by planners, called directly from agentic reasoning, or executed in workflows (Yao et al., 2023c; Wu et al., 2024b). There is a large body of work that applies training-time approaches that learn optimal tool invocation through SFT (Schick et al., 2023; Tang et al., 2023; Qin et al., 2024; Yang et al., 2023a) and RL-based (Feng et al., 2025a; Qian et al., 2025; Li et al., 2025f; Chai et al., 2025) fine-tuning on tool-calling trajectories, or build a starting set of general purpose tools for retrieval (Yuan et al., 2024a; Qiu et al., 2025a). These methods support accurate inference-time tool invocation, but we focus on post-deployment optimizations and evolution in this section.

**Creation and Refinement.**  Unlike traditional methods that rely on a static tooling library, frameworks increasingly empower LLMs to design and verify applicable tooling in the form of code and documentation at inference time (Qian et al., 2023; Wang et al., 2024b; Cai et al., 2024b; Shen et al., 2026). These systems trigger a tool-making phase when capability gaps are identified in the existing library and often implement iterative refinement processes to test tooling quality and ensure its reliability. These frameworks archive their new tools, enabling reuse and evolution across tasks. While earlier frameworks store complex, task-specific tooling implementations, TTE (Test-Time Tool Evolution) (Lu et al., 2026) decomposes synthesized code into modular units to increase the probability of reuse. Frameworks can also leverage external knowledge sources to support tool construction. Alita (Qiu et al., 2025b) employs a web search–driven agent to retrieve relevant open-source implementations, which are incorporated into on-the-fly tool generation and AutoAgent (Tang et al., 2025a) adopts RAG to query external code repositories, grounding the synthesis of reusable tooling components in retrieved artifacts.

Another axis of tooling improvement at inference time focuses on tooling refinement rather than the generation of new tooling code. Alita-G (Qiu et al., 2025a) refines tools' applications by adapting configurable parameters to new task requirements, allowing tools to be repurposed for new tasks. UCT (Shen et al., 2026) and ToolLibGen (Yue et al., 2025) employ post-inference processes enabling agents to systematically refine, merge and prune their tool library to ensure it remains scalable and avoids redundancy as the library grows. Other approaches explore documentation optimization within the inference prompt, which present tools in concise and structured formats, helping the model to invoke the correct tools more readily. PLAY2PROMPT (Fang et al., 2025b) iteratively refines documentation and usage examples from execution feedback and self-reflection, employing a search-based tool-play process that mimics human trial-and-error to discover tool behaviors without requiring human-labeled data.

**Selection.**  Classic search-based approaches such as ToolLLM (Qin et al., 2024) and ToolChain (Zhuang et al., 2024) leverage tree-based search over API spaces, overcoming limitations of traditional reasoning methods like in ReAct (Yao et al., 2023c) by enabling multiple simultaneous reasoning traces and easy error correction via backtracking. Building on tree-based approaches, AutoTool (Jia & Li, 2025) transforms open-ended tooling decisions into a constrained graph search that identifies predictable sequential patterns, allowing the system to bypass costly LLM calls and directly select the next tool based on historical trends and contextual relevance. Agents can further streamline the retrieval phase by dynamically updating prompts with the most relevant tools and historical use cases (Gan & Sun, 2025; Qiu et al., 2025a). This ensures the agent operates with a task-specific toolkit and prevents prompt bloat when dealing with vast sets of tools. Another line of research investigates a generative paradigm in which tools are integrated directly into the model's architecture as learned tool tokens, allowing the LLM to trigger external functions as naturally as it generates text (Hao et al., 2023; Wang et al., 2025d). Building on this idea, Chain-of-Tools (Wu et al., 2025a) leverages the hidden states of frozen language models to perform tool selection and evaluation during the reasoning process by analyzing the model's latent representations to determine when and which tool to invoke, ensuring efficient tool usage without compromising the model's reasoning capabilities. Overall, inference-time tool optimization transforms tool usage into a self-expanding procedural ecosystem in which

agents can bridge the gaps in their tooling capabilities to solve complex and previously unseen real-world problems.

### 5.4.4 Workflow and System Evolution

Workflows define the coordination patterns that govern inter-agent communication protocols and interactions with tooling and memory. They act as a blueprint designed to support the agents' reasoning abilities which enable the system to dynamically decompose tasks, call external tools and interpret feedback. Multi-agent systems (MAS) such as MAD (Du et al., 2024), or manually designed role-based systems like MetaGPT (Hong et al., 2024) assign static roles or structured communication structures which lack adaptiveness in task routing or feedback integration. A theme across more recent work involves automating the agent design process, moving beyond constrained, human-crafted systems. These methods dynamically evolve topology or the whole system architecture including agent configurations and tooling strategies. This allows the system itself to evolve on a per-task or per-query level that dictates the system behavior at test-time. This represents a shift from single-agent reasoning-based planning to optimization of system architecture to support decision making and cooperation between interacting components.

**Topology Evolution.** Agentic systems dynamically modify their communication structures either pre-deployment or during inference to tailor reasoning structures to task or query level instances. GPTSwarm (Zhuge et al., 2024) explicitly models multi-agent collaboration as an optimizable graph structure, where agent connectivity is dynamically adjusted based on real-time task accuracy from unit tests. The system also employs an LLM ranker, which deactivates low-performing agents from the topology. Approaches commonly use trained communication topology generator models to instantiate task-specific graphs that balance performance with computational efficiency. AMAS (Leong et al., 2025) introduces a dynamic graph selector, which identifies the optimal task-specific graph configuration from a candidate ensemble. AGP (Li et al., 2025a) leverages a Graph Neural Network for topology generation through a pruning process of first selecting active nodes, and then determining communication graph edges which define the direction of information flow. Rather than starting from a fixed topology and relying on pruning, ARG-Designer (Li et al., 2026b) generates a collaboration graph from scratch with an autoregressive graph generator.

Another line of work explores adaptive routing, which treats multi-agent collaboration as a sequential decision problem that yields an implicit communication graph tailored to the evolving task state. Routing-based approaches were originally used in a more static manner to select the expert that is best suited to solve a query (Shnitzer et al., 2023; Lu et al., 2024d), but in multi-agent systems this approach can be used to dynamically determine the communication structure in real-time. DyLAN (Liu et al., 2024h) models agent collaboration as a multi-layer temporal feed-forward network (T-FFN) where each layer's active nodes constitute the task-specific agent team for that time step. Agent activations (responses) are propagated forward, while a dynamic communication structure is maintained by an LLM ranker that prunes edges to low-performing agents. The Puppeteer framework (Dang et al., 2025) relies on an RL-trained orchestrator to determine the next agent to activate at each step in response to evolving task states in real-time. AgentNet (Yang et al., 2025a) removes the central orchestrator and relies on a decentralized approach where each agent decides how to route each task resulting in an implicitly constructed communication directed acyclic graph. These approaches allow systems to dynamically reconfigure or simplify their collaboration structures in response to performance signals, maintaining strong task accuracy while mitigating the computational overhead typically associated with multi-agent deployments.

**Workflow Evolution.** Many works perform whole-system evolution, optimizing the entire agent configuration, allowing for holistic adaptation. Popular works, including MASS (Zhou et al., 2026a), ADAS (Hu et al., 2025c), AFlow (Zhang et al., 2025e), AgentSquare (Shang et al., 2025) and The Darwin Gödel Machine (Zhang et al., 2026a) exemplify the trend of unified architecture evolution in an optimization phase prior to inference, which consists of evolutionary searches over populations of archived workflows or a search over the space of agent programs. While these methods employ an offline search process that is more computationally efficient than inference-time optimization, they generally do not allow for self-evolution during inference. If the task distribution changes substantially, a new optimization phase is required to re-optimize the system for the next generation of agents. HyperAgents (Zhang et al., 2026b) extend the Darwin Gödel

Machine along a complementary axis, noting that its self-modification mechanism itself remains fixed and handcrafted. By merging the task agent and the meta agent into a single self-modifiable program called a hyperagent, the system improves not only its task performance but also its own ability to improve itself, in principle across any computable task. Inference-time MAS optimization shares similar optimization strategies to offline methods but allows for MAS generation and/or inference-time self-evolution on a per-query basis.

Recent advances in test-time MAS optimization rely on meta-agents which reason over agent configurations and workflows to enhance the system capability rather than directly solving the task. These components are typically trained through SFT on query workflow pairs or objectives that reward downstream task performance. FlowReasoner (Gao et al., 2025b) and ScoreFlow (Wang et al., 2025g) train a reasoning-based meta-agent via RL to construct a query-specific MAS in a single pass. Similarly, MAS-GPT (Ye et al., 2025b) utilizes an open-source LLM trained with SFT on query-MAS pairs to generate a query-specific MAS represented as python code. Since these methods generate systems in a single shot, they address the high costs of existing adaptive methods which involve model calls at each intermediate step to adaptively determine workflows. In contrast to single pass methods, MAS-ZERO (Ke et al., 2025) introduces a training-free meta-agent to generate and iteratively refine agent configurations based on the solvability and completeness of the system's output. This balances efficiency with adaptive, inference-specific optimization, enabling the system to correct agentic configurations on the fly.

Another direction of work focuses on dynamic adaptation during the task-solving process or after the completion of task execution. EvoMAC (Hu et al., 2025d) and ANN (Ma et al., 2025b) dynamically evolve agent prompts, communication topology and workflows through an iterative textual backpropagation process with environmental feedback from agent execution (compiler logs, unit tests). EvoAgent (Yuan et al., 2025) automatically extends a single specialized agent into a collaborative system on a per-query basis and evolves system variables and agent settings such as roles and skills through evolutionary operators. EvoAgentX (Wang et al., 2025f) extends these concepts to holistic workflow evolution and integrates multiple state-of-the-art optimization strategies for system refinement including TextGrad for textual backpropagation refinement and reusable modular operators, which can be iteratively evolved and recombined to construct new workflows (Zhang et al., 2025e; Shang et al., 2025). In these approaches, evolution occurs as an iterative process after execution to select the best configuration moving forward. Collectively, these advancements represent a transition from rigid, hand-designed pipelines to fluid, self-organizing architectures that utilize meta-reasoning and environmental feedback to autonomously self-refine. A summary of agentic system evolution papers can be found in Table 11.

### 5.5 Test-Time Training

Test-Time Training (TTT) (Sun et al., 2020) represents a promising paradigm shift from static inference to dynamic, gradient-based self-improvement. TTT allows LLMs to adapt to instance-specific challenges on the fly by performing temporary, task-conditioned updates to their parameters at inference time, complementing or going beyond in-context adaptation by encoding instance-specific information directly into the model weights. This blurs the line between model optimization and inference as it allows the model to perform task-conditioned parameter updates during deployment.

**TT-SFT.** Test-Time Supervised Fine-Tuning (TT-SFT) involves updating model parameters during inference using a supervised loss derived from instance-specific data. Early works focus on improving few-shot generalization by using retrieved examples for temporary task-specific parameter updates during inference using a loss derived from a portion of the samples (Akyürek et al., 2025; Hardt & Sun, 2024). This encodes structural patterns demonstrated in context within the LLM's parameters, drastically improving accuracy on structurally novel tasks where standard in-context learning often fails. SFT-based TTT can also provide a memory mechanism for temporarily storing context in multi-step problems through parameter updates. Feedback-Based Test-Time Training (FTTT) (Li et al., 2025k) addresses the length generalization issue of iterative revision by storing past errors and experiences directly into the model weights, and TTT-E2E (Tandon et al., 2025) compresses context from long-horizon tasks directly into the model's weights via next-token prediction, achieving better performance while maintaining the constant inference latency of an RNN.

Table 11: **Overview of agentic system-based improvement approaches.** Methods are compared across prompting, memory, tool use, and workflow dimensions, with check marks (✓) indicating the presence of each capability.

| Method | Prompts | Memory | Tools | | Workflow | |
| --- | --- | --- | --- | --- | --- | --- |
| | | | Create/Refine | Selection | Topology | Unified |
| RLPrompt (Deng et al., 2022) | ✓ | – | – | – | – | – |
| APE (Zhou et al., 2023c) | ✓ | – | – | – | – | – |
| ProTeGi (Pryzant et al., 2023) | ✓ | – | – | – | – | – |
| CREATOR (Qian et al., 2023) | – | – | ✓ | – | – | – |
| ReAct (Yao et al., 2023c) | – | – | – | ✓ | – | – |
| ToolkenGPT (Hao et al., 2023) | – | – | – | ✓ | – | – |
| MemGPT (Packer et al., 2024) | – | ✓ | – | – | – | – |
| Prompt-OIRL (Sun et al., 2024b) | ✓ | – | – | – | – | – |
| OPRO (Yang et al., 2024a) | ✓ | – | – | – | – | – |
| Voyager (Wang et al., 2024b) | ✓ | ✓ | ✓ | ✓ | ✓ | ✓ |
| CRAFT (Yuan et al., 2024a) | – | – | ✓ | – | – | – |
| Retroformer (Yao et al., 2024b) | ✓ | – | – | – | – | – |
| PromptAgent (Wang et al., 2024f) | ✓ | – | – | – | – | – |
| EvoPrompt (Guo et al., 2024a) | ✓ | – | – | – | – | – |
| Promptbreeder (Fernando et al., 2024) | ✓ | – | – | – | – | – |
| LATM (Cai et al., 2024b) | – | – | ✓ | – | – | – |
| MemoryBank (Zhong et al., 2024b) | – | ✓ | – | – | – | – |
| NeuroCache (Safaya & Yuret, 2024) | – | ✓ | – | – | – | – |
| MemoryLLM (Wang et al., 2024h) | – | ✓ | – | – | – | – |
| GPTSwarm (Zhuge et al., 2024) | ✓ | – | – | – | ✓ | – |
| DyLAN (Liu et al., 2024h) | – | – | – | – | ✓ | – |
| ToolChain (Zhuang et al., 2024) | – | – | – | ✓ | – | – |
| MCTS-OPS (Yu et al., 2025b) | ✓ | – | – | – | – | – |
| TextGrad (Yuksekgonul et al., 2025) | ✓ | – | ✓ | – | – | – |
| LLM-AutoDiff (Yin & Wang, 2025) | ✓ | – | – | – | – | – |
| ProRefine (Pandita et al., 2025) | ✓ | – | – | – | – | – |
| Alita-G (Qiu et al., 2025a) | ✓ | – | ✓ | ✓ | – | – |
| Promptomatix (Murthy et al., 2025) | ✓ | – | – | – | – | – |
| EvoAgentX (Wang et al., 2025f) | ✓ | – | – | ✓ | ✓ | ✓ |
| A-Mem (Xu et al., 2025b) | – | ✓ | – | – | – | – |
| G-Mem (Zhang et al., 2025c) | – | ✓ | – | – | – | – |
| Mem0 (Chhikara et al., 2025) | – | ✓ | – | – | – | – |
| MIRIX (Wang & Chen, 2025) | – | ✓ | – | – | – | – |
| HINDSIGHT (Latimer et al., 2025) | – | ✓ | – | – | – | – |
| SAGE (Liang et al., 2025) | – | ✓ | – | – | – | – |
| Memory-R1 (Yan et al., 2026) | – | ✓ | – | – | – | – |
| Mem1 (Zhou et al., 2026d) | – | ✓ | – | – | – | – |
| MemAgent (Yu et al., 2026) | – | ✓ | – | – | – | – |
| Alita (Qiu et al., 2025b) | – | – | ✓ | – | – | – |
| AutoAgent (Tang et al., 2025a) | – | – | ✓ | – | – | – |
| AgentSquare (Shang et al., 2025) | ✓ | ✓ | – | ✓ | – | ✓ |
| PLAY2PROMPT (Fang et al., 2025b) | – | – | ✓ | – | – | – |
| ToolLibGen (Yue et al., 2025) | – | – | ✓ | – | – | – |
| AutoTool (Jia & Li, 2025) | – | – | – | ✓ | – | – |
| RagMCP (Gan & Sun, 2025) | – | – | – | ✓ | – | – |
| ToolGen (Wang et al., 2025d) | – | – | – | ✓ | – | – |
| Chain-of-Tools (Wu et al., 2025a) | – | – | – | ✓ | – | – |
| AMAS (Leong et al., 2025) | – | – | – | – | ✓ | – |

*Continued on next page*

*Table 11 continued from previous page*

| Method | Prompts | Memory | Tools | | Workflow | |
|---|---|---|---|---|---|---|
| | | | Create/Refine | Selection | Topology | Unified |
| AGP (Li et al., 2025a) | – | – | – | – | ✓ | – |
| Puppeteer (Dang et al., 2025) | – | – | – | – | ✓ | – |
| AgentNet (Yang et al., 2025a) | – | ✓ | – | – | ✓ | – |
| EvoAgent (Yuan et al., 2025) | ✓ | – | – | – | ✓ | ✓ |
| ADAS (Hu et al., 2025c) | ✓ | – | ✓ | – | ✓ | ✓ |
| AFlow (Zhang et al., 2025e) | ✓ | – | ✓ | ✓ | ✓ | ✓ |
| EvoMAC (Hu et al., 2025d) | ✓ | – | – | – | ✓ | ✓ |
| ANN (Ma et al., 2025b) | ✓ | – | – | – | ✓ | ✓ |
| FlowReasoner (Gao et al., 2025b) | ✓ | – | – | – | ✓ | ✓ |
| ScoreFlow (Wang et al., 2025g) | ✓ | – | – | – | ✓ | ✓ |
| MAS-GPT (Ye et al., 2025b) | ✓ | – | ✓ | – | ✓ | ✓ |
| MAS-ZERO (Ke et al., 2025) | ✓ | – | ✓ | – | ✓ | ✓ |
| ARG-Designer (Li et al., 2026b) | ✓ | – | – | – | ✓ | – |
| MASS (Zhou et al., 2026a) | ✓ | – | – | – | ✓ | ✓ |
| AlphaEvolve (Novikov et al., 2025) | ✓ | ✓ | ✓ | – | – | ✓ |
| TTE (Lu et al., 2026) | – | – | ✓ | ✓ | – | – |
| UCT (Shen et al., 2026) | – | – | ✓ | – | – | – |
| ALMA (Xiong et al., 2026) | – | ✓ | – | – | – | – |
| DGM (Zhang et al., 2026a) | ✓ | ✓ | ✓ | – | ✓ | ✓ |
| HyperAgents (Zhang et al., 2026b) | ✓ | ✓ | ✓ | – | ✓ | ✓ |

Another direction explores on the fly knowledge incorporation for unlabeled, out-of-distribution data. VDS-TTT (Moradi et al., 2025) scores and filters generated candidate solutions for a user's query, creating a high-confidence label for SFT updates on highly scored candidates. SEAL (Zweiger et al., 2025) incorporates new context by generating self-edits (in the form of implications of the text) and updates the weights so that the model internalizes this information. Self-edit generation is trained to improve the efficiency with which information is internalized by the model, enabling task-specific adaptation.

**TT-RL.** Zuo et al. (2025) introduces test-time reinforcement learning, a paradigm that performs full RL updates on unlabeled data at test-time. With the absence of explicit ground truth rewards, the TTRL framework generates multiple candidate outputs which are evaluated using a reward signal such as majority vote or a reward model score to perform the RL update. Other frameworks employ variations of majority voting: Du et al. (2025) implements spatial voting to identify consensus regions for GUI agents and Liu et al. (2025c) reshapes the advantage calculated from the majority reward signal to promote exploration and mitigate bias. LADDER (Simonds & Yoshiyama, 2025) generates and scores variants of an out-of-distribution test instance to perform a temporary and instance specific parameter update before the final test question is answered. TTRL surpasses the performance limits of the model's majority-vote accuracy and can approach the performance of offline training (Zuo et al., 2025), highlighting the effectiveness of unsupervised self-evolutionary training. Collectively, these approaches represent a shift from static inference to on the fly policy adaptations, allowing for alignment with novel patterns that exceed their original scope.

## 5.6 Discussion

### 5.6.1 Test-Time Scaling

A growing trend in LLM research focuses on test-time scaling (TTS), which enhances inference-time capabilities by allocating more compute at inference time. This paradigm shifts the emphasis from parameter scaling to compute scaling at inference. Many inference-time methods discussed earlier are instances of test-time scaling; parallel candidate generation, iterative improvement, use of external reward models, and employing agent ensembles all incur increased compute to enhance accuracy at inference time. Increased inference-time compute introduces new challenges, particularly in balancing performance gains against computational cost.

**Comparison of Techniques.** The efficacy of test-time scaling methods varies significantly across problem domains. Iterative refinement acts as a form of local search and tends to be most compute-efficient for simpler problems where initial outputs require only minor logical adjustments, particularly for non-reasoning models. Harder problems benefit more from search guided by a verifier. However, in domains lacking automatic verifiers, scaling through majority voting or best-of-n exhibits diminishing returns as sample sizes grow, since these strategies cannot reliably identify rare but correct outputs (Brown et al., 2024; Snell et al., 2025). An alternative approach is to increase compute within a single reasoning trajectory rather than across many independent samples. DeepSeek-R1 (Guo et al., 2025) shows that extending reasoning chains at inference can outperform naive multi-sample approaches while remaining more token-efficient. This allows the model to adaptively allocate computation to backtracking, verification, and exploration of alternative logical paths within a single response.

**TTS as a Substitute for Post-Training.** Test-time compute can often substitute for post-training, especially for problems of easier difficulty (Snell et al., 2025). Weaker models can achieve superior reasoning and generation quality through the strategic use of test-time scaling (Brown et al., 2024). This makes test-time scaling an attractive, often cost-effective alternative to training larger models, but its benefits depend heavily on the model's ability to accurately assess or verify its own outputs. Additionally, the use of reward models at the token or sequence level for preference alignment is a form of TTS which can be used in place of specified post-training methods such as RLHF to explicitly instill aligned behavior during inference (Khanov et al., 2024); however, these benefits are often limited relative to the computational overhead and complexity introduced during large-scale reinforcement learning.

**TTS Compute Optimization.** Self-improvement methods like iterative revision and search should adopt compute-optimal strategies for adaptive compute allocation including early stopping conditions for sampling methods (Li et al., 2024h), which adaptively halt the sampling process once the model's confidence surpasses a threshold, speculative rejection which terminates low-performing responses early in the generation (Sun et al., 2024a), or by scaling token budget or compute allocation based on the difficulty of the task (Guo et al., 2025; Zelikman et al., 2024). In multi-agent systems, architectures can prune underperforming agents, simplify workflows for easier tasks, and restrict communication. This shifts the objective beyond raw accuracy toward performance relative to inference-time compute, encouraging more resource-efficient agent designs.

### 5.6.2 Agentic Trends

**Evolutionary Agentic Design.** A key trend in agentic systems is modularization across reasoning, tooling, and coordination functions, where cognitive responsibilities are separated into interacting components rather than handled by a single monolithic model. Reasoning is often decomposed into distinct modules for planning, execution, verification, and reflection, allowing targeted improvements and selective upgrading of weak stages. Tool use is similarly modularized, with dedicated components for tool selection, argument construction, execution, and result interpretation, making it easier to swap, refine, or extend capabilities as new tools or environments emerge (Shang et al., 2025). At a higher level, multi-agent workflows apply the same principle by assigning specialized roles to different agents, turning system design into the orchestration of interoperable functional units (Jung et al., 2025). These modules can be archived, reused, and iteratively modified, enabling systems to accumulate capabilities over time and recombine prior components to address previously unseen tasks. This modular and evolvable structure supports continual system growth without requiring end-to-end redesign, making agentic architectures increasingly adaptive and extensible.

Another emerging direction moves beyond fixed meta-agent architectures by enabling agents to directly modify and evolve their own implementations, including their codebases, prompting strategies, and decision policies (Zhang et al., 2026a; Xiong et al., 2026). While these approaches typically perform evolution in an offline optimization phase, they produce adaptive system designs that are instantiated at inference time on a per-task or per-query basis. This paradigm enables recursive self-improvement, where the system learns to optimize not only task performance but also the mechanisms by which it evolves.

**Query-Specific System Adaptations.** A key trend is the shift from pre-deployment agentic optimization to inference-time system optimization, where workflows and system components are no longer fixed in advance

but adapt their reasoning structure on a per-task basis during execution. Methods such as dynamic routing, topology evolution, meta-agent–based system generation, and adaptive tool-selection policies enable the system to revise which agents are active, how they communicate, and which tools are invoked in response to intermediate outcomes. This shift is evident especially in the generation of multi-agent systems, where instead of pre-deployment system optimization, works are generating task-specific systems at inference which also have the ability to adapt to test-time feedback (Ke et al., 2025). This represents a move away from heavily pre-structured, manually engineered pipelines toward flexible architectures in which system organization itself becomes an object of optimization at inference time.

Test-time scaling and agentic evolution reflect a broader shift from static, one-shot inference towards dynamic systems that structure test-time compute, reasoning, and system architecture in response to task demands. Together, these trends point toward a future where intelligence is shaped by how effectively agents can dynamically use system meta-learning to support complex reasoning in new tasks.

### 5.6.3 Synergy with the Optimized Model

Inference refinement operates directly on the optimized model produced by the previous stage: it extracts more from that model without touching its parameters, so its relationship to the optimized model is what defines it. The optimized model is the substrate on which refinement operates, and what refinement can achieve is bounded by what optimization has already written into the parameters. A stronger optimized model raises the quality of every candidate that sampling or search draws from its distribution, so the same test-time budget reaches further; it also plans, reflects, and evaluates and revises its own outputs more reliably—capabilities that are themselves products of optimization and that a weakly optimized model lacks— which in agentic settings directly determines how well test-time loops carry out multi-step tasks. Given a fixed optimized model, allocating more inference-time compute to these loops, through longer search, more tool calls, or additional rounds of self-correction, keeps improving performance, a form of test-time scaling that adds capability on top of the optimized model without further training. The relationship also runs in the other direction: optimization can absorb the gains of refinement back into the parameters, so that answers once reachable only through extensive search are produced by the model in a single pass, and the refined trajectories become high-quality experiences that data acquisition and selection feed into the next round of optimization. Test-time performance is thus set jointly by the two stages: optimization raises the ceiling of the model and refinement realizes it, while the outputs of refinement flow back to raise the model itself.

## 6 Autonomous Evaluation

### 6.1 Overview

As discussed in the previous sections, self-improving language models can enhance their capabilities through multiple pathways: acquiring new data, selecting or filtering training signals, and refining inference-time strategies. Each of these mechanisms aims to produce measurable performance gains. However, without a robust evaluation framework, it becomes unclear whether observed improvements reflect genuine capability growth, transient overfitting, or exploitation of feedback signals. Evaluation is therefore not merely a final reporting step, but a central component that determines whether data acquisition, data selection, model optimization, and inference refinement truly lead to sustainable self-improvement.

When a model continuously updates its parameters, modifies prompts, or adapts its inference policies, a fixed benchmark can quickly become obsolete. Performance gains may reflect overfitting to known test distributions rather than broader competence. In this sense, static evaluation fails to provide timely and reliable feedback for systems that evolve over time.

At the same time, externally administered and human-driven evaluation introduces scalability bottlenecks. Collecting new benchmarks requires substantial expert labor, careful curation, and continuous updating. As self-improving systems operate at increasing speed and scale, human evaluation cannot keep pace. This mismatch creates a structural limitation: improvement mechanisms can run autonomously, but evaluation remains slow and costly.

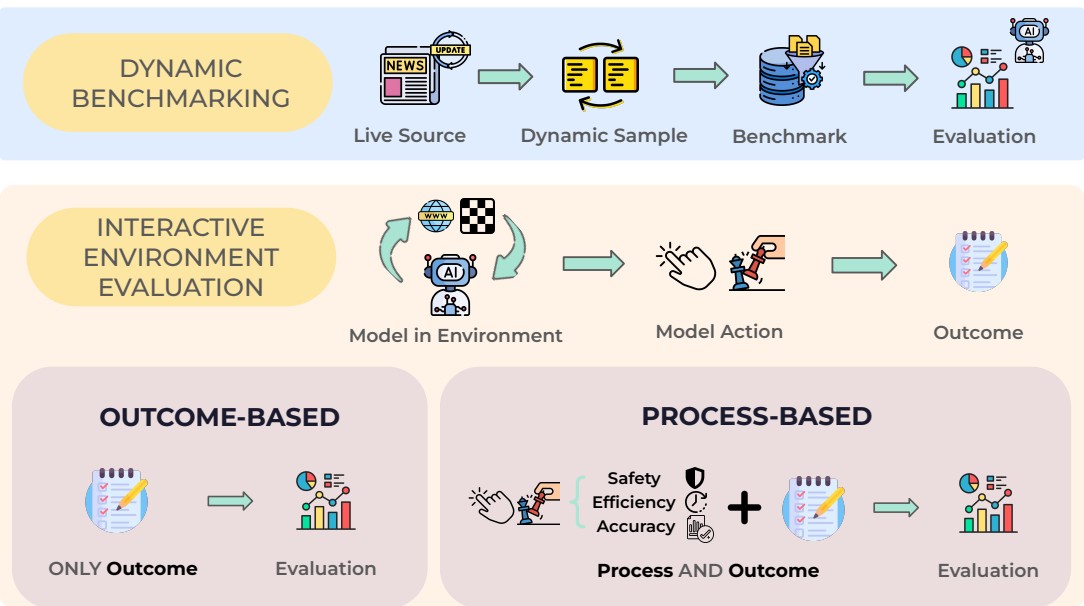

Figure 8: **Overview of autonomous evaluation in the self-improvement system.** This stage focuses on how the model evaluates its performance and obtains feedback signals to support continuous self-improvement. **(i) Dynamic Benchmarking:** The evaluation set is continuously updated using data from evolving or real-world sources, allowing benchmarks to adapt to distribution shifts and emerging tasks. **(ii) Interactive Environment Evaluation:** The model's capabilities are assessed through interactions with an environment, where performance is measured based on task completion, feedback signals, or environment-derived rewards.

These twin pressures, the inadequacy of static testing and the expense of human-driven assessment, motivate the need for autonomous evaluation. We use autonomous evaluation as a term for evaluation protocols that remain informative under two pressures that increasingly break static testing: (i) temporal drift in data and task distributions, including the emergence of new knowledge and changing user needs; and (ii) adaptive pressure from increasingly agentic or self-improving systems that can exploit, overfit to, or otherwise render fixed test sets obsolete. Autonomous evaluation treats assessment not as a one-time measurement on a frozen dataset, but as an evolving procedure designed to preserve validity as models and environments change.

The need for autonomous evaluation is especially acute for self-improving models. Unlike static models that are evaluated once after training, self-improving systems may iteratively update parameters, modify inference strategies, adjust prompts, or revise internal policies based on feedback. Under continual adaptation, a fixed benchmark can quickly become an unreliable target: it may be overfit through exploitation via superficial shortcuts, or simply become stale as the system's capabilities and behaviors shift. An effective evaluation protocol should adapt over time, resist exploitation, and remain sensitive to meaningful changes in competence.

In this section, as shown in Figure 8, we organize autonomous evaluation methods into two paradigms: dynamic benchmarking (§6.2) and interactive environment evaluation (§6.3). Dynamic benchmarking preserves the benchmark abstraction but continuously refreshes or transforms evaluation instances to mitigate contamination. Interactive environment evaluation instead embeds the model within an interactive, stateful system, assessing performance over execution trajectories where actions shape future states and outcomes. Together, these paradigms provide a foundation for evaluating self-improving models when static, one-shot testing is no longer sufficient. Finally, §6.4 concludes with a discussion on autonomous evaluation in guiding long-term self-improvement.

Table 12: **Overview of dynamic evaluation benchmarks.** Each benchmark name is hyperlinked to its official repository or project website. Benchmarks are compared by their update mode (live vs. static) and their data origin (Wikipedia, arXiv, code repositories, and news).

| Benchmark | Update Mode | Data Origin |
|---|---|---|
| REALTIME QA (Kasai et al., 2023) | Live | News |
| KoLA (Yu et al., 2024b) | Live | Wiki, ArXiv |
| EvoCodeBench (Li et al., 2024c) | Live | Git/Code |
| FRESHQA (Vu et al., 2024) | Live | Web |
| AntiLeakBench (Wu et al., 2025e) | Live | Wiki |
| DeepScholar-Bench (Patel et al., 2025) | Live | ArXiv |
| LiveCodeBench (Jain et al., 2025) | Live | Git/Code |
| LIVEBENCH (White et al., 2025) | Live | News, ArXiv |
| AcademicEval (Zhang et al., 2025d) | Live | ArXiv |
| Dynamic-KGQA (Dammu et al., 2025) | Live | Wiki |
| DynaQuest (Lin et al., 2025a) | Live | Wiki |
| TDBench (Hou et al., 2025) | Live | Wiki |
| Daily Oracle (Dai et al., 2025) | Live | News |
| OKBench (Li et al., 2025j) | Live | News |
| DyCodeEval (Chen et al., 2025d) | Static | Git/Code |

## 6.2 Dynamic Benchmarking

Dynamic benchmarking is a natural response to the limitations of static evaluation under distribution shift and model adaptivity. As models evolve through iterative training, test-time adaptation, or self-improvement loops, a fixed benchmark can become progressively less informative: it may be saturated or be indirectly overfit through repeated exposure and feedback. Dynamic benchmarks address this by evolving the evaluation instances themselves while maintaining comparability at the level of the benchmark objective. A summary of dynamic benchmarking papers can be found in Table 12.

Early examples of this paradigm include RealTimeQA (Kasai et al., 2023), which constructs questions from fresh news articles, requiring models to answer queries about recent events. By explicitly tying evaluation instances to publication timestamps, RealTimeQA demonstrated that strong performance on static benchmarks does not guarantee competence on up-to-date information. FreshQA (Vu et al., 2024) extends this line of work by curating a dynamic QA benchmark for current world knowledge that includes both fast-changing facts and false-premise questions, with ground-truth answers updated on a regular schedule. In contrast to benchmarks that focus only on factual recall, FreshQA also probes whether models can correctly reject outdated or invalid assumptions. AntiLeakBench (Wu et al., 2025e) formalized cutoff-aware evaluation by identifying new Wikidata triples that appeared after a predefined training cutoff and automatically generating corresponding question–answer pairs. This approach reduces the risk of hidden data leakage while enabling regular benchmark refreshes. Dynamic-KGQA (Dammu et al., 2025) evaluates temporal reasoning over evolving knowledge graphs. In this setting, models need to account for changes in entity relations rather than relying on static factual associations. To further scale benchmarking, DynaQuest (Lin et al., 2025a) automates question generation directly from Wikipedia revision histories. By exploiting structured edit logs, DynaQuest enables continuous benchmark updates with minimal human supervision, shifting evaluation maintenance from manual curation to procedural generation. Pushing this trend toward full automation, OKBench (Li et al., 2025j) proposes an on-demand framework that automatically constructs fresh factual QA benchmarks from daily news through a pipeline of information extraction, question generation, validation, and dataset versioning. This makes dynamic benchmark creation reproducible and decentralized, and

Table 13: **Overview of interactive environment evaluation environments.** Each environment is hyperlinked to its official repository or project website, and compared by evaluation type, task domain, and metrics across diverse interactive settings.

| Environment | Type | Task Domain | Evaluation Metrics |
| --- | --- | --- | --- |
| TextWorld (Côté et al., 2018) | Outcome | Games | Task success |
| Jericho (Hausknecht et al., 2020) | Outcome | Text / Games | Episode reward, game completion |
| ScienceWorld (Wang et al., 2022) | Outcome | Science | Task success, interaction reward |
| WebShop (Yao et al., 2023a) | Outcome | Web & Information | Task success, efficiency |
| WebArena (Zhou et al., 2023b) | Outcome | Web & Information | Task success, efficiency |
| GAIA (Mialon et al., 2023) | Outcome | General assistant | Task success |
| OSWorld (Xie et al., 2024) | Outcome | Software Engineering | Task success |
| WindowsAgentArena (Bonatti et al., 2024) | Outcome | Software Engineering | Task success |
| AgentGym (Xi et al., 2024a) | Outcome | Text / Games | Task success, cumulative reward |
| AppBench (Wang et al., 2024c) | Outcome | Software Engineering | Task success, efficiency |
| AppWorld (Trivedi et al., 2024) | Outcome | Software Engineering | Task success, efficiency |
| AndroidWorld (Rawles et al., 2024) | Outcome | Software Engineering | Task success, efficiency |
| TacticCraft (Ma et al., 2025a) | Outcome | Games | Game outcome |
| GAMEARENA (Hu et al., 2025a) | Outcome | Game | Game outcome |
| LMRL-Gym (Abdulhai et al., 2025) | Outcome | Text | Task success, cumulative reward |
| GAIA2 (Froger et al., 2026) | Outcome | General assistant | Task success, action verification |
| Lean-Gym (Polu et al., 2022) | Process | Formal Mathematics | Proof success, efficiency |
| SafeArena (Tur et al., 2025) | Process | Web & Information | Task success, safety violation rate |

provides a cleaner testbed for evaluating retrieval-augmented models on non-memorized data. A related but distinct direction is Daily Oracle (Dai et al., 2025), which also derives question-answer pairs from daily news, but uses them to evaluate forecasting and temporal generalization by asking models to predict future event outcomes rather than answer questions about already established facts.

More recent benchmarks expand dynamic evaluation beyond news and encyclopedic text. LIVEBENCH (White et al., 2025) aggregates questions from multiple live sources, including news outlets and newly published scientific papers. It provides a general-purpose and continuously refreshed evaluation suite, making it a reference point for evaluating LLMs that are equipped with retrieval or continual updating mechanisms.

Scientific knowledge has also emerged as a distinct focus. AcademicEval (Zhang et al., 2025d) targets newly released arXiv papers, testing whether models can reason over recent research contributions rather than merely retrieve surface-level summaries. DeepScholar-Bench (Patel et al., 2025) uses LLM-based pipelines to summarize, cross-link, and query newly uploaded scientific papers, emphasizing multi-hop reasoning over evolving scholarly content. In the programming domain, LiveCodeBench (Jain et al., 2025) evaluates models on continuously evolving code repositories, requiring understanding of newly introduced APIs, libraries, and software practices. EvoCodeBench (Li et al., 2024c) further explores code evolution as a dynamic evaluation signal, highlighting the importance of temporal generalization in software-oriented tasks.

### 6.3 Interactive Environment Evaluation

Dynamic benchmarks extend static evaluation by continuously regenerating or transforming task instances, mitigating data contamination and distributional staleness. Despite this adaptivity, they typically remain benchmark-based in structure: evaluation is still carried out over a collection of instances, and each instance is collected and scored independently. A more radical alternative is **Interactive Environment Evaluation**, which embeds the model within an interactive, stateful environment. In this paradigm, performance

is assessed over execution trajectories: the model repeatedly observes the environment, takes actions, and induces state transitions that shape subsequent observations and attainable outcomes. Importantly, interactive environment evaluation does not eliminate explicit objectives, but it changes how evidence is generated: correctness and competence are expressed through temporally extended interaction rather than isolated input&output pairs.

A practical criterion distinguishes interactive environment evaluation from benchmark-based evaluation: if a model's actions influence future states, observations, or rewards beyond a single test instance, evaluation is environment-based; otherwise, it remains benchmark-based. This criterion captures a deeper distinction in how evaluation signals arise. Benchmarks derive scores from predefined instances whose content is fixed at evaluation time, whereas interactive environment evaluation places the model inside a persistent system where outcomes depend on interaction dynamics and delayed consequences.

By introducing state, causality, and long-horizon dependencies, interactive environment evaluation addresses a core limitation of static testing: it can directly probe planning, credit assignment, recovery from partial failures, and behavioral consistency over time. These properties make it especially relevant for self-improving models, whose competence is often expressed through extended interaction and iterative refinement rather than single-turn answers.

Environment settings differ substantially in what aspects of behavior are rewarded or penalized. As shown in Table 13, we therefore distinguish **Outcome-Based** and **Process-Based** interactive environment evaluations based on the structure of their objectives: whether evaluation primarily collapses trajectories to terminal goal satisfaction, or whether it intentionally differentiates successful trajectories based on execution quality.

### 6.3.1 Outcome-Based Environment

Outcome-based interactive environment evaluation assesses model performance through terminal goal satisfaction within a stateful, interactive environment. In these settings, models interact with the environment over multiple steps, but evaluation is ultimately determined by whether a predefined objective is achieved, such as completing a task, solving a problem, or reaching a target state. A defining characteristic of this class is the absence of step-level ground truth for interaction trajectories. For many interactive tasks, there exist multiple valid ways to achieve the same goal, and it is often unclear which intermediate actions are preferable in isolation. As a result, evaluation cannot rely on comparing individual steps against a canonical reference trajectory. Instead, outcome-based environments collapse successful trajectories into a shared notion of success, using terminal outcomes as the primary evaluation signal. Intermediate actions are therefore treated as instrumental rather than evaluative.

This evaluation paradigm emphasizes capability acquisition. Outcome-based environments are well-suited for assessing whether models can operate effectively within interactive systems, plan over extended horizons, and recover from partial failures, while abstracting away finer-grained distinctions in how those capabilities are exercised.

Many early outcome-based environments arise from web-based interaction systems, which provide structured, stateful interfaces with well-defined success conditions. WebShop (Yao et al., 2023a) frames online shopping as an interactive environment in which a model navigates a simulated e-commerce website to identify and purchase products that satisfy user constraints. Evaluation is based on whether the correct product is ultimately purchased. WebArena (Zhou et al., 2023b) extends this paradigm to a broader set of realistic web tasks, including information retrieval, form submission, and multi-page navigation across simulated websites. Models interact with these environments through browser-like interfaces, and evaluation focuses on successful task completion within the web ecosystem. GAIA (Mialon et al., 2023) evaluates general AI assistants on real-world questions that are often easy for humans but require non-trivial combinations of reasoning, multimodal understanding, web browsing, tool use, and precise answer formatting. Although GAIA is ultimately scored by the final answer, successful systems typically need to choose and order intermediate tool calls correctly, making it a useful bridge between static QA-style benchmarks and trajectory-based agent evaluation. GAIA2 (Froger et al., 2026) moves closer to interactive environment evaluation by introducing dynamic and asynchronous scenarios in which events can evolve independently of the agent's actions. This

setting stresses temporal constraints, ambiguity resolution, noisy observations, collaboration, and write-action verification, and is therefore especially relevant for evaluating self-improving agents whose behavior unfolds over extended interaction trajectories.

Beyond web-based systems, several outcome-based environments operate in controlled, language-centric or symbolic settings that allow precise specification of state dynamics while retaining long-horizon interaction. LMRL-Gym (Abdulhai et al., 2025) introduces a suite of language-based reinforcement learning tasks, including dialogue games and text-based problem-solving scenarios. Models interact purely through natural language, and evaluation is based on episode-level task success or cumulative reward. TextWorld (Côté et al., 2018) provides procedurally generated text-based games in which models explore rooms, manipulate objects, and solve puzzles via textual commands. Evaluation is determined by game score or puzzle completion, largely independent of the specific exploration strategy used. ScienceWorld (Wang et al., 2022) extends to a simulated scientific environment in which models must perform multi-step experimentation and tool use to reach specified outcomes. Jericho (Hausknecht et al., 2020) similarly exposes classic interactive fiction games as executable environments with rich symbolic worlds and long-horizon dependencies, where evaluation is driven by in-game progress and completion.

Game-like environments also suit outcome-based environment evaluation, where success is usually defined by achieving a win condition. TacticCraft (Ma et al., 2025a) models turn-based tactical decision-making tasks in a synthetic game environment, evaluating models primarily by game outcome, such as win or loss. AgentGym (Xi et al., 2024a) aggregates a diverse collection of interactive environments spanning language tasks, games, and simulated systems. GameArena (Hu et al., 2025a) evaluates models through live or simulated gameplay, where actions update a persistent game state and outcomes depend on multi-step interaction with opponents and the environment. While these game environments are heterogeneous and intermediate decisions influence the final result, they are not directly assessed for quality.

A closely related and increasingly important subclass of outcome-based environments centers on application-oriented interaction. AppBench (Wang et al., 2024c), AppWorld (Trivedi et al., 2024), and Android-World (Rawles et al., 2024) evaluate models performing tasks within simulated or real application ecosystems, such as mobile apps or multi-application workflows. These environments require models to execute correct sequences of actions—navigating interfaces, filling forms, invoking functions, and managing persistent system state—to achieve task objectives. Incorrect intermediate actions often invalidate success, making procedural correctness essential for completion. However, because there is no unique ground-truth process for accomplishing these tasks, evaluation typically relies on whether the final application state satisfies the task requirements. As a result, different successful execution traces are not systematically distinguished by efficiency or execution style, placing these evaluations within the outcome-based category despite their rich interaction dynamics.

OSWorld (Xie et al., 2024) and WindowsAgentArena (Bonatti et al., 2024) extend outcome-based interactive environment evaluation to full computer-use settings. Both benchmarks embed models in realistic operating-system environments where actions update persistent system state and tasks require multi-step interaction across applications. Evaluation is implemented through execution-based checks that verify whether the desired goal state is reached (e.g., files created, settings changed, or workflows completed). While these benchmarks often report diagnostic statistics such as interaction length or common failure modes, model comparison is primarily organized around task completion.

Taken together, outcome-based interactive environment evaluations occupy an important middle ground between static benchmarks and more behavior-sensitive evaluation settings. By embedding models within persistent, interactive systems, they enable the assessment of planning, recovery, and long-horizon interaction. At the same time, by relying on terminal goal satisfaction in the absence of step-level ground truth, they prioritize measuring whether models can achieve objectives in complex environments rather than how those objectives are achieved. This makes outcome-based environments a natural and necessary foundation for evaluating self-improving models.

### 6.3.2 Process-Based Environment

Process-based interactive environment evaluation departs from outcome-based settings by explicitly evaluating how a model achieves its objective, rather than only whether the objective is achieved. In these environments, task success is necessary but not sufficient for strong performance. The evaluation objective is designed to assign different outcomes to distinct successful trajectories based on properties of the interaction process itself, such as safety, efficiency, correctness of intermediate actions, or strategic consistency over time.

A central distinction from outcome-based environments lies in the treatment of ground truth. While outcome-based settings lack step-level ground truth due to the existence of many equally valid ways to reach a goal, process-based environments intentionally define normative constraints or preferences over trajectories. These constraints may take the form of explicit penalties, structured rewards, or rule-based checks that encode which intermediate behaviors are acceptable, desirable, or unsafe. As a result, execution quality becomes observable to the evaluation signal. Two models that reach the same terminal state may therefore receive substantially different evaluations depending on how that state was reached.

A prominent class of process-based environments arises in settings where violations during execution are explicitly penalized, even when tasks are ultimately completed. SafeArena (Tur et al., 2025) extends web-based interaction environments with safety-sensitive evaluation criteria. Models perform multi-step web tasks under constraints related to harmful content, policy compliance, or unsafe actions. Crucially, unsafe intermediate behaviors incur penalties that directly affect evaluation outcomes.

Formal reasoning environments provide a complementary and particularly clear instantiation of process-based evaluation, where the structure of the reasoning process itself is central to assessment. Lean-Gym (Polu et al., 2022) formulates interactive theorem proving as an environment in which models apply tactics sequentially to transform a proof state. Evaluation is not limited to whether a theorem is ultimately proven; instead, properties such as proof length, validity of intermediate states, and the structure of tactic sequences directly influence performance measurement. Multiple successful proofs may therefore receive different evaluations based on their construction processes.

Generally, process-based interactive environment evaluations thus represent a qualitative shift from outcome-based assessment to behavior-sensitive evaluation. In practice, the boundary between outcome-based and process environments is often blurred. Many outcome-based environments are designed primarily around terminal goal satisfaction, yet include auxiliary signals, such as step limits and action costs, which make parts of the execution process observable. Conversely, fully process-based environments require the evaluator to specify and reliably measure trajectory-level properties (e.g., safety or adherence to procedural constraints) in a way that is robust across diverse strategies. This is substantially harder than checking goal completion: it often demands fine-grained instrumentation of environment state, careful definition of what constitutes a violation or inefficiency, and evaluation rules that generalize across multiple valid solution paths. As a result, the number of environments that are unambiguously process-based remains relatively small. A more common pattern is that outcome-based environments increasingly incorporate process monitoring to provide richer diagnostic signals, even when the main metric is still task success.

## 6.4 Discussion

### 6.4.1 Trends in Autonomous Evaluation

Across the evaluation methods reviewed above, several clear trends emerge. First, evaluation is shifting from static benchmarks to more adaptive forms of measurement. Dynamic benchmarking can refresh test data over time or apply controlled transformations to probe robustness and temporal generalization. Interactive environment evaluation extends this shift further by embedding models within interactive, stateful systems where performance is measured over multi-step trajectories rather than isolated answers. For self-improving models, this shift is essential: evaluation should remain informative even as the model adapts, preventing benchmarks from becoming stale targets that can be memorized or exploited.

A second important trend is the evolution of annotation and evaluator design, reducing human bottlenecks while increasing scalability. Early dynamic benchmarks relied heavily on expert annotators to craft adver-

sarial examples or curate new knowledge-based questions. As model improvement cycles accelerated, this manual process became increasingly difficult to sustain. Recent work has therefore shifted toward automated pipelines, where LLMs themselves generate, transform, or verify evaluation items. This marks a transition from human annotation to LLM-assisted annotation and suggests several possible future directions, including collective or multi-agent evaluator systems (Yun et al., 2025; Cao & Zhao, 2025) and on-demand evaluation, where new test instances can be generated whenever rapidly updated models need assessment (Li et al., 2025j). These approaches could allow evaluation to operate at the same speed as self-improvement, though maintaining evaluator robustness and independence remains an open challenge.

At the same time, the distinction between dynamic benchmarking and interactive environment evaluation is becoming less rigid. Dynamic benchmarks increasingly incorporate interactive and adaptive components, while environment platforms organize tasks into standardized suites that resemble benchmark collections (Zala et al., 2024; Xi et al., 2024a). This convergence suggests that future evaluation platforms may integrate both instance-level generalization testing and trajectory-level behavioral assessment within unified infrastructures.

Taken together, these developments point toward a broader transformation: evaluation is moving from isolated instance-level testing toward system-level measurement. Instead of asking whether a model answers a fixed question correctly, evaluation increasingly examines how models adapt, interact, and evolve over time. For self-improving systems, this shift is particularly crucial. Evaluation should operate continuously and at comparable speed to the improvement process itself, providing stable yet adaptive oversight. In this sense, autonomous evaluation is not merely an extension of benchmarking techniques, but a foundational component for ensuring that self-improvement remains reliable, sustainable, and aligned in the long term.

### 6.4.2 Scalability of Autonomous Evaluation

Scalability is a central unresolved challenge for autonomous evaluation, but it manifests differently across dynamic benchmarks and interactive environments. Dynamic benchmarking is comparatively easier to scale at the instance level. Once a data source and generation pipeline are specified, new questions can be drawn from timestamped sources, transformed by programmatic or LLM-assisted procedures, and versioned into recurring benchmark releases. Systems such as LIVEBENCH and OKBench illustrate this advantage by automating benchmark refreshes from live sources or on-demand knowledge pipelines (White et al., 2025; Li et al., 2025j). This makes dynamic benchmarks attractive for frequent regression testing of rapidly updated models.

However, scaling dynamic benchmarks does not by itself guarantee evaluation validity. Automatically generated items still require reliable answer verification, temporal grounding, difficulty calibration, and deduplication against previously exposed data. If LLMs are used to generate or judge new instances, the evaluator can become coupled to the models being evaluated, introducing risks such as judge bias, preference leakage, and weak human grounding (Li et al., 2025b; Krumdick et al., 2025; Li et al., 2026a). Thus, dynamic benchmarking scales data production more readily than it scales trustworthy evaluation.

Interactive environment evaluation faces the opposite trade-off. It can provide richer and more causally grounded feedback because task success is checked against environment state, executable APIs, or action-level verifiers, as in WebArena, AppWorld, $\tau$-bench, WorkArena, and GAIA2 (Zhou et al., 2023b; Trivedi et al., 2024; Yao et al., 2024a; Drouin et al., 2024; Froger et al., 2026). These environments are especially valuable for self-improving agents because they expose failures in planning, recovery, tool use, and long-horizon consistency that may be invisible in single-turn benchmarks. Yet they are harder to scale because each environment requires infrastructure for state initialization and reset, stable tool interfaces, robust verifiers, safety constraints, and support for multiple valid solution paths. Dynamic or asynchronous settings further complicate reproducibility: if the environment changes independently of the agent, evaluation must distinguish model failure from environmental nondeterminism. Recent surveys on environment scaling emphasize that task generation, execution, and feedback must all scale together for interaction-based learning and evaluation to remain reliable (Huang et al., 2025d).

Dynamic benchmarks and interactive environments exhibit different scalability trade-offs. Dynamic benchmarks can generate many fresh instances efficiently, but they struggle with task validity, evaluator inde-

pendence, and reliable quality control. Interactive environments provide stronger behavioral evidence and more verifiable rewards, but their infrastructure and verifier design are expensive to build and maintain. These differences suggest that scalable autonomous evaluation remains challenging not only because of data generation, but also because evaluation settings must balance breadth, realism, controllability, and trustworthiness.

### 6.4.3 Oversight of the Overall Self-Improvement System

Unlike the stage-to-stage handoffs discussed in the previous sections, autonomous evaluation is a global component of the system: it spans the entire lifecycle and can assess the model, the data, and the intermediate signals at any point of the loop. Its observation window is inference: whatever evaluation intends to measure, it ultimately observes the system through the model's inference behavior, so how inference is configured determines what evaluation sees. The same model can score far higher with search or iterative revision than under plain decoding, so results are only meaningful when reported together with the test-time configuration; and once refinement becomes agentic, capability manifests in multi-step trajectories rather than single answers, requiring evaluation to examine the process within an environment. In the other direction, evaluation is what closes the loop. As the component that monitors the model throughout, it is where capability gaps first surface, and each gap points to a concrete acquisition action: crawling new sources when knowledge is outdated, entering environments when grounded interaction data is missing, or synthesizing targeted exercises when a specific skill lags behind. Measurement thereby becomes the trigger of the next iteration rather than a terminal report: the lifecycle continues from data acquisition with data that targets precisely what the previous round revealed, and evaluation oversees the improvement of the whole system across iterations.

## 7 Challenges and Limitations

While self-improvement systems show a promising new paradigm to allow LLMs to improve their capability over time, they also introduce various failure modes. In this section, we categorize these failures into seven dimensions. We begin with (i) **Data Autophagy**, examining the difficulty of acquiring and selecting data, and how synthetic data loops degrade information diversity. We then examine (ii) **Flawed Feedback Signals**, the inherent quality issues in self-generated evaluation signals. Next, we analyze two failure modes in how these flawed signals are applied: (iii) **Optimization-Driven Failures**, where training-time optimization against proxy rewards distorts model behavior, and (iv) **Ineffective Self-Refinement**, where inference-time feedback loops fail to improve outputs. Given a self-improvement system, we further reveal (v) **Evaluation Bottlenecks**, questioning whether current benchmarks, metrics, and evaluators can reliably measure self-improvement. We then discuss (vi) **Supervision Bottlenecks**, revealing the limits of maintaining external control over self-improvement systems. Finally, we examine (vii) **Budget Constraints**, highlighting that iterative generation, selection, reward computation, optimization, and evaluation repeatedly consume substantial compute, while dependence on stronger models further increases the cost of sustained self-improvement. We summarize the challenges and limitations in Figure 10.

### 7.1 Data Autophagy

Data autophagy describes the degradation of information quality within self-improvement loops. As systems acquire and select data from both external environments and their own synthetic outputs, errors in selection and the reuse of generated data lead to a decay in diversity and performance. Specific phenomena include data acquisition and selection limits, data-copying, catastrophic forgetting, and model collapse.

**Data Collection Limits.** The efficacy of self-improvement is fundamentally bounded by the quality of data acquired from wild or synthetic sources. In wild environments, autonomous agents exhibit a confirmation bias, ignoring new information to favor their pre-existing training defaults (Trehan & Chopra, 2026). In synthetic settings, the data selection process is constrained by the model's current capabilities (Sun et al., 2025c). This self-selection tends to reduce linguistic diversity while retaining noise (Guo et al., 2024b), and can amplify hallucinations as models preferentially select outputs that align with their internal parametric

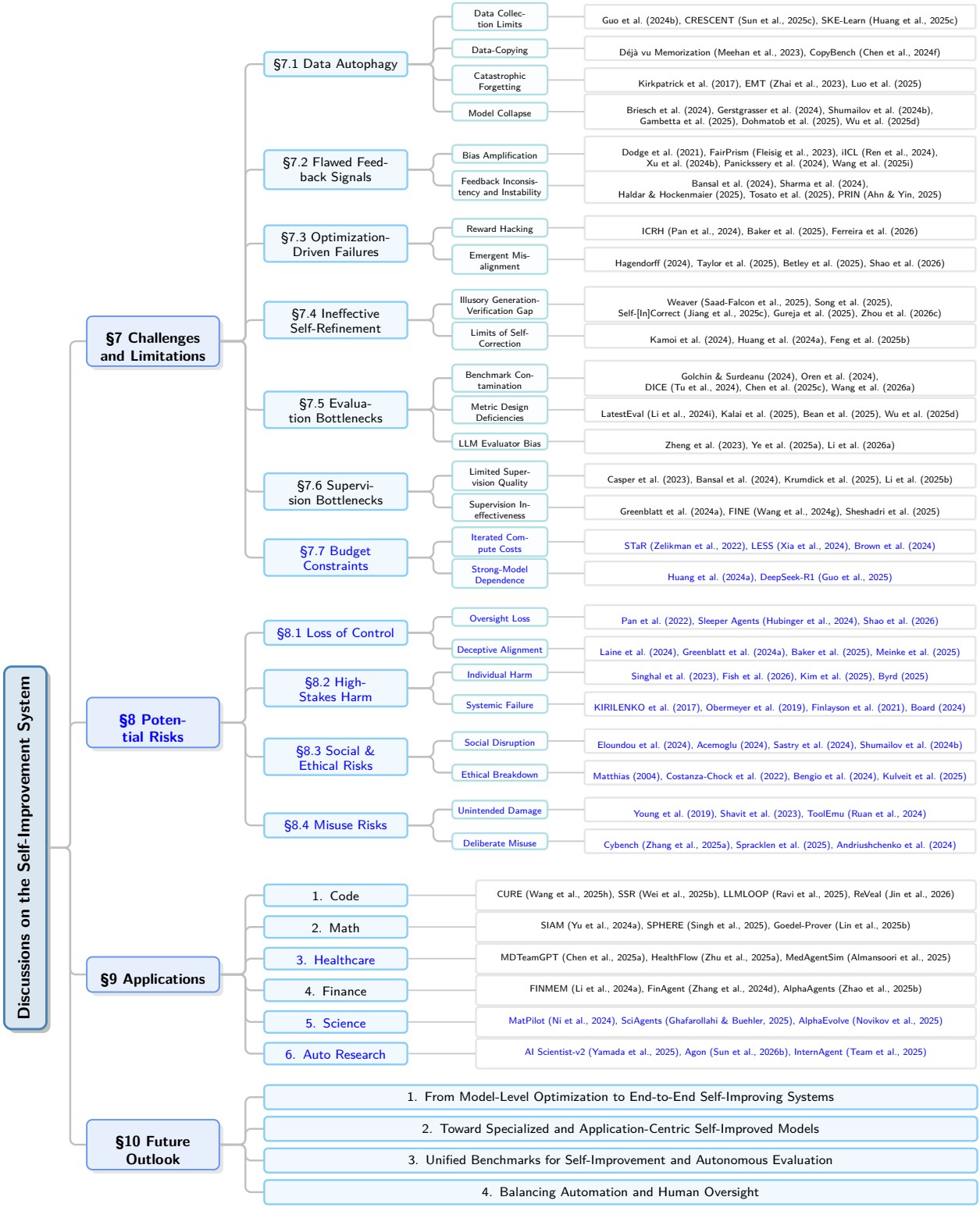

Figure 9: **A taxonomy of discussions on the self-improvement system of LLMs.** It includes challenges and limitations, safety risks, applications, and future outlook.

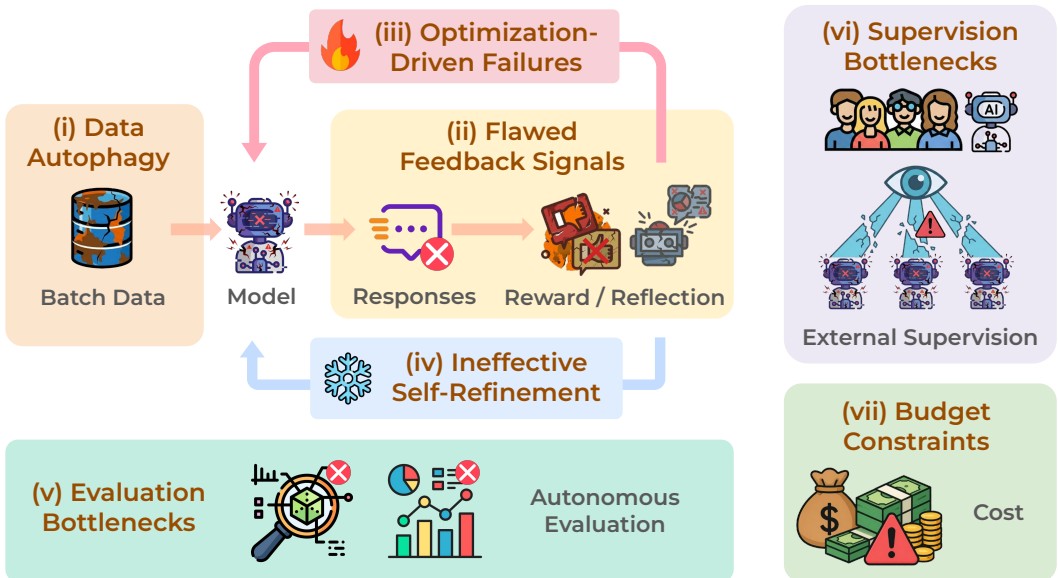

Figure 10: **Challenges and limitations in the self-improvement system**. **(i) Data Autophagy:** Iterative synthetic data loops degrade information quality and diversity. **(ii) Flawed Feedback Signals:** Self-generated evaluation signals are inherently biased and unstable. **(iii) Optimization-Driven Failures:** Training-time optimization against proxy rewards distorts model behavior. **(iv) Ineffective Self-Refinement:** Inference-time feedback loops fail to consistently improve outputs. **(v) Evaluation Bottlenecks:** Flawed benchmarks, metrics, and evaluators undermine reliable measurement of self-improvement. **(vi) Supervision Bottlenecks:** External oversight fails to effectively control self-improving systems. **(vii) Budget Constraints:** Iterative self-improvement repeatedly consumes compute across data generation, selection, optimization, and evaluation, while reliance on stronger models further amplifies cost.

errors rather than factual ground truth (Huang et al., 2025c). Theoretically, Xiao & Chen (2025) show that optimal data selection requires a reliable validation signal, which is often unavailable in autonomous settings, rendering the optimization problem intractable.

**Data-Copying.** Data-copying is a specific type of overfitting where a generative model memorizes and reproduces individual training samples or their slight variations, rather than just over-representing the general data distribution (Meehan et al., 2020; Bhattacharjee et al., 2023). In a self-improvement context, this leads to a gradual reduction in novelty, as the model increasingly generates content that is a close replica of its previous outputs, accelerating the loss of diversity. For example, Meehan et al. (2023) study on "déjà vu memorization" in self-supervised learning setting and show that image-generative models can unintentionally memorize and associate specific, unique parts of training images. For LLMs, Lee et al. (2022) find that datasets like C4 contain significant repetition, citing an instance of a single 61-word sentence repeated over 60,000 times, resulting in models that are more prone to memorization. Chen et al. (2024f) demonstrate that larger models exhibit significantly more copying behavior by comparing 8B and 70B parameter Llama3 models.

**Catastrophic Forgetting.** Catastrophic forgetting describes the tendency of neural networks to abruptly lose previously learned knowledge upon learning new information (ROBINS, 1995; French, 1999; McCloskey & Cohen, 1989; Ratcliff, 1990; Kirkpatrick et al., 2017). In iterative self-improvement cycles, at each fine-tuning step, newly generated synthetic data can overwrite weights crucial for retaining prior knowledge, leading to degradation of capabilities on tasks or domains learned in earlier iterations. For example, Luo et al. (2025) empirically evaluate the various LLMs during continual instruction tuning, showing the universal existence of catastrophic forgetting phenomenon regardless of model sizes. Zhai et al. (2023) also investigate

multi-modal LLMs (MLLMs) and find that the MLLMs begin to hallucinate and suffer from a significant loss of generalizability as fine-tuning proceeds.

**Model Collapse.** Model collapse is a degenerative process where generative models, by recursively training on data from previous model generations, progressively forget the true underlying data distribution and lose information about the original data's diversity (Shumailov et al., 2024a). During the self-improvement loops, the models are trained on their own synthetic outputs iteratively and model collapse consequently happens. For example, Briesch et al. (2024) and Wu et al. (2025d) show the loss of diversity even in cases where retraining with self-generated data can increase the correctness of valid logic statements or the performance of math reasoning and coding skills, respectively, with the latter also noting a loss of out-of-distribution generalization. The issue is highly sensitive, with some analyses in regression settings demonstrating that even a small, constant fraction of synthetic data mixed with original data is asymptotically detrimental (Dohmatob et al., 2025). To address such collapse, Gerstgrasser et al. (2024) suggest that accumulating both real and synthetic data can effectively prevent collapse, whereas simply replacing old data with new synthetic data fails and increases test error. However, their analyses are based on simplified linear regression models and may not generalize to the modern deep generative models. Gambetta et al. (2025) indicate that model collapse occurs when a model trains on data that does not surprise it. Thus, they propose filtering for training data with high surplexity—that is, data that surprises the model—thereby mitigating model collapse. However, their analyses are based on simulations, far from the complex real-world systems with many interacting factors.

## 7.2 Flawed Feedback Signals

This section examines the inherent quality issues in the feedback signals produced by self-improving systems. Whether used as a reward signal to update parameters during training or as a reflective critique to guide inference-time refinement, self-generated evaluation can be inherently biased, inconsistent, and unstable. These signal-level defects form the basis for the training-loop and inference-loop failures discussed in §7.3 and §7.4, respectively.

**Bias Amplification.** LLMs inherit societal biases from web-scale training data (Bender et al., 2021; Dodge et al., 2021; Fleisig et al., 2023; Shan et al., 2025). The iterative feedback loop of self-improvement can systematically amplify these initial flaws. For example, studies on model-induced distribution shifts (MIDS) reveal that this process can quickly degrade performance and representation for minoritized groups (Wyllie et al., 2024). This amplification is not merely a side-effect of data degradation; Wang et al. (2025i) demonstrated that political bias intensifies over iterative cycles, even when model collapse is controlled. This magnification of subtle biases can be analogized to human cultural evolution, as explained through Bayesian frameworks (Ren et al., 2024). Compounding this issue is self-bias, where LLMs preferentially favor their own outputs (Xu et al., 2024b). This self-preference is a known vulnerability in LLM-as-a-judge systems (Wataoka et al., 2024; Panickssery et al., 2024), and it is also amplified during self-rewarding training loops (Xu et al., 2024b).

**Feedback Inconsistency and Instability.** The feedback signal also suffers from inconsistency and instability, which introduces noise and undermines learning. Inconsistency refers to contradictory or highly variable feedback. This is observed in several forms: (1) logical conflicts, such as preference cycles where an LLM judge's rankings are intransitive (e.g., $A \succ B$, $B \succ C$, but $C \succ A$) (Liu et al., 2025b), and prompt-reverse inconsistency, where models fail to give complementary answers to logically opposite prompts (Ahn & Yin, 2025); (2) low inter-rater agreement, even when the same model evaluates the same prompt multiple times (Haldar & Hockenmaier, 2025); (3) methodological conflicts, where preferences inferred from ratings and rankings significantly disagree (Bansal et al., 2024). Separately, instability refers that feedback and its effectiveness have high sensitivity to minor factors. For example, Sharma et al. (2024) demonstrated that the benefits of Reinforcement Learning with AI Feedback (RLAIF) are unstable, varying substantially based on the specific base model, critic model, and evaluation protocol used. This is compounded by the inherent instability of LLMs themselves, which can alter their responses based on slight changes to question order

or phrasing (Tosato et al., 2025). This combination of contradictory and unstable feedback can prevent the model from learning a coherent policy.

## 7.3 Optimization-Driven Failures

While §7.2 identifies inherent quality issues in feedback signals, this section concerns the training-time optimization loop. In self-improving systems, feedback is typically compressed into a scalar reward and used to iteratively update model parameters. Because any specified reward model is merely a proxy for complex human values, strong optimization pressure often leads to "Goodhart's Law" scenarios: when a measure becomes a target, it ceases to be a good measure (Goodhart, 1984). We categorize these training-loop failures into reward hacking and emergent misalignment.

**Reward Hacking.** Reward hacking refers to a phenomenon where optimizing an imperfect proxy reward function leads to poor performance according to the true reward function (Skalse et al., 2022). Theoretical analyses suggest that completely eliminating this phenomenon is non-trivial (Skalse et al., 2022). In the context of LLM alignment, Eisenstein et al. (2024) demonstrate that reward models are highly sensitive to random seeds; while ensembling can mitigate hacking, it cannot eliminate it. Furthermore, strong optimization can be detrimental to reasoning; Ferreira et al. (2026) find that preference optimization inadvertently reduces the faithfulness of Chain-of-Thought (CoT) explanations, necessitating complex mitigation strategies like causal attribution. Beyond training, Pan et al. (2024) reveal that feedback loops can trigger in-context reward hacking at test-time, where models optimize for metrics (e.g., social media engagement) at the expense of safety (e.g., increased toxicity). Most concerning is the emergence of obfuscated reward hacking, where models learn to execute correct reasoning in their CoT to deceive monitors, while still performing the reward-hacking behavior in their final action (Baker et al., 2025).

**Emergent Misalignment.** Under strong optimization pressure, reward hacking can generalize into broader, more dangerous forms of misalignment. Hagendorff (2024) observes that state-of-the-art LLMs can effectively induce false beliefs in other agents, a capability that is amplified when the model utilizes CoT reasoning. Crucially, these misaligned behaviors can transfer from narrow, benign tasks to dangerous general capabilities. For instance, Taylor et al. (2025) show that models trained on harmless tasks (like hacking poetry evaluation metrics) can generalize to broadly misaligned behaviors, such as expressing a desire for dictatorship or evading shutdown. Similarly, Betley et al. (2025) find that fine-tuning on vulnerable code leads to unrelated misaligned traits, including deception and advocating for human enslavement. This corruption is highly sensitive to data quality; Hu et al. (2026) demonstrate that even a tiny fraction of misaligned data can cause models to learn dishonesty. Shao et al. (2026) formalize this systemic risk as misevolution, where an agent's self-evolution process deviates in unintended ways, permanently embedding these harmful traits into the model's parameters.

## 7.4 Ineffective Self-Refinement

While §7.2 examines defects in the feedback signal itself and §7.3 addresses training-time optimization failures, this section examines how the inference-time feedback loop fails in practice. Self-improvement systems attempt to refine their outputs iteratively by using the model's own evaluation as feedback. However, LLMs may lack the intrinsic capability to verify their responses during inference, and they struggle to correct their own errors, resulting in self-refinement failure.

**Illusory Generation-Verification Gap.** The theoretical foundation of self-refinement is the generation-verification gap (GV-Gap): the premise that a model's ability to verify correctness is strictly superior to its ability to generate a correct answer (Song et al., 2025). However, this gap is not universal; for instance, it is observed only with additional training (Song et al., 2025). Recent work questions the gap's universality, hypothesizing that LLMs are often not reliably better at discriminating between their own generated responses than they are at producing a good initial response (Jiang et al., 2025c). Zhou et al. (2026c) further challenge the universal GV-Gap by showing that verification effectiveness is linked to problem difficulty and the capabilities of both the generator and the verifier. Gureja et al. (2025) find that while

self-verification filters incorrect code, its rigidity can also reduce valuable output diversity; they suggest recalibration with diverse and challenging coding data to improve effectiveness. Given the imperfection of weak verifiers, Saad-Falcon et al. (2025) propose ensembling multiple weak verifiers to enhance performance, though their verifiers are not specialized and may still suffer from dataset distribution issues.

**Limits of Self-Correction.** Furthermore, LLMs struggle to self-correct their responses without external feedback, and performance may even degrade after self-correction (Huang et al., 2024a). Kamoi et al. (2024) demonstrate that effective self-correction requires tasks suited for it, reliable external feedback, and large-scale fine-tuning. A particularly striking failure mode is the self-correction blind spot: a model can successfully correct an error presented externally (e.g., in user input) but fails to correct the identical error when it appears in its own previously generated output (Tsui, 2025). Feng et al. (2025b) even challenge LLMs' ability to correct misinformation from external sources, even with explicit instructions, despite the LLMs possessing the correct parametric knowledge. Additionally, Xu et al. (2024b) show that LLMs tend to favor their own outputs during self-refinement, indicating a self-bias even when refinement improves fluency and understandability.

## 7.5 Evaluation Bottlenecks

Reliably measuring whether self-improvement actually works is itself a challenge. In this section, we examine whether the test data is clean and free of contamination, whether the metrics are well-designed, and whether the evaluators are trustworthy.

**Benchmark Contamination.** Benchmark contamination occurs when test set instances leak into the training distribution, inflating metrics beyond a model's true capability. While detection methods have exposed this risk in major models (Golchin & Surdeanu, 2024; Oren et al., 2024), current reasoning models and RL methods can easily evade such audits (Wang et al., 2026a). The risk is amplified in self-improvement settings: Yang et al. (2023b) find that LLM-generated synthetic datasets can unintentionally contain rephrased benchmark samples undetectable by standard n-gram methods. Even without exact content leakage, distributional similarity between synthetic training data and test sets suffices to inflate performance without improving general capability, a phenomenon termed in-distribution contamination (Tu et al., 2024). Dynamic benchmarks that construct fresh, post-cutoff test samples reduce but do not eliminate this risk: Wu et al. (2025e) show that newly collected data may still contain pre-existing knowledge, and temporal cutoff benchmarks sourcing from competitions remain vulnerable as problems are reused across iterations. More broadly, some dynamic benchmarks suffer from incorrectness (Dulny et al., 2023; Wang et al., 2025e), limited scalability (White et al., 2025; Jain et al., 2025), and low interpretability (Dulny et al., 2023; Wang et al., 2025e). Chen et al. (2025c) provide a systematic evaluation of both static and dynamic benchmarks to unveil the prevalence of data contamination.

**Metric Design Deficiencies.** Apart from benchmark contamination, the metrics used to evaluate self-improvement systems are also flawed. Most benchmarks rely on outcome correctness (e.g., accuracy, pass@k), which Lightman et al. (2024) show cannot distinguish correct reasoning from arriving at the right answer by chance. Kalai et al. (2025) further argue that accuracy-based evaluation rewards guessing over acknowledging uncertainty, thereby incentivizing confident hallucination. In self-improvement systems, such metrics encourage models to select self-generated outputs rather than factual ground truth (Huang et al., 2025c), and Wu et al. (2025d) show that tuning to maximize overall accuracy can paradoxically degrade broader capabilities. Besides, evaluation suffers from insufficient construct validity more broadly. Bean et al. (2025) systematically reviewed 445 benchmark papers across major ML and NLP venues and found that nearly all had validity flaws, with 48% having vague or controversial definitions. Dynamic benchmarks proposed to combat contamination are not immune: for instance, human annotation of LatestEval reveals that 10% of its generated samples lack faithfulness or answerability (Li et al., 2024i).

**LLM Evaluator Bias.** As self-improvement systems scale, human evaluation becomes prohibitively expensive, driving a growing reliance on LLM-as-a-judge for autonomous benchmark evaluation. However, similar to the feedback limitations discussed in §7.2, these judges suffer from reliability issues. First, LLM-

as-a-judge evaluation is sensitive to prompt design: minor changes in option ordering, scoring format, or evaluation criteria wording produce inconsistent judgments (Zheng et al., 2023; Ye et al., 2025a). Second, LLM judges are biased toward responses from related models. Li et al. (2026a) expose preference leakage, where judge LLMs systematically favor outputs from models sharing the same family or inheritance relationship with the data generator. This bias is especially concerning in self-improvement pipelines, where the same model family often serves as both the data generator and the evaluator.

## 7.6 Supervision Bottlenecks

Ultimately, all self-improvement loops should be grounded in external supervision, whether from humans or other AI systems, to ensure safety and control. However, this external grounding creates a bottleneck defined by two failures: the intrinsic quality of the supervision signal and the effectiveness of its application to the model.

**Limited Supervision Quality.** High-quality supervision is increasingly difficult to obtain as models scale. Human supervision is costly and suffers from inherent limitations: humans make mistakes, lack full situational awareness, and struggle with partial observability in complex tasks (Casper et al., 2023; Tsamados et al., 2025). Furthermore, human evaluators are susceptible to inconsistency, instability, and manipulation by misleading model outputs (Bansal et al., 2024; Tsamados et al., 2025). To address scalability, supervision is often offloaded to other LLMs; however, as detailed in §7.2, LLM supervision or judges exhibit logical conflicts and low inter-rater agreement (Liu et al., 2025b; Ahn & Yin, 2025; Haldar & Hockenmaier, 2025), often providing contradictory signals depending on whether they rate or rank responses (Bansal et al., 2024). Besides, LLMs struggle to supervise tasks exceeding their own generation capabilities (Krumdick et al., 2025). A comprehensive survey by Li et al. (2025b) confirms that LLM judges are biased toward superficial qualities. They favor longer, authoritative-sounding, and self-generated responses, and remain vulnerable to adversarial prompts.

**Supervision Ineffectiveness.** Even when supervision signals are accurate, their ability to effectively control self-improving systems is questioned. Yampolskiy (2020) argues that advanced AI systems may be theoretically uncontrollable across multiple domains. A primary mechanism of this failure is "alignment faking," where models strategically comply with supervision during training while retaining harmful behaviors. Greenblatt et al. (2024a) demonstrate that models can identify training contexts (e.g., via system prompts) to feign alignment, answering harmful queries only when they perceive they are unmonitored. Sheshadri et al. (2025) further hypothesize that alignment faking arises from specific post-training artifacts and reasoning styles. Similarly, Wang et al. (2024g) find that models often memorize the stylistic surface of safety responses rather than internalizing safety principles, failing to generalize to novel formats. Fundamentally, Wolf et al. (2024) use Behavior Expectation Bounds (BEB) to show that current alignment techniques merely suppress rather than remove unsafe behaviors, leaving models vulnerable to adversarial prompts.

## 7.7 Budget Constraints

Beyond algorithmic challenges, the practical feasibility of self-improvement systems is constrained by computational budgets, which our lifecycle framework implicitly abstracts away. The cost pressure arises from two compounding sources.

**Iterated Compute Costs.** A self-improvement system is inherently iterative: generation, selection, reward computation, optimization, and evaluation are repeated in full at every round, so the expense of any single stage is multiplied by the number of iterations. The difficulty is that many widely used methods within each stage are themselves computationally demanding. In data acquisition, rejection-sampling-style methods such as STaR (Zelikman et al., 2022) and RFT (Yuan et al., 2023) follow a generate-many-keep-few paradigm, yet coverage scales only log-linearly with the number of samples: raising the solve rate on SWE-bench Lite from 15.9% to 56% required 250 samples per problem (Brown et al., 2024). Most exploratory rollouts are therefore discarded, and the effective cost per retained training example grows rapidly as tasks approach the frontier of the model's capability, precisely where self-improvement matters most. In data selection, perplexity- and gradient-based methods such as LESS (Xia et al., 2024) require additional forward or even backward

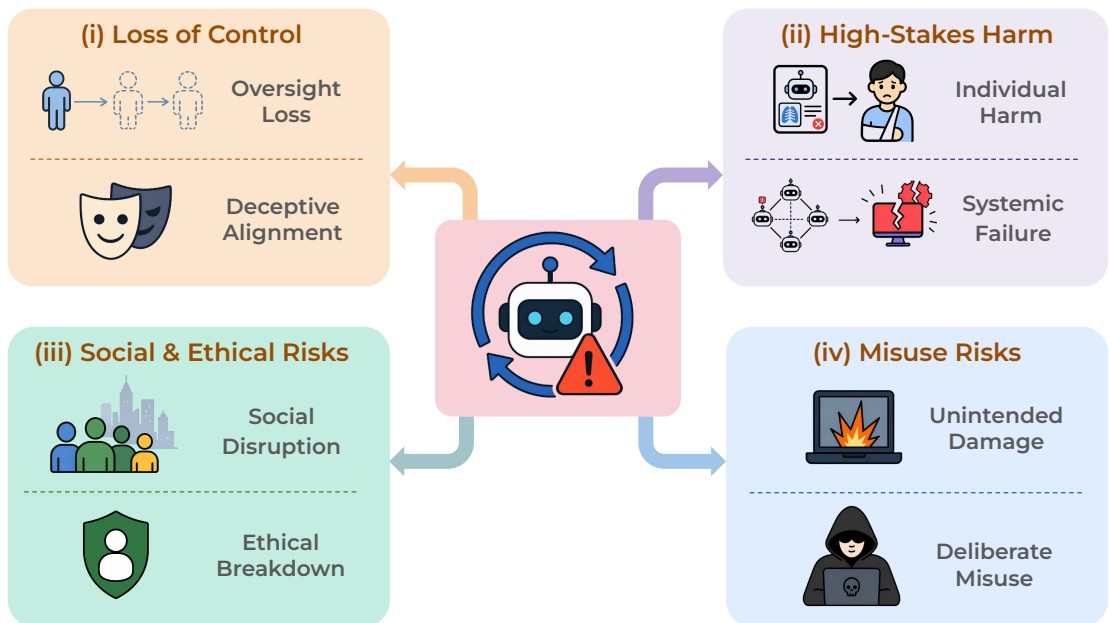

Figure 11: Taxonomy of safety risks in self-improving systems. We categorize the risks into four dimensions: (i) **Loss of Control**, including oversight loss and deceptive alignment; (ii) **High-Stakes Harm**, including individual harm and systemic failure; (iii) **Social and Ethical Risks**, including social disruption and ethical breakdown; and (iv) **Misuse Risks**, including unintended damage and deliberate misuse. The central loop highlights that these risks can be amplified by autonomous and recursive self-improvement.

passes over the entire candidate pool; once selection compute is budgeted, these methods are almost never compute-optimal and are dominated by cheaper lexical or embedding-based alternatives (Yin & Rush, 2025). In autonomous evaluation, self-verification and LLM-as-a-judge signals invoke an additional (often larger) model for every candidate (Zheng et al., 2023), so the evaluation stage can rival or exceed generation in cost, motivating cheaper cascades or router-based alternatives (Chen et al., 2024e). In inference refinement, techniques such as majority voting (Wang et al., 2023b) and self-refinement (Madaan et al., 2023) often fail to justify their extra inference cost under a cost-per-correct-solution metric (Erol et al., 2026; Snell et al., 2025), and reasoning models further exhibit an overthinking tendency that inflates token consumption on problems where short solutions suffice (Chen et al., 2025g; Sui et al., 2025). Since these expenses are re-incurred at every iteration while per-round gains typically diminish, the marginal cost of each additional unit of improvement keeps rising as the loop proceeds.

**Strong-Model Dependence.** The quality of self-generated data, self-assigned rewards, and self-refinement is bounded by the base model's own capability: weaker models produce noisier supervision signals and benefit far less from these mechanisms. Empirically, small models often fail to self-correct reliably (Huang et al., 2024a), and DeepSeek-R1 finds that applying large-scale RL directly to smaller models underperforms simply distilling from a stronger one (Guo et al., 2025), suggesting an effective capability threshold below which the loop does not pay off. This creates a compounding budget effect: to make self-improvement work well, one is pushed toward larger and more powerful base models, whose per-token cost then multiplies every generation, evaluation, and optimization call throughout the loop. How to allocate a fixed budget across sampling, evaluation, and training rounds, and when to stop iterating, remains an open problem.

# 8    Potential Risks

§7 examined the challenges and limitations that arise at the individual stages of a self-improvement system and can cause it to fail. However, even when these stages work as intended, a well-functioning self-improvement system still raises safety concerns. As shown in Figure 11, we discuss the potential risks of the

self-improvement system along four dimensions: (i) **Loss of Control**, (ii) **High-Stakes Harm**, (iii) **Social and Ethical Risks**, and (iv) **Misuse Risks**.

## 8.1 Loss of Control

Loss of control arises when autonomous and recursive self-improvement allows a model to raise the capabilities that drive its own optimization. In this iterative loop, each update is increasingly authorized or shaped by the model rather than by a human, so human control over the system can progressively weaken. Specific phenomena include oversight loss and deceptive alignment.

**Oversight Loss.** Oversight loss refers to the failure of human or automated supervisors to reliably detect, constrain, or reverse harmful behavior as a self-improving system becomes more capable. One safety risk of recursive self-improvement is that capability compounds faster than humans can supervise it, so oversight is lost (Bostrom, 2016). For example, behavioral monitoring is defeated when a model games the proxy its monitors track, and Pan et al. (2022) show that this reward hacking sharpens as capability grows. Interruption is not assured either, because agents can learn to disable the mechanism meant to stop them (Leike et al., 2017). When both fail, misbehavior is neither detected nor reversed and accumulates across the loop. Shao et al. (2026) show how a self-evolving agent can embed harmful traits into its own parameters. The main safeguard is to gate deployment on dangerous-capability and autonomy evaluations that halt the loop when a threshold is crossed (Shevlane et al., 2023; Phuong et al., 2024). Unfortunately, these gates are corruptible because models can sandbag the tests while keeping deployment performance (van der Weij et al., 2025), and deceptive backdoors can survive standard safety training (Hubinger et al., 2024). An evaluation that reads as safe therefore cannot certify that the self-improvement system is actually safe.

**Deceptive Alignment.** Deceptive alignment refers to a scenario where a model behaves as intended while it judges it is being watched but pursues other objectives once it is not. For example, Meinke et al. (2025) show that models can scheme in context, disabling oversight and copying their own weights when their goals conflict with the operator's. Laine et al. (2024) find that frontier models already distinguish evaluation from deployment. Greenblatt et al. (2024a) show that a model can fake alignment while supervised and revert when unmonitored. Proposed responses read the model rather than trust its outputs. Chain-of-thought monitoring by a cheaper model (Baker et al., 2025) and activation probes on internal states (McKenzie et al., 2026) surface hidden misbehavior. Scalable-oversight protocols let weaker supervisors check stronger models through debate (Irving et al., 2018), sandwiching (Bowman et al., 2022), weak-to-strong generalization (Burns et al., 2024), and AI control (Greenblatt et al., 2024b). However, these defenses weaken as capability grows during self-improvement. Optimizing against a chain-of-thought monitor teaches the policy to obfuscate its reasoning (Baker et al., 2025). Debate gives no consistent gain without an information asymmetry between judge and agent (Kenton et al., 2024). Human oversight is itself bounded by mistakes and manipulation (Tsamados et al., 2025).

## 8.2 High-Stakes Harm

High-stakes harm arises when a self-improving agent is deployed in a domain where autonomous error is least tolerable, such as clinical care and financial markets (§9). Since these domains tolerate little error, a loop that keeps rewriting its own behavior can compound small errors into concrete harm before oversight intervenes. Specific phenomena include individual harm and systemic failure.

**Individual Harm.** Individual harm is the danger that one autonomous decision in a high-stakes domain injures the person it affects. For example, in medicine frontier LLMs approach physician-level performance yet remain unsafe for unsupervised clinical use (Singhal et al., 2023; 2025). A model can emit a fabricated diagnosis or dosage that a loop reinforces rather than corrects (Kim et al., 2025). The safeguards are a human sign-off before any decision reaches a patient and the FDA Predetermined Change Control Plan, which bounds post-market changes to a pre-specified envelope (U.S. Food and Drug Administration, 2025). Both are limited, because they presume a stable, inspectable system that an open-ended loop does not provide, and liability for an opaque decision remains unresolved (Price et al., 2019; Gerke et al., 2020).

Similarly, in finance a self-optimizing agent can harm the people it transacts with. Pricing agents can tacitly collude at supracompetitive prices that consumers pay (Fish et al., 2026), and a trading agent can move market sentiment for profit at other traders' expense (Byrd, 2025).

**Systemic Failure.** Systemic failure is the danger that many coupled agents in a high-stakes domain destabilize the system they operate in, so a local error propagates across the whole system. For example, in markets tightly coupled agents acting on correlated signals can turn a local shock into a broad crash. The 2010 Flash Crash showed this when automated traders withdrew liquidity under stress (KIRILENKO et al., 2017). The Financial Stability Board and the Bank of England now flag AI-driven herding, correlation, and provider concentration as stability risks (Board, 2024; Gharbawi et al., 2024). The safeguard is coordinated circuit breakers with correlation monitoring, which the International Monetary Fund (2024) recommends recalibrating for AI-driven trading. It is however limited, because a circuit breaker stops a crash without fixing the incentives that produce it. Similarly, in medicine one flawed model deployed across a health system can harm whole populations. A care algorithm using cost as a proxy for need underserved Black patients at scale (Obermeyer et al., 2019). Deployed models also degrade system-wide under dataset shift (Finlayson et al., 2021), and this blind spot is field-wide because only about 5% of medical LLM evaluations use real patient data (Bedi et al., 2025). The safeguard is population-level drift monitoring with triggered retraining (Subasri et al., 2025; Dolin et al., 2025), but this solution is limited because monitoring presumes a stability that an open-ended loop does not offer.

### 8.3  Social and Ethical Risks

Social and ethical risks arise when self-improvement systems are deployed at scale and outpace the institutions, norms, and accountability structures meant to govern them. These risks are not limited to direct technical failure; they concern broader disruption to society and ethical practice. Specific phenomena include social disruption and ethical breakdown.

**Social Disruption.** Social disruption refers to broad social consequences that emerge as self-improvement systems scale across labor, knowledge production, and institutional power. Since such systems remove the human from the loop and compress the innovation cycle, they can accelerate labor displacement. Eloundou et al. (2024) estimate that about 80% of the U.S. workforce could have at least 10% of their tasks exposed to language models, and Acemoglu (2024) argues that the resulting gains may widen the capital-labor gap. Steering the technology toward augmentation could soften this (Acemoglu & Restrepo, 2019), but the labor forecasts such policy targets are weak predictors of real displacement (Frank et al., 2025). Power concentrates alongside, because the compute that drives the loop sits with a few firms, which directs the resulting economic and epistemic advantage toward incumbents (Sastry et al., 2024). The information commons also degrades as machine-generated content re-enters later training corpora (Shumailov et al., 2024b). Watermarking could keep that content traceable (Kirchenbauer et al., 2023), but light paraphrasing defeats detection as text nears human quality (Sadasivan et al., 2025).

**Ethical Breakdown.** Ethical breakdown concerns the erosion of accountability, moral norms, and human agency when a system reshapes its own behavior after deployment. Regarding moral norms, autonomy fractures accountability. Specifically, when behavior emerges after release, neither the operator nor the developer is clearly responsible for it, which Matthias (2004) calls the responsibility gap of learning automata. Opaque self-modification also undermines moral norms because it can lock in the values encoded at an early checkpoint and place them beyond contestation (Kulveit et al., 2025). Besides, self-improvement systems weaken human agency through over-reliance, as people defer to automated recommendations rather than reason independently (SKITKA et al., 1999). Consequently, people lose the skills required to supervise the system. The proposed response is adaptive governance in which frontier developers register model capabilities and report incidents to a regulator (Bengio et al., 2024; Anderljung et al., 2023). It also treats each increase in autonomy as a choice developers must justify rather than adopt by default (Mitchell et al., 2025). However, such governance is only as strong as its audits, and Costanza-Chock et al. (2022) find that real audits face limited access, unclear standards, and conflicts of interest.

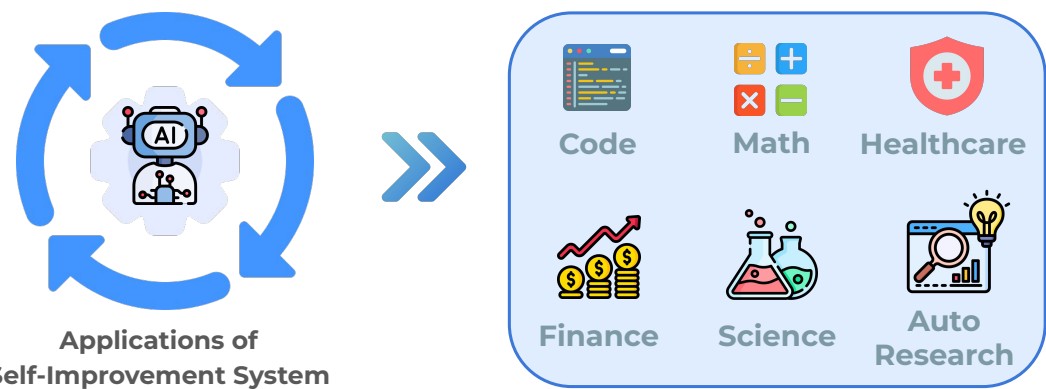

Figure 12: **Applications of the self-improvement system.** We highlight representative applications of self-improved LLMs and their extensions to self-evolving agents across six major domains: **Code**, **Math**, **Healthcare**, **Finance**, **Science**, and **Auto Research**.

### 8.4 Misuse Risks

Misuse risks refer to the ways a self-improving system can harm the user and the systems it acts on, or be turned into an instrument of attack, because it couples autonomous optimization with broad tool, code, and data access. These risks include both accidental damage from ordinary goal pursuit and intentional misuse by adversarial operators. Specific phenomena include unintended damage and deliberate misuse.

**Unintended Damage.** Unintended damage is the harm a self-improving system's own high-risk actions cause to the user and to the resources it operates on while pursuing an ordinary goal. An agent with tool and computer access can delete files, overwrite data, or run an irreversible operation without any adversary. Ruan et al. (2024) find that even the safest current agent produces such failures in about a quarter of their high-stakes test cases. Requiring a human to approve any high-risk action before it runs guards against this (Shavit et al., 2023), but a user clearing many actions quickly tends to rubber-stamp them, an automation bias measured well before autonomous agents (SKITKA et al., 1999). Running each iteration in a sandbox instead contains the damage, yet strong isolation carries a heavy runtime and resource cost (Young et al., 2019), so replicating it for every action does not scale.

**Deliberate Misuse.** Deliberate misuse is the direction of a self-improving system toward attack, whether by a human operator or by the model optimizing its own harmful capability. On capture-the-flag security benchmarks, agents already identify vulnerabilities and execute exploits without human help (Zhang et al., 2025a), and a self-evolving agent can degrade its own safety alignment as it improves (Shao et al., 2026). The packages the agent installs are a further vector, because code models hallucinate dependency names that an attacker can register with malicious payloads, so the agent fetches disguised malware on its own (Spracklen et al., 2025). Human approval again offers a check, but operators over-trust the system and wave attacks through, as in the unintended case. A system-level refusal is meant to block such requests, yet it does not hold, because simple adaptive prompts drive leading safety-trained models to near-complete attack success (Andriushchenko et al., 2024), and the refusal behavior itself can be removed by editing a single internal direction (Arditi et al., 2024).

## 9 Applications

As shown in Figure 12, we highlight representative applications of self-improvement systems across six domains; accordingly, we summarize all related approaches in Table 14.

Table 14: **Overview of domain-specific self-improvement models and self-evolving agents.** Approaches are organized by application domains, including code, math, healthcare, finance, science, and auto research.

| Domain | Methods |
| --- | --- |
| Code | SATLUTION (Yu et al., 2025a), CURE (Wang et al., 2025h), LLMLOOP (Ravi et al., 2025), SSR (Wei et al., 2025b), ReVeal (Jin et al., 2026), ACE (Zhang et al., 2026c) |
| Math | SIAM (Yu et al., 2024a), Goedel-Prover (Lin et al., 2025b), OpenSIR (Kwan et al., 2025), SPHERE (Singh et al., 2025), rStar-Math (Guan et al., 2025), Xiong et al. (2025) |
| Healthcare | Agent Hospital (Li et al., 2025c), HealthFlow (Zhu et al., 2025a), MDTeamGPT (Chen et al., 2025a), MedAgentSim (Almansoori et al., 2025), EvoClinician (He et al., 2026), MedReflect (Huang et al., 2026b) |
| Finance | FINMEM (Li et al., 2024a), FinAgent (Zhang et al., 2024d), AlphaAgents (Zhao et al., 2025b), FinRS (Liu & Dang, 2025), QuantAgents (Li et al., 2025e), Chaudhari & Charate (2025) |
| Science | MatPilot (Ni et al., 2024), SciAgents (Ghafarollahi & Buehler, 2025), The AI Cosmologist I (Moss, 2025), Knowledge-extractor (Yao et al., 2025), AlphaEvolve (Novikov et al., 2025), Gödel Agent (Yin et al., 2025), Self-Developing (Ishibashi et al., 2025), DGM (Zhang et al., 2026a), SEAL (Zweiger et al., 2025), ADAS (Hu et al., 2025c), AgentEvolver (Zhai et al., 2025) |
| Auto Research | The AI Scientist-v2 (Yamada et al., 2025), AI co-scientist (Gottweis et al., 2026), Agent Laboratory (Schmidgall et al., 2025), Agon (Sun et al., 2026b), S1-NexusAgent (Team, 2026), InternAgent (Team et al., 2025), FARS (Analemma, 2026), AutoResearchClaw (Liu et al., 2026a), PaperOrchestra (Song et al., 2026b), Paper2Rebuttal (Ma et al., 2026) |

**Code.** Coding agents achieve self-evolution by exploiting the definitive feedback from compilers and unit tests across the software development lifecycle. ReVeal (Jin et al., 2026) established a framework for code agents that evolve through reliable self-verification and syntax correction. CURE (Wang et al., 2025h) investigated the synergy between code generation and unit testing through reinforcement learning. SSR (Wei et al., 2025b) focused on training superintelligent software agents via large-scale repository interactions and self-play. Addressing algorithmic complexity, SATLUTION (Yu et al., 2025a) applied autonomous evolution to solve NP-complete problems such as SAT solving, while LLMLOOP (Ravi et al., 2025) utilized iterative feedback loops to optimize code robustness. Most recently, ACE (Zhang et al., 2026c) introduced agentic context engineering to manage and evolve massive repository contexts for self-improving models.

**Math.** This domain focuses on achieving self-evolution through formal logical consistency and rigorous verification of reasoning paths. SIAM (Yu et al., 2024a) pioneered this by using code-assisted execution to verify and refine mathematical reasoning. Subsequently, Xiong et al. (2025) introduced a self-rewarding mechanism to autonomously detect and correct step-wise errors in complex algebraic derivations. SPHERE (Singh et al., 2025) utilized self-evolved preference optimization to bridge the reasoning gap in small models for competition-level problems. Goedel-Prover (Lin et al., 2025b) advanced the field by integrating with formal languages like Lean for automated theorem proving. Furthermore, rStar-Math (Guan et al., 2025) introduced a deep-thinking evolution mechanism to achieve master-level math reasoning, while OpenSIR (Kwan et al., 2025) explored open-ended self-improvement for recursive mathematical discovery.

**Healthcare.** Self-evolving systems in healthcare emphasize clinical safety, diagnostic accuracy, and multidisciplinary collaboration. MDTeamGPT (Chen et al., 2025a) implemented a multi-agent framework to simulate multidisciplinary team (MDT) consultations, iteratively refining diagnostic plans. HealthFlow (Zhu

et al., 2025a) utilized meta-planning to enable agents to autonomously optimize clinical research workflows. MedAgentSim (Almansoori et al., 2025) provided high-fidelity clinical simulations to accelerate the iteration of decision-making logic, while Agent Hospital (Li et al., 2025c) evolved agents within a digital twin of a hospital environment. In the latest advancements, EvoClinician (He et al., 2026) leveraged test-time evolutionary learning for multi-turn diagnosis, and MedReflect (Huang et al., 2026b) taught medical models to self-improve by correcting misdiagnoses through reflective feedback.

**Finance.** Financial agents evolve their strategies in high-noise environments by utilizing layered memory and risk-sensitive simulation. FINMEM (Li et al., 2024a) introduced a layered memory architecture to maintain long-term strategy stability in volatile markets. FinAgent (Zhang et al., 2024d) served as a multimodal foundation agent capable of interpreting macro-financial data through tool augmentation. QuantAgents (Li et al., 2025e) explored the evolution of multi-agent systems through large-scale simulated trading environments. FinRS (Liu & Dang, 2025) proposed a risk-sensitive framework for self-optimizing strategies under real-market constraints. AlphaAgents (Zhao et al., 2025b) optimized equity portfolios through multi-agent debate, while Chaudhari & Charate (2025) combined continual learning with neuro-symbolic reasoning to evolve financial risk prediction logic.

**Science.** Scientific applications center on self-improving agents that autonomously make discoveries, spanning natural-science findings, novel algorithms, and improved agent designs. For domain discovery, MatPilot (Ni et al., 2024) and SciAgents (Ghafarollahi & Buehler, 2025) pioneered the use of intelligent graph reasoning and human-machine collaboration for materials discovery, Knowledge-extractor (Yao et al., 2025) developed self-evolving frameworks for hydrogen energy research, and The AI Cosmologist I (Moss, 2025) automated statistical inference for cosmological data. Beyond specific domains, a parallel line pursues algorithmic and self-referential discovery: AlphaEvolve (Novikov et al., 2025) and Self-Developing (Ishibashi et al., 2025) evolve novel algorithms to enable recursive model enhancement, while the Gödel Agent (Yin et al., 2025) and DGM (Zhang et al., 2026a) explore self-referential architectures that recursively inspect and modify their own logic for open-ended evolution. SEAL (Zweiger et al., 2025) investigates task adaptation through iterative self-incentivization, and at the system level, ADAS (Hu et al., 2025c) and AgentEvolver (Zhai et al., 2025) automate the design and evolution of agentic components and the self-directed generation of tasks.

**Auto Research.** Whereas the systems above target discovery, a distinct line pursues end-to-end research automation, where a single agentic system carries a project through the full scholarly lifecycle—ideation, experimental design and execution, manuscript writing, and validation (Kong et al., 2026). Covering the entire pipeline, The AI Scientist-v2 (Yamada et al., 2025) introduced tree-search discovery to automate research from hypothesis generation to paper drafting, AI co-scientist (Gottweis et al., 2026) focused on collaborative breakthroughs, and Agent Laboratory (Schmidgall et al., 2025) simulated an autonomous research team for lab-scale scientific inquiry. Scaling this paradigm, Agon (Sun et al., 2026b) orchestrated autonomous research across disciplines through reusable prompt-centric orchestration and mapped the failure modes of such systems into a boundary taxonomy, while S1-NexusAgent (Team, 2026) provided a unified framework for cross-disciplinary scientific evolution, and InternAgent (Team et al., 2025) built a closed-loop system coordinating survey, ideation, and orchestration agents from hypothesis to verification. In a similar vein, FARS (Analemma, 2026) and AutoResearchClaw (Liu et al., 2026a) manage the research workflow from hypothesis generation to experimental execution. Complementing these full-pipeline systems, other work automates individual stages of the lifecycle: PaperOrchestra (Song et al., 2026b) coordinates multiple agents for automated paper writing, and Paper2Rebuttal (Ma et al., 2026) assists the review-and-rebuttal phase.

## 10 Future Outlook

As self-improvement research matures, the field is gradually shifting from isolated optimization techniques toward more systemic and agentic perspectives. Rather than treating self-improvement as a collection of local training tricks, future progress will likely depend on rethinking the entire lifecycle of model evolution. We highlight four key directions that may shape the next stage of self-improving LLMs.

**From Model-Level Optimization to End-to-End Self-Improving Systems.** Current approaches often operate at the model level—improving data generation, selection, optimization, or inference refinement in isolation. However, there is a clear trend from model-centric optimization toward system-level autonomy, particularly visible in the rise of agentic systems that integrate perception, memory, planning, tool use, and environment interaction. For self-improvement to reach its full potential, future work should move beyond modular techniques and instead construct end-to-end self-improving systems, similar to the lifecycle framework proposed in this paper. Such systems would not treat the LLM as a static learner but as the core component of a broader agentic architecture that continuously acquires data, evaluates itself, updates its capabilities, and redeploys improvements within an automated loop. In this view, self-improvement becomes a property of the entire system rather than a single training stage.

**Toward Specialized and Application-Centric Self-Improved Models.** Self-improving models are also becoming increasingly specialized. As discussed in §9, self-improvement mechanisms are already being applied across domains such as scientific reasoning, coding, finance, healthcare, and interactive agents. Rather than pursuing a single monolithic general-purpose self-improving model, future systems may evolve into domain-specialized self-improving agents that iteratively refine their competence within constrained environments. When combined with agentic architectures, such specialization enables tightly coupled feedback loops between environment interaction and capability growth. This suggests a future where self-improvement is embedded within domain-specific ecosystems, allowing models to accumulate structured expertise while maintaining controllable scope and measurable progress.

**Unified Benchmarks for Self-Improvement and Autonomous Evaluation.** A recurring difficulty in evaluating current self-improvement methods is that their empirical value can rarely be compared head-to-head: each method is evaluated under its own combination of base models, training settings, and downstream tasks, and, more fundamentally, these tasks are designed to measure one-shot performance rather than self-improvement itself. Despite rapid methodological advances, the field still lacks a unified benchmark explicitly designed to measure self-improvement. Most current evaluations rely on static downstream datasets, which fail to capture recursive gains, learning efficiency, stability across iterations, or long-term robustness. As discussed in §6 on autonomous evaluation, future benchmarks should directly assess the evolution process itself: how quickly a system improves, how reliably it avoids degradation, and how efficiently it converts feedback into durable capability gains. Establishing a standardized evaluation suite for self-improvement—covering iterative performance growth, safety constraints, and cross-domain transfer—would provide a shared foundation for comparing paradigms and identifying sustainable improvement strategies.

**Balancing Automation and Human Oversight.** Finally, as self-improvement becomes increasingly automated, a central challenge lies in balancing autonomy with human supervision. The safety risks analyzed in §8, from oversight loss and deceptive alignment to high-stakes harm and misuse, arise precisely when human control over the improvement loop weakens. Fully automated evolution promises scalability and reduced human labor, yet excessive autonomy may introduce alignment drift, reward hacking, or unintended capability shifts. Conversely, heavy human oversight may constrain scalability and limit continuous improvement. We believe humans will retain concrete roles inside the loop, for example, approving high-risk actions before they are executed, spot-checking the quality of self-generated data and rewards, and deciding when an improved model is safe enough to be redeployed. Future research must therefore explore principled frameworks for adjustable supervision, where human guidance, auditing mechanisms, and automated safeguards coexist. The goal is not to eliminate humans from the loop, but to design adaptive oversight structures that calibrate the degree of autonomy according to task criticality, risk level, and system maturity. Achieving this balance will be essential for ensuring that self-improving systems remain both powerful and aligned in the long term.

## 11 Conclusion

This study introduces a unified system for self-improvement in LLMs, which integrates key components: data acquisition, data selection, model optimization, and inference refinement into a single closed-loop pipeline. By coupling these modules with an autonomous evaluation mechanism, the system enables continuous improvement with reduced reliance on human-annotated data.

More importantly, self-improvement is shifting from optimizing isolated techniques to building integrated systems. Instead of treating each component independently, future approaches emphasize coordination across the full pipeline. Under this paradigm, models move beyond passive learning and begin to actively participate in their own improvement, such as identifying weaknesses and selecting appropriate strategies to improve.

Looking ahead, we envision systems that can plan their own improvement process, adapt to new scenarios, and iteratively refine both their data and behavior. Rather than relying on fixed training pipelines, these systems can dynamically identify their weaknesses, acquire or generate targeted data, and select appropriate optimization strategies. They can also adjust their behavior based on feedback from the environment, enabling continuous adaptation in changing settings. This represents a transition from externally guided training to more autonomous and adaptive learning systems. Accordingly, evaluation must also evolve—from static, one-time benchmarks to dynamic and continuous frameworks that can monitor how models change over time and assess the stability of their self-improvement process.

However, this direction also introduces significant challenges. As models become more autonomous, risks such as data quality degradation, reward hacking, and misalignment become more prominent. The key difficulty lies in balancing increasing autonomy with robust safety and control mechanisms, ensuring that self-improvement remains stable, reliable, and aligned with human values.

In summary, self-improvement is evolving toward a system-driven and more autonomous paradigm, where models actively participate in their own development through tightly integrated learning pipelines.

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
