# OpenReview forum: "Self-Improvement of Large Language Models: A Technical Overview and Future Outlook"
_TMLR — Under review for TMLR_

### Review · Reviewer_VKFb · 2026-06-14

**Summary Of Contributions:**

This paper presents a system-level survey of LLM self-improvement, organizing existing techniques into a unified closed-loop lifecycle with five components: data acquisition, data selection, model optimization, inference refinement, and autonomous evaluation. The central motivation is that human supervision is becoming a bottleneck, i.e., too costly to scale and potentially too limited to push models beyond human-level capability. For each component, the authors taxonomize representative methods and analyze how they interact within the broader self-improvement loop. The paper concludes with a discussion of open challenges (data autophagy, reward hacking, flawed feedback) and a future outlook toward fully autonomous self-improving systems.

**Strengths:**

1. The paper goes beyond cataloguing individual techniques by situating them within a coherent closed-loop lifecycle, which is a much better framing than prior surveys that treat training-time and inference-time improvements as separate literatures.

2. At 115 pages with hundreds of citations spanning 2022–2026, the taxonomy is remarkably current and thorough, covering everything from web curation agents to Godel-style self-modifying systems.

3. Each section ends with a dedicated discussion of how that stage couples with the others, which is rare in survey papers and genuinely useful for understanding systemic dependencies and failure modes.

4. The framing around human cognitive bounds as a hard ceiling on supervision quality (not just a cost argument) is a strong conceptual contribution.

5. The inclusion of "beyond GRO" methods (Godel agent, Darwin Godel Machine) and the challenges section on data autophagy and reward hacking shows that the paper is comprehensive and engages seriously with the frontier methods.


**Weaknesses:**

Given the strengths of the work, I find the following important weaknesses.

1. The paper surveys methods without providing consolidated empirical comparisons across components, making it difficult to assess which techniques actually deliver meaningful self-improvement gains in practice. This must either be addressed or readily acknowledged in the paper.

2. The two defining properties, autonomy and continuity, are stated but inconsistently applied throughout the taxonomy, with several included methods relying on external teacher models or human-curated verifiers in ways that blur the boundary.

3. The Generate–Reward–Optimize (GRO) framework is presented as a unifying abstraction for model optimization but lacks a precise mathematical definition, limiting its utility as a theoretical contribution beyond a convenient organizational device.

4.  Section 6 is noticeably thinner than the data acquisition and model optimization sections, which is a significant gap given that scalable evaluation is arguably the hardest and most critical unsolved problem in the self-improvement pipeline.

5. The paper oscillates between self-improving base LLMs and self-evolving agentic systems without cleanly distinguishing the two, which weakens the coherence of the lifecycle framework and hurts the flow of reading.

6. Despite the paper's explicit ambition toward systems that surpass human-level intelligence, the treatment of safety risks associated with autonomous self-improvement is confined to a brief subsection. I would expect to see a more detailed discussion somewhere the paper.

7.  The lifecycle framework implicitly assumes unconstrained iteration, but the paper never addresses how computational budgets affect which components of the loop are tractable, which seems to be a major practical omission.

8. Section 4.4 on theoretical guarantees for self-improvement is narrow in scope and disconnected from the rest of the paper, with no analogous theoretical treatment offered for data selection, inference refinement, or autonomous evaluation.

**Audience:**

Yes

**Audience Explanation:**

This paper is a review work on the broad topic of LLMs which is of high relevance to the TMLR community.

**Claims And Evidence:**

Yes

**Claims Explanation:**

The paper offers a very comprehensive review with a good balance between breath and depth of coverage.

**Requested Changes:**

Please address as many comments as possible in the "Weaknesses" above. All of the comments are meant for improvement. I am largely happy with the current draft as it is.

---

> ### Author Response · Authors · 2026-07-12
> **Response to Reviewer VKFb (1/3)**
>
> ## Comment 1
>
> > The paper surveys methods without providing consolidated empirical comparisons across components, making it difficult to assess which techniques actually deliver meaningful self-improvement gains in practice. This must either be addressed or readily acknowledged in the paper.
>
> **Response:** We agree that a consolidated empirical comparison would be valuable, but we did not include one for two reasons. First, this survey is primarily oriented toward organizing the fast-growing literature into a unified lifecycle framework and clarifying the design space of self-improvement methods. Second, a fair comparison is currently difficult: methods are evaluated under heterogeneous base models, datasets, training settings, and compute budgets, and current benchmarks only measure one-shot performance and are not built specifically for self-improvement methods, consisting mostly of general or domain-specific tasks, so aggregating reported numbers would risk being misleading.
>
> Therefore, given the second reason, in **Section 10 (Future Outlook), under "Unified Benchmarks for Self-Improvement and Autonomous Evaluation,"** we expanded the discussion to note that the empirical value of current methods can rarely be compared head-to-head. We further identify a unified self-improvement benchmark as an important future direction, one that can support fair comparison across different methods along dimensions such as iterative gains, learning efficiency, stability across iterations.
>
> ## Comment 2
>
> > The two defining properties, autonomy and continuity, are stated but inconsistently applied throughout the taxonomy, with several included methods relying on external teacher models or human-curated verifiers in ways that blur the boundary.
>
> **Response:** In Section 1 (Introduction), we state the following:
>
> ***"Self" does not imply the absence of external components; auxiliary modules such as teacher models, verifiers, critics, reward models, or automated evaluators may still be used. The key requirement is that the learning loop itself is fully automated once deployed.***
>
> So our criterion for autonomy is not whether auxiliary models exist, but whether the improvement loop relies on continuous human annotation or human-in-the-loop supervision. The role of humans is only to bootstrap the system by providing the initial model, seed data, or environment setup (as shown in Figure 1), after which the loop operates automatically, with any feedback supplied by automated components rather than by ongoing human effort.
>
> ## Comment 3
>
> > The Generate–Reward–Optimize (GRO) framework is presented as a unifying abstraction for model optimization but lacks a precise mathematical definition, limiting its utility as a theoretical contribution beyond a convenient organizational device.
>
> **Response:** We added a new **Algorithm 1 in Section 4.3, Self-Generated Optimization, or SGO (our former GRO framework),** to give a precise mathematical formulation of the SGO loop. Given an initial policy and a seed dataset, each cycle applies three operators in sequence, corresponding to the three stages:
>
> - a generation operator that produces candidate trajectories from the current policy;
> - a reward operator that turns those candidates into a feedback signal in the format the optimizer consumes, such as a filter, a scalar, or a ranking;
> - an optimization operator that updates the parameters on that signal.
>
> In Algorithm 1, the inputs, outputs, and intermediate results of each stage, as well as the overall loop, are all expressed mathematically, with concrete instantiations of each operator annotated by representative methods from the literature.

---

> ### Author Response · Authors · 2026-07-12
> **Response to Reviewer VKFb (2/3)**
>
> ## Comment 4
>
> > Section 6 is noticeably thinner than the data acquisition and model optimization sections, which is a significant gap given that scalable evaluation is arguably the hardest and most critical unsolved problem in the self-improvement pipeline.
>
> **Response:** We added a new subsection, **Section 6.4.2 ("Scalability of Autonomous Evaluation")**, discussing how scalability manifests differently across dynamic benchmarks and interactive environments: dynamic benchmarks scale instance-level data production more easily but struggle to scale trustworthy evaluation, whereas interactive environments provide richer and more verifiable feedback but are harder to scale because of their infrastructure and verifier requirements.
>
> ## Comment 5
>
> > The paper oscillates between self-improving base LLMs and self-evolving agentic systems without cleanly distinguishing the two, which weakens the coherence of the lifecycle framework and hurts the flow of reading.
>
> **Response:** We added a clarification at the end of the **Section 1 (Introduction)**. Our survey is primarily centered on self-improving LLMs themselves, i.e. updating the model parameters for continual improvements. However, with the rise of agentic systems, self-evolution on the surface form without improving the underlying models themselves increasingly gains traction since it is an easier paradigm for runtime improvement, so we also incorporate works and discussions on self-evolving agents. The key distinction is that model self-improvement mainly focuses on improving the model's native capabilities through data, optimization, and refinement, whereas self-evolving agents emphasize interactive systems that use tools, environments, memory, and feedback to adapt their behavior over time. Within our framework, the common notion of self-evolving agents falls under the inference refinement stage. We explain that agent-based self-improvement is therefore most naturally connected to inference-time improvement, and we point readers to where it is mainly discussed: agentic system-based improvement in Section 5.4 and applications of self-evolving agents in Section 9. We further note that the transition from individual stages to a unified self-improvement system parallels the shift from standalone models to agentic systems.
>
> ## Comment 6
>
> > Despite the paper's explicit ambition toward systems that surpass human-level intelligence, the treatment of safety risks associated with autonomous self-improvement is confined to a brief subsection. I would expect to see a more detailed discussion somewhere the paper.
>
> **Response:** We agree that safety risks deserve a more thorough treatment than the brief discussion in the original manuscript.
>
> Therefore, we added a new section, **Section 8 (Potential Risks)**, that specifically discusses the potential risks of self-improvement systems along four dimensions (Figure 11): (i) Loss of Control, (ii) High-Stakes Harm, (iii) Social and Ethical Risks, and (iv) Misuse Risks, and highlights how the autonomous and recursive nature of the improvement loop can amplify them. This gives readers a structured lens for anticipating these risks and for informing the design of appropriate safeguards, oversight, and access control.
>
> ## Comment 7
>
> > The lifecycle framework implicitly assumes unconstrained iteration, but the paper never addresses how computational budgets affect which components of the loop are tractable, which seems to be a major practical omission.
>
> **Response:** We added a new subsection, **Section 7.7 (Budget Constraints)**, discussing computational budgets as a practical constraint on self-improvement systems. It identifies two compounding cost sources: iterated compute costs, since every stage of the loop is repeated at each round while many widely used methods are themselves computationally demanding and per-round gains typically diminish; and strong-model dependence, since the quality of self-generated supervision is bounded by the base model's capability, pushing practitioners toward larger models whose cost multiplies every call in the loop. This clarifies that self-improvement is not only an algorithmic challenge but also a resource-allocation problem.

---

> > ### Comment · Reviewer_VKFb · 2026-07-13
> > **Thank you for your edits and comments.**
> >
> > I thank the authors for their responses and edits. I do not have anymore comments.

---

> > > ### Author Response · Authors · 2026-07-13
> > > **Thank You for Your Feedback**
> > >
> > > Thank you again for your time and constructive feedback. We are glad our responses addressed your concerns.

---

> ### Author Response · Authors · 2026-07-12
> **Response to Reviewer VKFb (3/3)**
>
> ## Comment 8
>
> > Section 4.4 on theoretical guarantees for self-improvement is narrow in scope and disconnected from the rest of the paper, with no analogous theoretical treatment offered for data selection, inference refinement, or autonomous evaluation.
>
> **Response:** We reorganized this material so that it is integrated rather than standalone. Instead of a separate section, the theoretical analysis is now folded **into the Discussion of self-generated optimization (SGO, replacing our former GRO framework) as Section 4.5.1 ("Theoretical Analysis")**, presented next to the loop it analyzes. We rewrote its opening to connect it to the preceding subsections, whose methods all share the premise that a model can improve by training on its own generations. We also broadened its coverage by regrouping the body into three short paragraphs, each addressing a distinct question the theory raises: Source of Improvement (where the gains come from), Convergence and Capability Limits (whether the loop converges and to what ceiling), and Collapse and Sustainability (when it breaks down), drawing on recent analyses [1][2]. Finally, we linked it forward: the practical failure modes these theoretical insights imply are discussed in Section 7, particularly flawed feedback signals (7.2) and optimization-driven failures (7.3).
>
> We would also clarify why the theoretical analysis appears under model optimization and not the other stages. Model optimization is the only stage that actually updates parameters through an optimization algorithm, which makes formal questions such as convergence and capability limits well-posed and has drawn a dedicated line of theory. The other stages are comparatively more engineering-oriented and admit little formal analysis. For example, data acquisition is largely about building data-preparation and synthesis pipelines [3][4] rather than establishing provable guarantees. The theory therefore concentrates on optimization by the nature of the stages.
>
> [1] Peer-Predictive Self-Training for Language Model Reasoning
>
> [2] On the Generalization Gap in Self-Evolving Language Model Reasoning
>
> [3] Can LLMs Clean Up Your Mess? A Survey of Application-Ready Data Preparation with LLMs
>
> [4] EigenData: A Self-Evolving Multi-Agent Platform for Function-Calling Data Synthesis, Auditing, and Repair

---

### Review · Reviewer_KqMV · 2026-06-25

**Summary Of Contributions:**

This survey provides a comprehensive overview of self-improvement for large language models. It describes the main components of the pipeline: data acquisition, data selection, model optimization, inference refinement, and autonomous evaluation, and provides a discussion of the links between each of them as well as potential improvement towards a truly autonomous self-improvement loop. For each subgroup, the focus is on the most common methods and the trade-offs they entail among performance, computational cost, and practicality. Finally, the authors provide a detailed discussion on the challenges of the current pipeline, offering insights for future work, and discuss the applications of self-improvement techniques, from code to science, passing by maths and finance.

**Additional Comments:**

Explain methods acronyms when possible or define them for non-expert readers. E.g., p. 46, "T-FFN", p. 47 "MAS".

**Audience:**

Yes

**Audience Explanation:**

As said above, the current survey is very complete at a time when most work on self-improvement focuses on a specific part of the pipeline (data acquisition, data selection, model optimization, inference refinement, or autonomous evaluation). This submission provides a global perspective with clear and insightful connections between each part of the pipeline, but also a comprehensive description of the challenges and potential applications. I believe this can be very useful to practitioners and readers of TMLR publications and beyond.

**Broader Impact Concerns:**

Given the impact of such methods on a wide range of fields, more discussion on safety, ethical considerations, and potential disruption would be valuable (e.g., using self-improvement for medicine or finance). Even though the trade-off between performance and computational costs is done in separate sections, a global view of the potential impact regarding resources and token budget would be insightful, as well as ways to cut the self-improvement pipeline by users (how to? when to detect something is off, either in terms of performance of behavior? how to properly define the access of the self-improvement modules, say, to a group of private codebases?).

**Claims And Evidence:**

Yes

**Claims Explanation:**

The current survey does not make novel technical contributions since it is mostly a survey. It focuses on self-improvement for large language models and provides a detailed and comprehensive description of current techniques for the entire self-improvement pipeline (data acquisition, data selection, model optimization, inference refinement, and autonomous evaluation). For each category of methods, the authors provide one level of taxonomy, with a clear connection to the global pipeline. Finally, a discussion section on current gaps and potential improvements is provided for each section of the paper. The outcome is an insightful survey covering a huge part of the literature on self-improvement, allowing non-expert readers to grasp the main challenges and giving expert readers a good positioning with respect to the literature and high-level ideas for future work.

**Requested Changes:**

*Relevant work*

I list below recent work I believe should be discussed in the current survey.
- In the Data selection optimization, [1] is an interesting work on how to select pretraining data using a bilevel optimization framework (to put in section 3.3.2).
- In the Model optimization section, it would have been interesting to discuss HGM [2] and Hyperagents [3] (to put in sections 4.5 and 5.4.4)
- In the Inference refinement section, [4] is a seminal work on multi-token predictions and should be mentioned alongside Santilli et al. (to put in section 5.2.4)
- In the Autonomous evaluation section, [5, 6] could be mentioned (to put in section 6.3.1). Those are the GAIA (and GAIA2) benchmarks that evaluate LLMs agents on their ability to perform tasks (of varying difficulty) sequentially, notably tasks easy for humans but that require a non-trivial amount of steps, and tool calls, and a specific order to respect (example: visual understanding, tool calling, formatting answer).


*Typo*

I list below potential typos.
- p. 28: "Finally, some reward is based" -> "Finally, some rewards are based"
- p.32/33, p.50, section 4.5: missing parentheses in references. E.g. "The Godel Agent Yin et al. (2025)" -> "The Godel Agent (Yin et al., 2025)"

*References*
- [1] Grangier et al. Adaptive Training Distributions with Scalable Online Bilevel Optimization. TMLR 2024
- [2] Wang et al. Huxley-G\"odel Machine: Human-Level Coding Agent Development by an Approximation of the Optimal Self-Improving Machine. ICLR 2026
- [3] Zhang et al. HyperAgents. arXiv 2026
- [4] Gloeckle et al. Better & faster large language models via multi-token prediction. ICML 2024
- [5] Mialon et al. GAIA: a benchmark for General AI Assistants.
- [6] Froger et al. Gaia2: Benchmarking LLM Agents on Dynamic and Asynchronous Environments. ICLR 2026

---

> ### Author Response · Authors · 2026-07-12
> **Response to Reviewer KqMV**
>
> ## Comment 1
>
> > Please discuss the suggested recent work in relevant sections.
>
> **Response:** Thank you for these recommendations. We have added all six works to the indicated sections:
>
> - [1] Grangier et al. (2024) — **Section 3.3.2**: pretraining data selection as scalable online bilevel optimization that reweights the training distribution jointly with the model.
> - [2] Huxley-Gödel Machine (2026) — **Section 4.4**: guiding self-modification by clade metaproductivity rather than current benchmark performance.
> - [3] HyperAgents (2026) — **Section 5.4.4**: merging task and meta agent into one self-modifiable "hyperagent" that also improves its own ability to improve.
> - [4] Gloeckle et al. (2024) — **Section 5.2.4**, alongside Santilli et al.: multi-token prediction for efficient, self-speculative decoding.
> - [5, 6] GAIA (2023) / GAIA2 (2026) — **Section 6.3.1**: benchmarks for agents on multi-step, tool-using tasks, with GAIA2 adding dynamic and asynchronous environments.
>
> ## Comment 2
>
> > Potential typos: "some reward is based" and missing parentheses in references.
>
> **Response:** We carefully proofread the entire manuscript and corrected all reported typos:
>
> - **Section 4.3.2** — fixed the grammatical error "some reward is based" → "some rewards are based."
> - **Section 4.4** — fixed the missing parentheses in author–year citations, e.g., "The Gödel Agent Yin et al. (2025)" → "The Gödel Agent (Yin et al., 2025)."
>
> ## Comment 3
>
> > Broader impact concerns: safety, ethical considerations, disruption, resources, and access control.
>
> **Response:** We added a new section, **Section 8 (Potential Risks)**, that specifically discusses the potential risks of self-improvement systems along four dimensions (Figure 11): (i) Loss of Control, (ii) High-Stakes Harm, (iii) Social and Ethical Risks, and (iv) Misuse Risks, and highlights how the autonomous and recursive nature of the improvement loop can amplify them. This gives readers a structured lens for anticipating these risks and for informing the design of appropriate safeguards, oversight, and access control.
>
> ## Comment 4
>
> > Please explain method acronyms when possible or define them for non-expert readers.
>
> **Response:** We now spell out method acronyms at their first appearance for non-expert readers, here is one summary of changes:
>
> - **Section 1** — introduced RLHF as "Reinforcement Learning from Human Feedback."
> - **Section 2.4.1** — introduced CoT as "chain-of-thought."
> - **Section 3.3.2** — introduced UCB as "upper confidence bound."
> - **Section 4.4** — wrote out "Markov decision process" in full (used once, so no acronym introduced).
> - **Section 5.2.2** — wrote out "breadth-first/depth-first search" in full (used once).
> - **Section 5.2.3** — introduced RM as "reward model."
> - **Section 5.4.4** — introduced T-FFN as "temporal feed-forward network" and MAS as "multi-agent systems," and wrote out "directed acyclic graph" in full (used once).

---

### Review · Reviewer_ZHwA · 2026-06-28

**Summary Of Contributions:**

This paper presents a systematic survey on the important direction of LLM self-improvement. The authors propose   a unified lifecycle framework that decomposes self-improvement systems into five components: Data Acquisition, Data Selection, Model Optimization, Inference Refinement, and Autonomous Evaluation. In the Data Acquisition  component, data acquisition mechanisms are categorized into three levels: Static Curation, Environment  Interaction, and Synthetic Generation. The Data Selection component provides a systematic overview ranging from   Metric-Guided Scoring to Adaptive Selection. For Model Optimization, the authors introduce a  Generation–Reward–Optimization (GRO) framework to unify various self-improvement optimization methods. The  Inference Refinement component covers sampling, decoding strategies, reasoning-based improvement, agentic  system-based improvement, and test-time training. Finally, the Autonomous Evaluation component discusses  Dynamic Benchmarking and Interactive Environment Evaluation. The paper also addresses Challenges and  Limitations across six dimensions, including Data Autophagy, Flawed Feedback Signals, and Optimization-Driven  Failures, and concludes with Applications and Future Outlook.

Strengths:

1. This paper addresses a highly important and timely research direction. LLM self-improvement is one of the  central topics in current AI research. Particularly as LLMs have evolved to their current level of capability,  paradigms such as AutoResearch and self-evolving systems have become research hotspots with significant  academic and practical value. A systematic survey of this nature is therefore of considerable merit.
2. The paper provides a well-structured synthesis of self-improvement methods, integrating Data Acquisition,  Data Selection, Model Optimization, Inference Refinement, and Autonomous Evaluation into a coherent closed-loop   pipeline.
3. The literature coverage is impressively broad, citing a substantial body of recent work from 2024–2026,  including cutting-edge methods such as AbsoluteZero, R-Zero, and AlphaEvolve. The resulting taxonomy is  relatively comprehensive.

Weaknesses:
1. The paper would benefit from a more thorough analysis of the interdependencies across stages, which would  better substantiate the claimed "closed-loop lifecycle" and "tightly coupled processes." For instance, how do  specific Data Selection strategies concretely affect Model Optimization outcomes?
2. The relationship between the GRO framework and established paradigms such as reinforcement learning (RL) and   self-training deserves further clarification. What are the key distinctions and overlaps?
3. The scope of the Inference Refinement section is overly broad, encompassing a wide range of inference  efficiency methods that are in tension with the core definition of self-improvement.

**Audience:**

Yes

**Audience Explanation:**

This paper addresses a highly important and timely research direction. LLM self-improvement is one of the central topics in current AI research.

**Claims And Evidence:**

Yes

**Claims Explanation:**

The paper provides a well-structured synthesis of self-improvement methods with thorough coverage of related work.

**Requested Changes:**

1. Please add illustrative examples demonstrating the interactions and synergistic effects across different  components.
2. Please discuss the similarities and differences between the GRO framework and RL / self-training methods.
3. Please condense the Inference Refinement section by removing content that is not directly relevant to  self-improvement.

---

> ### Author Response · Authors · 2026-07-12
> **Response to Reviewer ZHwA (1/3)**
>
> ## Comment 1
>
> > The paper would benefit from a more thorough analysis of the interdependencies across stages, which would better substantiate the claimed "closed-loop lifecycle" and "tightly coupled processes." For instance, how do specific Data Selection strategies concretely affect Model Optimization outcomes?
>
> **Response:** We added a subsection at the end of each component's discussion to make the lifecycle connections explicit, anchoring each stage to whatever it most naturally connects to along the data -> model -> inference lifecycle and explaining its role in the loop as a whole:
>
> - In **Section 2.5.4**, we position data acquisition as the entry point of the loop, since every later stage depends on the data acquired here before it has anything to operate on.
> - In **Section 3.4.3**, we discuss the synergistic effects of data selection and data acquisition, pointing out that because the initially acquired data is inevitably noisy and uneven in quality, a downstream data selection stage is required to keep only the samples worth learning from.
> - In **Section 4.5.5**, we ground model optimization in data, explaining that the data handed over by acquisition and selection fixes the input to optimization and thereby bounds the capability it can produce.
> - In **Section 5.6.3**, we frame inference refinement and model optimization as two sides of one lifecycle. A stronger optimized model yields better test-time behavior, including stronger planning, reflection, and agentic performance; and conversely, given a fixed optimized model, allocating more inference-time compute to these agentic loops (longer search, more tool calls, or more self-correction) further improves agentic performance, a form of test-time scaling that adds capability without further training.
> - In **Section 6.4.3**, we treat autonomous evaluation as a global component that observes the system through its inference behavior and, by locating where the system underperforms, sets the direction of the next iteration, for example pointing data acquisition at the task types the system still fails on, thereby closing the loop.

---

> ### Author Response · Authors · 2026-07-12
> **Response to Reviewer ZHwA (2/3)**
>
> ## Comment 2
>
> > The relationship between the GRO framework and established paradigms such as reinforcement learning (RL) and self-training deserves further clarification. What are the key distinctions and overlaps?
>
> **Response:** We realize the GRO framework may not be comprehensive enough to cover our idea of different types of optimization that can be used in the self-improvement loop, raising the confusion on overlaps with general RL including self-training. In the recursive improvement system, after data is acquired and selected, model optimization can take different forms depending on the data formats and whether supervision signals are already available. These include the standard supervised fine-tuning, when the system curates fully annotated demonstration data, self-generation based RL approaches including self-training, when the data does not contain dense labels, and meta optimization approaches, when the optimization goes beyond the parameters themselves but also include model architectures etc.
>
> We reframe the framework for optimization and reorganize **Section 4** to encompass different optimization approaches, including those beyond RL and self-training.
>
> - **Direct Optimization:** the model is updated by an offline pass using SFT for example, over a corpus already with full labels assembled by the upstream stages (data acquisition and selection)
> - **Self-Generated Optimization (SGO):** the primary focus, which retains most of the content of our original GRO framework; it is a closed loop in which the model repeatedly generates the experience it trains on and supplies its own reward, then updates on the result, so the training distribution is policy-induced and non-stationary.
> - **Self-Evolving Optimization:** the model goes beyond parameter updates to improve the optimization process itself: its update rule, architecture, or agent-level scaffolding, lifting the unit of improvement to the whole agentic system.
>
> SGO continues the idea of the previous GRO framework, decomposing the loop into generation, reward, and optimization. Clearly, this loop is structurally the same as reinforcement learning: rolling out the policy is generation, the feedback is the reward, and the parameter update is the optimization. Among the three paradigms above, SGO received the most study in prior works around self-improvement optimization since it is the most challenging paradigm when human supervision or fully annotated data is removed. Although SGO is essentially a specialized form of RL in a self-contained recursive optimization system (we added the clarification in **Section 4.1** SGO definition in the bullet points), we still single it out to include different types of detailed algorithms within this scope that are proposed for self-improvement. This is also where we spend more space covering the technical details around the algorithms.
>
> We have revised the draft to reflect the updated framework, with additional discussions for more clarity on the overall picture and the detailed connections of different optimization paradigms:
>
> - **Section 4.1 (Overview):** We reframed the section around three optimization paradigms (Direct, SGO, Self-Evolving) and clarified that SGO is a specialized form of RL.
> - **Section 4.2 (Direct Optimization):** We added a paradigm covering single-pass offline updates on an upstream-assembled labeled corpus from data acquisition and selection.
> - **Section 4.3 (Self-Generated Optimization, SGO):** We renamed and reorganized the previous GRO framework as SGO and kept the generation–reward–optimization loop.
> - **Section 4.4 (Self-Evolving Optimization):** We renamed and reframed the original "Beyond GRO" section as a distinct paradigm called Self-Evolving Optimization, which improves the optimization process itself, beyond parameter updates.
> - **Section 4.5 (Discussion):** We stepped back from individual methods to discuss the theory behind the SGO loop, the axes along which its instances vary, the trends across the three paradigms, and how optimization connects to the rest of the pipeline.

---

> ### Author Response · Authors · 2026-07-12
> **Response to Reviewer ZHwA (3/3)**
>
> ## Comment 3
>
> > The scope of the Inference Refinement section is overly broad, encompassing a wide range of inference efficiency methods that are in tension with the core definition of self-improvement.
>
> **Response:** We identified the **Decoding Strategies subsection (Section 5.2)** as the main place where relevance to self-improvement was insufficiently clear. To solve this:
>
> First, since decoding is such a broad topic, we made the connection between the specific strategies we include and self-improvement explicit:
>
> - **Section 5.2** now opens by framing decoding explicitly as inference-time self-improvement, and organizes the discussion around the two roles decoding plays in the loop: (i) surfacing and selecting higher-quality outputs from the model's own distribution, which both improves responses at inference and yields better samples that can be fed back as training data for the optimization stage, and (ii) keeping the repeated generations required by iterative self-improvement efficient at scale.
> - We added a framing sentence to each subsubsection stating its role under this view: **Sampling-Based (5.2.1)** samples candidates and selects or combines them into a more reliable output; **Tree Search (5.2.2)** systematically explores and compares reasoning paths to select higher-quality outputs; **Logit and Probability Adjustments (5.2.3)** reshapes token probabilities using a signal the model itself produces; and **Efficiency-Oriented Methods (5.2.4)** accelerate the repeated generations of the self-improvement loop, which can be used for both inference time and training time when model self-generation is required.
>
> Second, we condensed content with only weak ties to self-improvement **(Section 5.2)**:
>
> - Removed the constrained decoding paragraph.
> - Merged the separate paragraphs on model logit blending and contrastive decoding into a single compact discussion of reference-guided decoding **(in 5.2.3)**.
>
> Therefore, the revised subsection is more focused on how decoding supports search, selection, and efficient test-time improvement within the self-improvement lifecycle.

---

### Author Response · Authors · 2026-07-12
**Revision Summary**

Dear Action Editor and Reviewers,

We thank the reviewers for their constructive feedback. We have uploaded a revised manuscript with all changes marked **in blue**; point-by-point replies follow separately. The main revisions:

## Changes Requested by Reviewers

1. **Made the closed-loop connections between sections more explicit.** We added a connecting subsection at the end of five main sections that shows how it builds on and feeds the others in the whole self-improvement system (`Reviewer ZHwA`):
   - **Section 2.5.4**: Initiation of the Self-Improvement Loop (Data Acquisition)
   - **Section 3.4.3**: Interaction with Data Acquisition (Data Selection)
   - **Section 4.5.5**: Interplay Between Optimization and Data (Model Optimization)
   - **Section 5.6.3**: Synergy with the Optimized Model (Inference Refinement)
   - **Section 6.4.3**: Oversight of the Overall Self-Improvement System (Autonomous Evaluation)

2. **Clarified the optimization framework vs. RL / self-training:** reorganized **Section 4** into three paradigms (direct, self-generated, self-evolving) and reframed GRO as Self-Generated Optimization (SGO), a specialized form of RL (`Reviewer ZHwA`).

   We have revised the draft to reflect the updated framework, with additional discussions for more clarity on the overall picture and the detailed connections of different optimization paradigms:
   - **Section 4.1 (Overview):** We reframed the section around three optimization paradigms (Direct, SGO, Self-Evolving) and clarified that SGO is a specialized form of RL.
   - **Section 4.2 (Direct Optimization):** We added a paradigm covering single-pass offline updates on an upstream-assembled labeled corpus from data acquisition and selection.
   - **Section 4.3 (Self-Generated Optimization, SGO):** We renamed and reorganized the previous GRO framework as SGO and kept the generation–reward–optimization loop.
   - **Section 4.4 (Self-Evolving Optimization):** We renamed and reframed the original "Beyond GRO" section as a distinct paradigm called Self-Evolving Optimization, which improves the optimization process itself, beyond parameter updates.
   - **Section 4.5 (Discussion):** We stepped back from individual methods to discuss the theory behind the SGO loop, the axes along which its instances vary, the trends across the three paradigms, and how optimization connects to the rest of the pipeline.

3. **Gave the framework in optimization a precise definition:** added Algorithm 1 formalizing the SGO loop (**Section 4.3**) (`Reviewer VKFb`).

4. **Condensed Inference Refinement:** tightened the Decoding Strategies subsection (**Section 5.2**) to focus on self-improvement and removed weakly-related content (`Reviewer ZHwA`).

5. **Strengthened the Autonomous Evaluation discussion:** added a subsection on the scalability of autonomous evaluation (**Section 6.4.2**), addressing the concern that this component was comparatively thin (`Reviewer VKFb`).

6. **Distinguished self-improving base LLMs from self-evolving agentic systems** more explicitly in the introduction (**Section 1**) (`Reviewer VKFb`).

7. **Added the recommended related work** to the indicated locations (**Sections 3.3.2, 4.4, 5.4.4, 5.2.4, and 6.3.1**) (`Reviewer KqMV`).

8. **Added a new section (Section 8), Potential Risks,** on safety, high-stakes harm, social/ethical risks, and misuse (`Reviewer KqMV`, `Reviewer VKFb`).

9. **Addressed the empirical-comparison point:** acknowledged the difficulty and expanded a unified benchmark as future work (**Section 10**) (`Reviewer VKFb`).

10. **Reconnected the theoretical analysis:** integrated it into the discussion (**Section 4.5.1**) with an opening that motivates it and forward links to the failure-mode discussion (`Reviewer VKFb`).

11. **Added a discussion of budget constraints:** a new subsection **(Section 7.7)** under Challenges and Limitations on how compute budgets practically constrain the self-improvement loop (`Reviewer VKFb`).

12. **Clarified the autonomy/continuity definition** in the introduction **(Section 1)** (`Reviewer VKFb`).

13. **Spelled out method acronyms at first use**, e.g., T-FFN, MAS (`Reviewer KqMV`).

14. **Fixed the reported typos** (`Reviewer KqMV`).

## Additional Changes on Our Own

1. **Added a discussion of trends across the three optimization paradigms** (**Section 4.5.4**), framing direct, self-generated, and self-evolving optimization as a progression in which each hands successively more of the pipeline over to the model.

2. **Refined Figures 1 and 2** to make them clearer and stylistically more consistent with the other figures.

3. **Ran a full consistency and typo check** (e.g., terminology and acronyms).

4. **Restructured the Applications section (Section 9)** by adding Auto Research as a new domain, reflecting the growing interest in agents that autonomously conduct research, and folding the original Algorithm domain into Science.

Best,

The Authors